# Host-selected mutations converging on a global regulator drive an adaptive leap towards symbiosis in bacteria

M Sabrina Pankey[1,2†], Randi L Foxall[1,2†], Ian M Ster[1,2,3], Lauren A Perry[1,4], Brian M Schuster[1‡], Rachel A Donner[1], Matthew Coyle[1,4], Vaughn S Cooper[2‡], Cheryl A Whistler[1,2*]

[1]Department of Molecular, Cellular and Biomedical Sciences, University of New Hampshire, Durham, United States; [2]Northeast Center for Vibrio Disease and Ecology, College of Life Science and Agriculture, University of New Hampshire, Durham, United States; [3]Graduate Program in Biochemistry, University of New Hampshire, Durham, United States; [4]Graduate Program in Microbiology, University of New Hampshire, Durham, United States

*For correspondence: cheryl.whistler@unh.edu

[†]These authors contributed equally to this work

Present address: [‡]Microbiology and Molecular Genetics, University of Pittsburgh School of Medicine, Pittsburgh, United States

Competing interests: The authors declare that no competing interests exist.

**Abstract** Host immune and physical barriers protect against pathogens but also impede the establishment of essential symbiotic partnerships. To reveal mechanisms by which beneficial organisms adapt to circumvent host defenses, we experimentally evolved ecologically distinct bioluminescent *Vibrio fischeri* by colonization and growth within the light organs of the squid *Euprymna scolopes*. Serial squid passaging of bacteria produced eight distinct mutations in the *binK* sensor kinase gene, which conferred an exceptional selective advantage that could be demonstrated through both empirical and theoretical analysis. Squid-adaptive *binK* alleles promoted colonization and immune evasion that were mediated by cell-associated matrices including symbiotic polysaccharide (Syp) and cellulose. *binK* variation also altered quorum sensing, raising the threshold for luminescence induction. Preexisting coordinated regulation of symbiosis traits by BinK presented an efficient solution where altered BinK function was the key to unlock multiple colonization barriers. These results identify a genetic basis for microbial adaptability and underscore the importance of hosts as selective agents that shape emergent symbiont populations.

## Introduction

Identifying traits that are under selection by hosts is crucial to understanding the processes governing nascent symbiotic interactions between animals and microbes. The remarkable efficiency with which some bacteria evolve variation that enhances access to novel host niches indicates that adaptability may be an attribute of some bacterial genomes. Adaptive evolution to a new niche, such as a novel host, may involve reconciliation of constraints imposed by genomic content, conflicting regulation, and pleiotropy (*Morley et al., 2015*; *Bedhomme et al., 2012*). Given this context, global regulators could serve as effective targets of selection that drive adaptive leaps made by pathogenic or mutualistic microbes, as long as essential metabolic pathways are both sufficiently insulated from detrimental effects of mutation and available for integration with accessory functions (*Davenport et al., 2015*; *Wolfe et al., 2004*; *Jansen et al., 2015*). Studies using experimental evolution have often revealed that adaptive evolution initially proceeds through regulatory changes, but few have identified the underlying mechanisms that promote adaptation or linked these processes to natural symbiotic systems (*Morley et al., 2015*; *Bedhomme et al., 2012*; *Kawecki et al., 2012*; *Marchetti et al., 2010*; *Guan et al., 2013*).

**eLife digest** Most bacteria that associate with animals do not cause harm, and many are essential to health or provide other benefits. An animal's immune system must permit these beneficial associations and at the same time block harmful microbes. This ultimately means that even beneficial bacteria must adapt to the immune barriers that they encounter.

Different species that live in a close relationship with each other are known as symbionts. A species of bacteria called *Vibrio fischeri* can form a mutually beneficial symbiotic relationship with squid. The squid provide food for the bacteria, but only the bacteria that successfully navigate immune barriers and reach the squid's "light organ" are fed. In return, the bacteria produce bioluminescence, making the nocturnal squid appear like moonlight in the water.

As the bacteria reproduce, some individuals randomly acquire genetic mutations, some of which might improve the bacteria's chances of survival. Which mutations and associated traits allow bacteria to beat out the competition and evolve to become animal symbionts? To investigate, Pankey, Foxall *et al.* grew *V. fischeri* bacteria from several ancestors that were poor at colonizing squid. Groups of newly hatched squid selected potential symbionts from the resulting mix of bacteria. The selected symbionts were allowed to reproduce within the squid to form a new population of bacteria and were later vented out for a new batch of squid to sort through. This was repeated to ultimately form a final group of bacteria that had passed through 15 squid in turn.

Unexpectedly, the bacteria in the final group all found the same solution to help them adapt to symbiotic life with the squid: mutations to the gene that encodes a signaling protein called BinK. Eight distinct mutations arose that dramatically changed how the bacteria interacted with squid. The evolved bacteria created a coating that hid them from squid immune cells and protected them from chemicals that squid use to kill invaders. The mutations also altered how the bacteria communicated with each other. This adjusted the intensity of light that they produced for their host to a more natural level, and improved their ability to grow on squid-provided food.

Overall, the results presented by Pankey, Foxall *et al.* demonstrate that small genetic mutations can transform non-symbionts into symbionts, enabling them to evolve rapidly to form a symbiosis with a new host. This demonstrates that these bacteria already had the ability to coordinate the complex behaviors necessary to overcome the multiple barriers provided to them by the squid immune system. Other beneficial animal–bacteria associations are likely to work on similar principles; the study exemplifies the utility of experimental evolution systems and lays a foundation for further work to investigate these principles in more detail.

Members of the genus *Vibrio*, halophilic bacteria with a broad distribution in marine and brackish environments, have repeatedly evolved to colonize varied host niches (*Nishiguchi, 2002*; *Takemura et al., 2014*; *Guerrero-Ferreira and Nishiguchi, 2007*), and as such, their study can provide an understanding of adaptability to host association. Bioluminescent *Vibrio fischeri* can be found among marine plankton (*Lee and Ruby, 1992*) but the species is best known for its mutualistic light organ symbiosis with squid and fish species. *V. fischeri* is also well-known for its social quorum-sensing behavior, whereby communities of bacteria use diffusible pheromone signal molecules to synchronize gene expression in response to cell density (*Schuster et al., 2013*; *Verma and Miyashiro, 2013*; *Waters and Bassler, 2005*). In squid-symbiotic *V. fischeri*, quorum sensing occurs through sequential activation by two different pheromone signals: the first signal (C8-HSL) 'primes' sensitive perception of the second signal (3-oxo-C6-HSL) through enhanced LitR activity, which increases the levels of the LuxR pheromone sensor, thereby lowering the threshold for signal perception (*Fidopiastis et al., 2002*; *Lupp and Ruby, 2004*; *Miyashiro et al., 2010*). In turn, when LuxR binds to 3-oxo-C6-HSL, LuxR homodimerizes and directly activates the expression of the *lux* bioluminescence operon to produce light, which squid use for counter-illumination camouflage during their nocturnal foraging behavior (*Lupp et al., 2003*; *Jones and Nishiguchi, 2004*).

The symbiotic association between *V. fischeri* and the squid *Euprymna scolopes* has become a powerful system for interrogating mechanisms underlying bacterial colonization of metazoan host mucosal surfaces where colonists must overcome host defenses that limit infection by non-symbiotic

bacteria, including pathogens (*Figure 1A*). Once newly hatched squid entrap bacteria in mucus near the light organ, symbionts aggregate in this mucus and, in response to host attractants, subsequently swim through pores at the entrance of the nascent light organs (*Nyholm et al., 2000*). As *V. fischeri* bacteria swim down the ducts and into the crypts, they face a 'gauntlet' of defenses that includes host-derived oxidative species (*Davidson et al., 2004*; *Weis et al., 1996*; *Small and McFall-Ngai, 1999*), as well as patrolling macrophage-like hemocytes that attach to other species of marine bacteria with higher affinity, subsequently killing these invading cells (*Nyholm et al., 2009*; *Nyholm and McFall-Ngai, 1998*; *Koropatnick et al., 2007*). These barriers ensure that only the

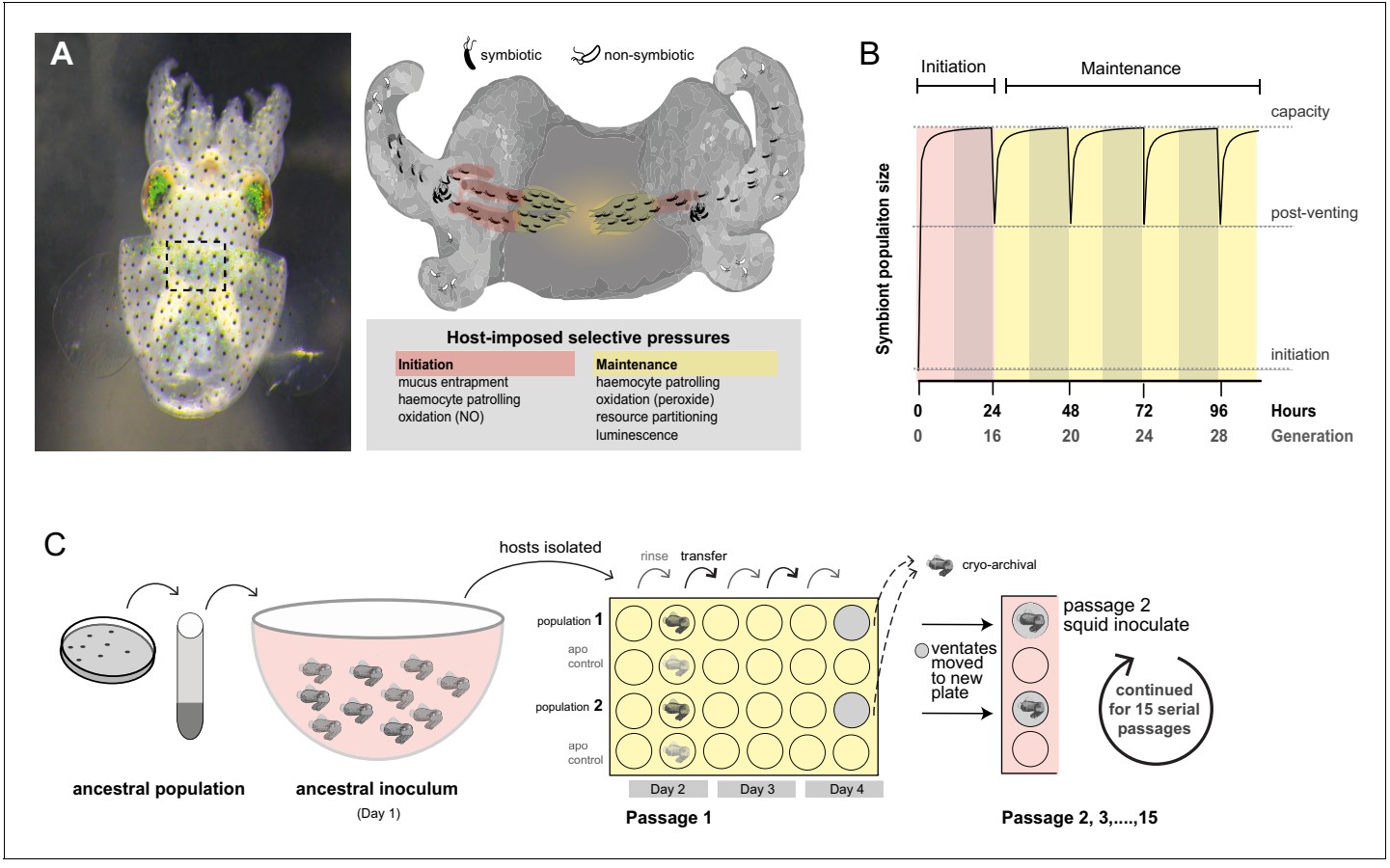

**Figure 1.** Host selection mechanisms that shape adaptive evolution by *V. fischeri*. (A) Dorsal view of juvenile host *E. scolopes* (left) with box indicating the relative position of the ventrally situated symbiotic light organ. On the right, a schematic illustrating the stages at which host-imposed selection occurs during squid–*V. fischeri* symbiosis: host recruitment (mucus entrapment, aggregation at light organ pores), initiation of symbiosis (host defenses, including hemocyte engulfment and oxidative stress), and colonization and maintenance (nutrient provisioning, sanctioning of non-luminous cheaters, continued hemocyte patrolling, and daily purging). (B) Symbiont population growth modeled for a single passage on the basis of growth dynamics of *V. fischeri* ES114. Light-organ populations are initiated with as few as ~10 cells (*Wollenberg and Ruby, 2009*; *Altura et al., 2013*) or as much as 1% of the inoculum, but are reduced by 95% following venting of the light organ at dawn (every 24 hr) (*Boettcher et al., 1996*). Shaded areas represent night periods whereas light areas represent daylight, which induces the venting behavior. (C) Experimental evolution of *V. fischeri* under host selection as described in *Schuster et al. (2010)*. Each ancestral *V. fischeri* population was prepared by recovering cells from five colonies, growing them to mid-log phase, and sub-culturing them into 100 mL filtered seawater at a concentration sufficient to colonize squid (≤20,000 CFU/mL). On day 1, ten un-colonized (non-luminous) juvenile squid were communally inoculated by overnight incubation, during which bacteria were subjected to the first host-selective bottleneck. Following venting of ~95% of the light organ population, the squid were separated into isolated lineages in individual wells of a 24-well polystyrene plate containing filtered sea water with intervening rows of squid from an un-inoculated control cohort, the aposymbiotc control ('apo control'). Note that only two of the ten passage squid populations are shown. On days 2, 3, and 4, after venting, squid were rinsed and transferred into 2 mL fresh filtered seawater. Luminescence was measured at various intervals for each squid to monitor colonization and the absence of contamination in aposymbiotic control squid. On the fourth day, the squid and half of the ventate were frozen at −80°C to preserve bacteria, and the remaining 1 mL ventate was combined with 1 mL of fresh filtered seawater, and used to inoculate a new uncolonized 24-hr-old juvenile squid. The process continued for 15 squid only for those lineages in which squid were detectably luminous at 48 hr post inoculation.

correct symbiotic partner gains access to the crypts where host-provided nutrients support bacterial growth (*Graf and Ruby, 1998*; *Heath-Heckman and McFall-Ngai, 2011*). Striking parallels between beneficial *V. fischeri* colonization and pathogenic infection suggest that the selective pressures exerted by animal hosts may act on a common repertoire of bacterial traits used to circumvent host defensive obstacles (*Nyholm and McFall-Ngai, 2004*).

Not all lineages of *V. fischeri* excel in symbiosis; this observation is reflective of the varied selective regimes that shape both genetic variation and adaptive potential as symbionts (*Lee and Ruby, 1994a*; *Nishiguchi et al., 1998*). In habitats where squid hosts are present, they influence local *V. fischeri* populations by enriching the planktonic community with those strains that are most adept at symbiosis (*Lee and Ruby, 1994b*). Squid recruit small founder populations (~10 bacteria) and subject these to daily cycles of expulsion ('venting') and regrowth of 95% of light organ populations to >$10^5$ bacteria (*Wollenberg and Ruby, 2009*) (*Figure 1B*), thereby increasing the relative abundance of their light organ inhabitants in the surrounding seawater (*Lee and Ruby, 1994b*). The bottlenecks within the venting cycle limit light organ microbial diversity, including variation that impairs symbiosis, such as 'cheaters' that do not contribute to the mutualism but benefit from symbiotic association (*Wollenberg and Ruby, 2009*; *Ruby and McFall-Ngai, 1999*; *Visick and McFall-Ngai, 2000*). However, host-imposed selection that drives the evolution of some lineages towards efficient colonization could hinder future adaptation and entail fitness trade-offs in other environments (*Soto et al., 2014*; *Caley and Munday, 2003*). So, by contrast, planktonic *V. fischeri* strains that reside in habitats without hosts, or that are unable to compete for prime host niches, may maintain greater adaptability while being ineffective as symbionts (*Takemura et al., 2014*). Deficiency in squid colonization correlates with insufficient or excessive luminescence or inadequate production of a symbiotic polysaccharide (known as Syp), which is controlled by a horizontally acquired activator (RscS) in squid native strain ES114 (*Nishiguchi et al., 1998*; *Yip et al., 2006*; *Mandel et al., 2009*). However, the absence of the *rscS* gene in some symbiotically proficient *V. fischeri* strains (and likewise, the presence of *rscS* in deficient strains) indicates that this regulator alone does not strictly determine squid colonization capacity (*Figure 1A*, *Figure 2—figure supplement 1*). Genomic similarity among closely related yet ecologically diverse strains has obscured relevant functional differences that are sometimes undetectable except in the symbiotic context (*Yip et al., 2006*; *Mandel et al., 2009*; *Travisano and Shaw, 2013*).

For this study, we conducted a series of evolution experiments in which hatchling squid select among *V. fischeri* populations for mutants that are capable of initiating symbiosis, of persisting in the light organ, and of colonizing new squid when purged from the light organ (*Schuster et al., 2010*). This cycle of host selection was designed to identify traits underlying symbiotic adaptive evolution and to reveal the evolutionary and genomic dynamics of this process. We chose as ancestors of our experimental lineages five *V. fischeri* strains that had variable aptitudes for squid symbiosis and were isolated from different niches, including the light organs of squid and fish, and various planktonic aquatic environments, including one without known hosts (*Table 1*). After we experimentally evolved replicate populations derived from each ancestor in parallel, we evaluated the genetic and phenotypic changes that occurred under host selection to examine how starting fitness and past evolutionary history influenced adaptability to squid symbiosis. To delineate the effects of host selection from neutral mutation accumulation, we also subjected *V. fischeri* to laboratory evolution in minimal seawater media. Previously, we demonstrated that altered luminescence was associated with several isolates following 15 serial host passages (*Schuster et al., 2010*). Here, we report the genetic basis of this adaptation as well as the population dynamics of the symbionts under host selection. Importantly, we also identify the precise traits under selection that enabled these early-sweeping mutants to bypass key barriers imposed by hosts.

## Results

### Squid experimental evolution of ecologically diverse *V. fischeri* repeatedly produced adaptive mutations in the *binK* sensor kinase gene

To study the dynamic process of adaptation during symbiosis, we capitalized upon the squid's natural recruitment process to found parallel populations of *V. fischeri*, and used the daily squid venting

**Table 1.** Strains and plasmids used in this study.

| Strain name | Description[*] | Reference/source |
|---|---|---|
| *Vibrio fischeri* strains[†] | | |
| ES114 | Isolated from *Euprymna scolopes* | (*Boettcher and Ruby, 1990*) |
| MJ11 | Isolated from *Monocentris japonica* light-organ | (*Haygood et al., 1984*) |
| EM17 | Isolated from *Euprymna morseii* light-organ | (*Ruby and Lee, 1998*) |
| H905 | Isolated from Hawaiian plankton | (*Lee and Ruby, 1992*) |
| WH1 | Isolated from Massachusetts plankton | (*Lee, 1994*) |
| RF1A4 | MJ11 Δ*binK*::*ermB*; Em$^R$ | This study |
| RF1A5 | MJ11 Δ*sypK*::*aphA1*; Km$^R$ | This study |
| RF1A6 | MJ11 Δ*binK*::*ermB* Δ*sypK*::*aphA1*; Em$^R$ Km$^R$ | This study |
| RF1A7 | MJ11 *binK1* Δ*sypK*::*aphA1*; Km$^R$ | This study |
| MJ11EP2-3-2 | MJ11 *binK4* | This study |
| MJ11EP2-3-3 | MJ11 *binK4* | This study |
| MJ11EP2-3-4 | MJ11 *binK4* | This study |
| MJ11EP2-3-5 | MJ11 *binK4* | This study |
| MJ11EP2-3-6 | MJ11 *binK4* | This study |
| MJ11EP2-3-7 | MJ11 *binK4* | This study |
| MJ11EP2-3-8 | MJ11 *binK4* | This study |
| MJ11EP15-3-1 | MJ11 *binK4* | This study |
| MJ11EP15-3-3 | MJ11 *binK4* | This study |
| MJ11EP15-3-4 | MJ11 *binK4* | This study |
| MJ11EP15-3-7 | MJ11 *binK4* | This study |
| MJ11EP15-3-8 | MJ11 *binK4* | This study |
| MJ11EP2-4-1 | MJ11 *binK1* | This study |
| MJ11EP2-4-3 | MJ11 *binK1* | This study |
| MJ11EP2-4-4 | MJ11 *binK1* | This study |
| MJ11EP2-4-5 | MJ11 *binK1* | This study |
| MJ11EP2-4-6 | MJ11 *binK1* | This study |
| MJ11EP15-4-1 | MJ11 *binK1 tadC1*$^{G593T}$ | (*Schuster et al., 2010*) |
| MJ11EP15-4-6 | MJ11 *binK1* | This study |
| MJ11EP15-4-7 | MJ11 *binK1* | This study |
| MJ11EP15-4-8 | MJ11 *binK1* | This study |
| MJ11EP2-5-2 | MJ11 *binK3* | This study |
| MJ11EP2-5-3 | MJ11 *binK3* | This study |
| MJ11EP2-5-4 | MJ11 *binK3* | This study |
| MJ11EP2-5-5 | MJ11 *binK3* | This study |
| MJ11EP2-5-6 | MJ11 *binK3* | This study |
| MJ11EP15-5-2 | MJ11 *binK4* | This study |
| MJ11EP15-5-3 | MJ11 *binK3* | This study |
| MJ11EP15-5-4 | MJ11 *binK3* | This study |
| MJ11EP15-5-5 | MJ11 *binK3* | This study |
| MJ11EP2-6-1 | MJ11 *binK2* | This study |
| MJ11EP15-6-1 | MJ11 *binK2* | (*Schuster et al., 2010*) |
| MJ11EP15-6-2 | MJ11 *binK2* | This study |
| MJ11EP15-6-3 | MJ11 *binK2* | This study |

*Table 1 continued on next page*

Sabrina Pankey *et al*. eLife 2017;6:e24414. DOI: 10.7554/eLife.24414

*Table 1 continued*

| Strain name | Description* | Reference/source |
|---|---|---|
| MJ11EP15-6-4 | MJ11 *binK2* | This study |
| MJ11EP15-6-5 | MJ11 *binK2* | This study |
| MJ11CE4-1 | MJ11 *fliA*$^{G80D}$ | This study |
| MJ11CE5-1 | MJ11 *fliP*$^{\Delta476}$ | This study |
| Strain name | Description* | Reference/source |
| *Escherichia coli* strains | | |
| DH5α | F⁻ *recA1 endA1 hsdR17 supE44 thi-1 gyrA96 relA1Δ (argF-lacZYA) U169φ 80lacZΔM15λ ⁻* | Gibco-BRL |
| DH5αλ*pir* | *supE44 ΔlacU169 (φlacZΔM15) recA1 endA1 hsdR17 thi-1 gyrA96 relA1*; λpir phage lysogen | (*Kolter and Helinski, 1978*) |
| CC118λ*pir* | *Δ(arg-leu) araD ΔlacX74 galE galK phoA20 thi-1 rpsE rpoB argE*(Am) *recA1*, lysogenized with λ *pir dam dcm* | (*Martín-Mora et al., 2016*) |
| NEB 10-beta | *Δ(ara-leu)7697 araD139 fhuA ΔlacX74 galK16 galE15 e14- Φ80dlacZΔM15 recA1 relA1 endA1 nupG rpsL* (Sm$^R$) *rph spoT1 Δ(mrr-hsdRMS-mcrBC)* | New England Biolabs, Ipswich, MA |
| TOP10 | F- *mcrA Δ(mrr-hsdRMS-mcrBC) Φ80lacZΔM15 ΔlacX74 recA1 araD139 Δ(ara-leu)7697 galU galK rpsL* (Sm$^R$) *endA1 nupG* | Invitrogen, Carlsbad, CA |
| Plasmids | | |
| pCR2.1-TOPO | Commercial cloning vector; Ap$^R$ Km$^R$ | Invitrogen, Carlsbad, CA |
| pVSV105 | Mobilizable vector; Ch$^R$ | (*Dunn et al., 2006*) |
| pRAD2E1 | pVSV105 carrying wild-type *binK*; Ch$^R$ | This study |
| pRF2A2 | pVSV105 carrying *binK1*; Ch$^R$ | This study |
| pCLD48 | pVSV105 carrying ES114 *sypE*; Ch$^R$ | (*Hussa et al., 2008*) |
| pRF2A3 | pVSV105 carrying MJ11 *binA*; Ch$^R$ | This study |
| pVSV104 | Mobilizable vector; Km$^R$ | (*Stabb and Ruby, 2002*) |
| pRF2A1 | pVSV104 carrying *sypE*; Km$^R$ | This study |
| pRF2A4 | pVSV104 carrying *binA*; Km$^R$ | This study |
| pKV111 | Mobilizable vector containing *gfp*; Ch$^R$ | (*Nyholm et al., 2000*) |
| pRF2B7 | pCR2.1-TOPO containing MJ11 Δ*sypK::aph1* SOE fragment; Km$^R$ | This study |
| pVSV103 | Mobilizable vector containing *lacZ*; Km$^R$ | (*Dunn et al., 2006*) |
| pCAW7B1 | pVSV103 containing *lacZ*Δ147–1080 bp; Km$^R$ | This study |

*Ap$^R$, ampicillin resistance; Ch$^R$, chloramphenicol resistance; Em$^R$, erythromycin resistance; Km$^R$, kanamycin resistance; Sm$^R$ streptomycin resistance.

†Experimentally evolved strains are designated 'MJ11EP#-#-#', where the first and second numbers after the 'P' designates the squid passage and population from which the strain was isolated, and the third number designates isolate number; strains derived from evolution in culture are designated 'MJ11CE'.

behavior to restrict and re-grow bacterial populations, which were passaged through 15 serial squid, encompassing 60 bottlenecking events and an estimated 290–360 generations (*Figure 1C*) (*Schuster et al., 2010*). Multiple populations were derived in parallel from each of five ancestral strains using high-density inocula, up to 10 times the concentration required for native strain colonization, in order to overcome the colonization deficiencies of squid-maladapted strains (*Figure 2A* and Materials and methods).

Genome sequencing of evolved isolates revealed that, although few detectable mutations arose during squid passaging, the majority of mutations that arose to a detectable frequency converged in a conserved gene (locus VF_A0360 in *V. fischeri* ES114) (*Figure 2A–B*, *Figure 2—figure supplement 1*, *Table 2*), which was recently identified as a biofilm inhibition kinase (*binK*) in the native symbiotic strain ES114 (*Brooks and Mandel, 2016*). Nine independent mutations mapping to the *binK* locus, most often without other co-occurring mutations, dominated multiple parallel evolved populations of the two strains initially most impaired at squid symbiosis: MJ11 and H905 (*Figure 2A*, *Table 2*).

Given that MJ11 is a fish symbiont that lacks *rscS*, and H905 is a planktonic isolate from the squid habitat that is a poor squid colonizer despite harboring *rscS*, starting fitness better predicted the path of evolution than *rscS* content or past evolutionary history as inferred by either lineage or life-style (*Figure 2A*, *Figure 2—figure supplement 1*) (*Mandel et al., 2009*; *Lee and Ruby, 1994a*). By contrast, very few mutations, all at unique loci, occurred in representative isolates derived from strains EM17 (an *Euprymna morsei* squid symbiont) and WH1 (a planktonic strain from an environment without known hosts) (*Figure 2A*, *Table 2*). Both of these strains have relatively greater starting fitness than MJ11 and H905, further demonstrating that starting symbiont fitness influences its evolutionary path (*Wang et al., 2016*). Finally, mutations were not detected in any of the representative isolates from the native squid symbiont ES114 (*Figure 2A*, *Table 2*), even though several mutations are known to improve its competitive dominance (*Fidopiastis et al., 2002*; *Brooks and Mandel, 2016*). Laboratory-culture evolution of strain MJ11 that mimicked the population dynamics of squid-induced bottlenecks produced few mutations except for those localizing to flagellar genes *fliA* and *fliP* (*Table 2*).

To examine more thoroughly the evolutionary process giving rise to the convergent *binK* mutations, we focused on lineages derived from the fully sequenced and relatively well-characterized fish symbiont MJ11. Only five of ten squid exposed to the same inoculum population successfully passaged symbionts to the second recipient squid, and each successful lineage harbored *binK* variants (*Table 2*). Among these were four unique alleles wherein the acquired substitutions mapped to two of the five conserved functional domains of the deduced BinK protein (*Figure 2B*, *Table 2*). Despite standing variation in *binK* across *V. fischeri* strains, the four point mutations in experimentally evolved MJ11 lineages occurred at positions that, with the exception of *binK3* (S311L), are invariant in natural strains and thus are likely to represent novel allelic variants that are not convergent with the native symbiont (*Figure 2B*). Further analysis of the acquired mutations using a position-specific scoring matrix (PSSM) also provided evidence that the mutations in *binK1* (R537C), *binK2* (K482N) and *binK3* (S311L) would influence protein function (*Figure 2B*). In each of the five successful squid-evolved lineages of MJ11, *binK* variants dominated the light-organ populations by the third experimental squid (*Table 2*). If beneficial variants in this or any other locus were among the remaining five light-organ populations, their failure to colonize the second experimental squid amounted to early extinction of these lineages.

## The large selective advantage conferred by squid-adapted *binK* improved fitness during both the initiation and maintenance stages of symbiosis, consistent with theoretical predictions

The repeated sweeps of novel *binK* mutations that occurred during squid evolution, but not during laboratory culture evolution, suggested that *binK* variants were squid-adaptive (*Table 2*) (*Dillon et al., 2017*). To evaluate the contribution of evolved *binK* alleles specifically to improved symbiotic colonization, we assessed the colonization efficiency of the squid-evolved isolates and the ancestor using inoculum doses typically used for the native symbiont strain ES114 (*Figure 2A*). Each squid-evolved *binK* variant vastly improved colonization efficiency (*Figure 3A*), but they were not significantly more fit in laboratory culture (which would be indicative of mutants enhancing general vigor) when compared to ancestral MJ11 (*Figure 3B*). Moreover, whereas two of the five culture-evolved populations of MJ11 evolved culture-adaptive flagellar mutations that improved fitness in culture (*Figures 2A* and *3B*, *Table 2*), none accrued *binK* mutations (*Table 2*) or improved as squid symbionts (*Figure 3A*). Evolved isolates that have mutations mapping to different *binK* domains were competitively indistinguishable from each other in symbiotic fitness (permutation t-test, p=0.348) (*Figure 3—figure supplement 1*), despite evidence that the *binK1* allele (encoding an R537C substitution, *Figure 2B*, *Table 2*) appeared slightly more efficient at squid colonization when singly inoculated (*Figure 3A*).

To quantify empirically the selective advantage (selective coefficient: *s*) conferred by a representative *binK* allele that arose to early dominance before co-occurring mutations, we co-inoculated squid with MJ11 and low densities of a *binK1* variant (a fully sequenced second passage squid isolate that we named MJ11EP2-4-1, see *Tables 1* and *2*), simulating the conditions under which we predict the variants evolved given the low mutation rate of *V. fischeri* (*Dillon et al., 2017*) (*Figure 4A–B*). These experiments revealed that even at an extremely low frequency (*e.g.*, one *binK1* variant per 10,000 wild-type MJ11 bacteria, which amounted to only 50 *binK1* variant cells in an $10^4$ CFU•ml$^{-1}$

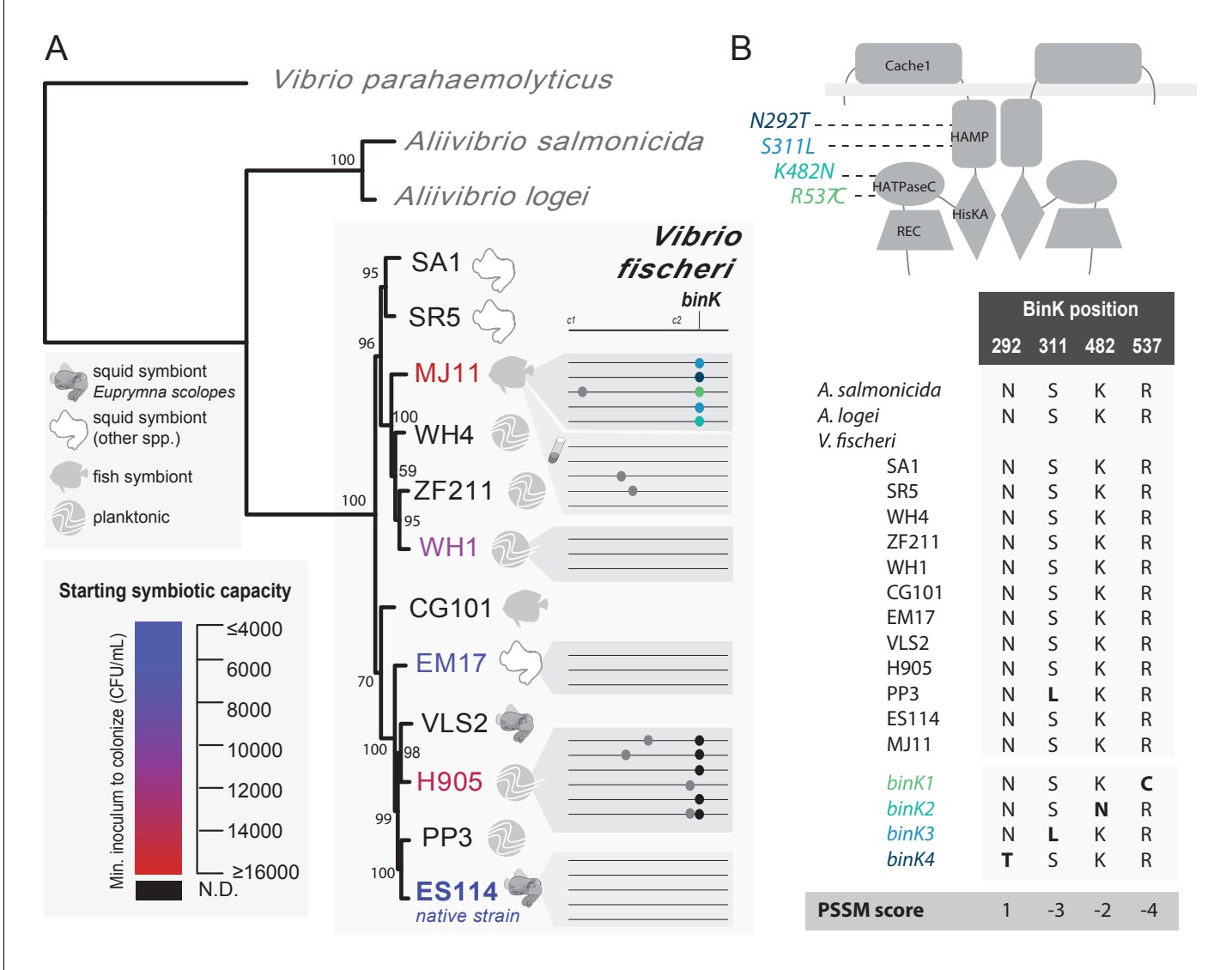

**Figure 2.** Experimental evolution of *Vibrio fischeri* produced multiple alleles in the sensor kinase BinK. (**A**). Phylogenetic relationship, symbiotic capacity, and mutations accrued during squid experimental evolution of ecologically diverse *Vibrio fischeri* strains. Strain relationships were inferred under maximum likelihood using whole genomes with RealPhy (*Bertels et al., 2014*) and with node supports calculated from 1,000 bootstraps. Graphic symbols for ecological niches represent the source of isolation. Intrinsic squid symbiotic capacities of the five experimentally evolved strains, as determined by the minimum inoculum concentration required for successful colonization of 90% of squid with a 3 hr (ES114, EM17, and WH1) or overnight (H905 and MJ11) inoculum, are represented by color spectrum. Consensus genomes for each of the parallel *V. fischeri* populations evolved through *E. scolopes* are shown on the right, with variants indicated by circles. Mutation details are shown in *Table 2*. The mutations that were selected in host-passaged populations improved symbiotic capacity rather than general vigor. (**B**) BinK mutations arising in squid-evolved populations of MJ11 occurred in the HAMP and HATPaseC domains. A homo-dimer structural model for BinK using TMPRed and hybrid histidine kinase domain modelling (*Anantharaman and Aravind, 2000*; *Stewart and Chen, 2010*) predicts that the accessory sensory Cache1 domain localizes to the periplasm whereas the remaining four functional domains (accessory HAMP, and conserved HisKA, HATPaseC, and REC phosphorelay domains) are cytoplasmic (shown as gray band). A position-specific scoring matrix (PSSM) analysis for each of the squid-evolved BinK positions indicates whether a given amino acid is more (positive) or less (negative) likely to be functionally neutral. Scores for the substitutions incurred at these sites are shown in bold. Please refer to *Figure 2—figure supplement 1* for a phylogenetic assessment of BinK orthology across *Aliivibrio* and *V. fischeri* strains.

The following figure supplement is available for figure 2:

**Figure supplement 1.** BinK orthology, conserved domains and squid-adapted *binK* alleles.

**Table 2.** Summary of mutations detected following experimental evolution of *V. fischeri* using Illumina genome resequencing and targeted Sanger sequencing. For culture-evolved populations of *V. fischeri* MJ11, five isolates from each evolved population were combined to generate five metagenomes. For squid-evolved populations of MJ11, EM17, WH1 and H905, individual isolates were sequenced from lineages that ultimately survived 15 host passages. Isolates saved from early evolutionary time-points (host passage 2) are shown along with isolate genomes from the endpoint (host passage 15). Mean read depth and genome coverage for isolates analyzed with WGS are also provided.

| | | | | Detected mutations[‡] | | | | Illumina sequencing statistics | | | |
| | | | | *binK* (VFMJ11_A0397) | | *tadC1* (MJ11_0520); mutation (reads) | All other mutations detectected by WGS gene (locus); mutation (reads) | | | Coverage | |
| Ancestor | Evolved Passage (EP) | Population | Isolate[†] | allele/ mutation | Method (reads)[§] | | | Reads | % Mapped to ancestor | ChI | ChII |
|---|---|---|---|---|---|---|---|---|---|---|---|
| MJ11 | 2 | 1 | 1 | *binK3*/S311L | WGS (35) | – | – | 3753352 | 99.5 | 135.2 | 118 |
| MJ11 | 2 | 1 | 3 | *binK3*/S311L | WGS (32) | – | – | 3717088 | 99.6 | 134.2 | 113.5 |
| MJ11 | 15 | 1 | 4 | *binK3*/S311L | WGS (17) | – | – | 1716144 | 99.5 | 46.8 | 42.5 |
| MJ11 | 2 | 3 | 3 | *binK4*/ N292T | PCR/SS | n.d. | n.d. | | | | |
| MJ11 | 2 | 3 | 4 | *binK4*/ N292T | PCR/SS | n.d. | n.d. | | | | |
| MJ11 | 2 | 3 | 5 | *binK4*/ N292T | PCR/SS | n.d. | n.d. | | | | |
| MJ11 | 2 | 3 | 6 | *binK4*/ N292T | PCR/SS | n.d. | n.d. | | | | |
| MJ11 | 2 | 3 | 7 | *binK4*/ N292T | PCR/SS | n.d. | n.d. | | | | |
| MJ11 | 2 | 3 | 8 | *binK4*/ N292T | PCR/SS | n.d. | n.d. | | | | |
| MJ11 | 15 | 3 | 1 | *binK4*/ N292T | WGS (42) | – | – | 3031149 | 98.9 | 104.3 | 93.5 |
| MJ11 | 15 | 3 | 3 | *binK4*/ N292T | WGS (63) | – | – | 3777714 | 99.4 | 114.6 | 105.2 |
| MJ11 | 15 | 3 | 4 | *binK4*/ N292T | WGS (42) | – | – | 3420212 | 99.5 | 106.4 | 97.1 |
| MJ11 | 15 | 3 | 7 | *binK4*/ N292T | WGS (41) | – | – | 3304891 | 99.5 | 90.3 | 82.5 |
| MJ11 | 15 | 3 | 8 | *binK4*/ N292T | WGS (63) | – | – | 2948743 | 99.6 | 85.5 | 81.2 |
| MJ11 | 2 | 4 | 1 | *binK1*/ R537C | WGS (62) | – | – | 2511256 | 99 | 84 | 78 |
| MJ11 | 2 | 4 | 3 | *binK1*/ R537C | PCR/SS | n.d. | n.d. | | | | |
| MJ11 | 2 | 4 | 4 | *binK1*/ R537C | PCR/SS | n.d. | n.d. | | | | |
| MJ11 | 2 | 4 | 5 | *binK1*/ R537C | PCR/SS | n.d. | n.d. | | | | |
| MJ11 | 2 | 4 | 6 | *binK1*/ R537C | PCR/SS | n.d. | n.d. | | | | |
| MJ11 | 2 | 4 | 7 | *binK1*/ R537C | PCR | n.d. | n.d. | | | | |
| MJ11 | 2 | 4 | 8 | *binK1*/ R537C | PCR | n.d. | n.d. | | | | |
| MJ11 | 2 | 4 | 9 | *binK1*/ R537C | PCR | n.d. | n.d. | | | | |
| MJ11 | 2 | 4 | 10 | *binK1*/ R537C | PCR | n.d. | n.d. | | | | |
| MJ11 | 2 | 4 | 11 | *binK1*/ R537C | PCR | n.d. | n.d. | | | | |

*Table 2 continued on next page*

Sabrina Pankey *et al*. eLife 2017;6:e24414. DOI: 10.7554/eLife.24414

Table 2 continued

| | | | | Detected mutations[‡] | | | All other mutations detected by WGS gene (locus); mutation (reads) | Illumina sequencing statistics | | | |
|---|---|---|---|---|---|---|---|---|---|---|---|
| | | | | binK (VFMJ11_A0397) | | tadC1 (MJ11_0520); mutation (reads) | | | | Coverage | |
| Ancestor | Evolved Passage (EP) | Population | Isolate[†] | allele/ mutation | Method (reads)[§] | | | Reads | % Mapped to ancestor | ChI | ChII |
| MJ11 | 2 | 4 | 12 | binK1/ R537C | PCR | n.d. | n.d. | | | | |
| MJ11 | 2 | 4 | 13 | binK1/ R537C | PCR | n.d. | n.d. | | | | |
| MJ11 | 2 | 4 | 14 | binK1/ R537C | PCR/SS | n.d. | n.d. | | | | |
| MJ11 | 2 | 4 | 15 | binK1/ R537C | PCR/SS | n.d. | n.d. | | | | |
| MJ11 | 2 | 4 | 16 | binK1/ R537C | PCR/SS | n.d. | n.d. | | | | |
| MJ11 | 15 | 4 | 1 | binK1/ R537C | WGS (131) | G198V (85) | – | 4126149 | 99.4 | 117.8 | 106.1 |
| MJ11 | 15 | 4 | 6 | binK1/ R537C | WGS (61) | G198V (55) | – | 2266821 | 99.2 | 60.8 | 52.5 |
| MJ11 | 15 | 4 | 7 | binK1/ R537C | WGS (89) | G198V (93) | – | 3074437 | 99.6 | 92 | 83.6 |
| MJ11 | 15 | 4 | 8 | binK1/ R537C | WGS (47) | G198V (96) | – | 2902977 | 99.5 | 84 | 77.5 |
| MJ11 | 2 | 5 | 2 | binK3/S311L | WGS (26) | – | – | 3771048 | 99.6 | 132.4 | 123.7 |
| MJ11 | 2 | 5 | 3 | binK3/S311L | WGS (46) | – | – | 2595518 | 99.6 | 84.2 | 83.7 |
| MJ11 | 2 | 5 | 4 | binK3/S311L | WGS (20) | – | – | 1785713 | 99.5 | 60.6 | 57.2 |
| MJ11 | 2 | 5 | 5 | binK3/S311L | WGS (62) | – | – | 3641346 | 99.6 | 117.4 | 113.1 |
| MJ11 | 2 | 5 | 6 | binK3/S311L | WGS (81) | – | – | 4128751 | 99.6 | 141.1 | 134.8 |
| MJ11 | 15 | 5 | 2 | binK4/ N292T | WGS (89) | – | – | 4430823 | 99.1 | 152.3 | 138.4 |
| MJ11 | 15 | 5 | 3 | binK3/S311L | WGS (10) | – | – | 3248580 | 99.3 | 88 | 81.1 |
| MJ11 | 15 | 5 | 4 | binK3/S311L | WGS (59) | – | – | 3609382 | 99.5 | 106.8 | 97.1 |
| MJ11 | 15 | 5 | 5 | binK3/S311L | WGS (28) | – | – | 2915570 | 99.5 | 87.4 | 82.6 |
| MJ11 | 2 | 6 | 1 | binK2/ K482N | WGS (104) | – | – | 4748569 | 99.1 | 164.6 | 147 |
| MJ11 | 2 | 6 | 2 | binK2/ K482N | PCR/SS | n.d. | n.d. | | | | |
| MJ11 | 15 | 6 | 1 | binK2/ K482N | WGS (75) | – | – | 2764910 | 99.4 | 83.2 | 75.5 |
| MJ11 | 15 | 6 | 2 | binK2/ K482N | WGS (63) | – | – | 3240968 | 99.2 | 88 | 72.6 |
| MJ11 | 15 | 6 | 3 | binK2/ K482N | WGS (93) | – | – | 3814367 | 99.5 | 108.1 | 101.7 |
| MJ11 | 15 | 6 | 4 | binK2/ K482N | WGS (108) | – | – | 3714638 | 99.5 | 121.4 | 85.7 |
| MJ11 | 15 | 6 | 5 | binK2/ K482N | WGS (90) | – | – | 3006362 | 99.4 | 85.5 | 72 |
| MJ11 | 15 | Culture1 | mg | – | | – | – | 10319291 | 98 | 272.8 | 237.8 |
| MJ11 | 15 | Culture3 | mg | – | | – | – | 7496847 | 98.2 | 196.7 | 195 |
| MJ11 | 15 | Culture4 | mg | – | | – | fliA (VF_1834); G80D (63) | 2894160 | 98.3 | 76.6 | 67.4 |
| MJ11 | 15 | Culture5 | mg | – | | – | fliP (VF_1842); Δ1 @ 476/870nt (110) | 5571439 | 97.9 | 148.5 | 132.1 |

Table 2 continued on next page

Table 2 continued

| | | | | Detected mutations[‡] | | tadC1 | All other mutations | Illumina sequencing statistics | | | Coverage | |
| | | | | binK (VFMJ11_A0397) | | (MJ11_0520); | detected by WGS | | | | | |
| Ancestor | Evolved Passage (EP) | Population | Isolate[†] | allele/ mutation | Method (reads)[§] | mutation (reads) | gene (locus); mutation (reads) | Reads | % Mapped to ancestor | ChI | ChII |
|---|---|---|---|---|---|---|---|---|---|---|---|
| MJ11 | 15 | Culture2 | mg | – | | – | – | 5411032 | 98 | 144.2 | 129.4 |
| WH1 | 15 | 4 | 1 | – | | – | – | 7273244 | 98.6 | 257.8 | 251.1 |
| WH1 | 15 | 4 | 2 | – | | – | – | 2144381 | 99.6 | 61.4 | 65.1 |
| WH1 | 15 | 4 | 3 | – | | – | – | 2260232 | 99.6 | 62.1 | 66.6 |
| WH1 | 15 | 4 | 4 | – | | – | – | 2341428 | 99.7 | 61.6 | 65 |
| WH1 | 15 | 5 | 1 | – | | – | NADH oxidase (VF_A0027); A402T (62) | 1732106 | 99.5 | 60.8 | 64.7 |
| WH1 | 15 | 5 | 2 | – | | – | NADH oxidase (VF_A0027); A402T (61) | 1737095 | 99.4 | 61.9 | 64.9 |
| WH1 | 15 | 5 | 3 | – | | – | NADH oxidase (VF_A0027); A402T (80) | 2194847 | 96 | 60.8 | 63.4 |
| WH1 | 15 | 5 | 4 | – | | – | – | 2191986 | 99.8 | 61.9 | 64.9 |
| WH1 | 15 | 6 | 1 | – | | – | – | 9256547 | 99.3 | 212.6 | 220.3 |
| WH1 | 15 | 6 | 2 | – | | – | – | 2131144 | 99.6 | 62 | 64.7 |
| WH1 | 15 | 6 | 3 | – | | – | – | 1908857 | 99.5 | 62.4 | 60.5 |
| EM17 | 15 | 6 | 2 | – | | – | – | 2611609 | 99.6 | 93.3 | 89.3 |
| EM17 | 15 | 7 | 1 | – | | – | – | 6690137 | 98.6 | 225.8 | 227.1 |
| EM17 | 15 | 7 | 4 | – | | – | – | 2977429 | 99.5 | 83.4 | 82.1 |
| EM17 | 15 | 7 | 5 | – | | – | icmF (VF_0992);S171N, (72) | 2414288 | 99.5 | 71.6 | 71.5 |
| EM17 | 15 | 8 | 1 | – | | – | – | 3177981 | 99.5 | 97.5 | 94.6 |
| EM17 | 15 | 8 | 2 | – | | – | – | 3138175 | 99.5 | 92.4 | 92.3 |
| EM17 | 15 | 8 | 3 | – | | – | – | 2810099 | 99.5 | 81.2 | 80 |
| EM17 | 15 | 8 | 5 | – | | – | – | 5230411 | 99.6 | 144.9 | 143.2 |
| EM17 | 15 | 9 | 1 | – | | – | – | 8022935 | 99.4 | 184.2 | 173.5 |
| EM17 | 15 | 9 | 2 | – | | – | – | 3346216 | 99.6 | 113.7 | 106.9 |
| EM17 | 15 | 9 | 3 | – | | – | gdh2 (VF_1284); E732D (72) | 3484188 | 99.5 | 95.7 | 93.2 |
| EM17 | 15 | 9 | 5 | – | | – | – | 2445758 | 99.5 | 72.8 | 72.6 |
| H905 | 15 | 1 | 1 | (Δ37168 bp/ 25 genes) | WGS (230) | – | IlvY (VF_2529); M25I (233) | 7645508 | 94.2 | 250.4 | 222.1 |
| H905 | 15 | 1 | 2 | (Δ37168 bp/ 25 genes) | WGS (167) | – | IlvY (VF_2529); M25I (112) | 3531114 | 96.8 | 117.5 | 104.4 |
| H905 | 15 | 1 | 3 | (Δ37168 bp/ 25 genes) | WGS (175) | – | IlvY (VF_2529); M25I (97) | 3596689 | 97 | 122.3 | 109.1 |
| H905 | 15 | 2 | 2 | Δ16 bp@ 498/2595 | WGS (75) | – | purR (VF_1572); N71T (60) | 2819387 | 97.6 | 91.4 | 79.6 |
| H905 | 15 | 2 | 4 | Δ16 bp@ 498/2595 | WGS (94) | – | purR (VF_1572); N71T (52) | 2992978 | 96.9 | 103.3 | 91.4 |
| H905 | 15 | 2 | 5 | Δ16 bp@ 498/2595 | WGS (90) | – | purR (VF_1572); N71T (95) | 3844830 | 96.3 | 123.6 | 109 |
| H905 | 2 | 3 | 1 | – | | – | tadF2 (VF_A0228); G21D (68) | 3393611 | 90.7 | 99.5 | 92.2 |
| H905 | 15 | 3 | 1 | – | | – | tadF2 (VF_A0228); G21D (140) | 7974773 | 91.5 | 147.9 | 143.9 |

*Table 2 continued*

| | | | | Detected mutations[‡] | | | Illumina sequencing statistics | | | | |
| | | | | *binK* (VFMJ11_A0397) | | *tadC1* (MJ11_0520); mutation (reads) | All other mutations dettected by WGS gene (locus); mutation (reads) | | | Coverage | |
| Ancestor | Evolved Passage (EP) | Population | Isolate[†] | allele/ mutation | Method (reads)[§] | | | Reads | % Mapped to ancestor | ChI | ChII |
|---|---|---|---|---|---|---|---|---|---|---|---|
| H905 | 15 | 3 | 2 | T195I | WGS (65) | – | *tadF2* (VF_A0228); G21D (28) | 1989875 | 95.5 | 65.4 | 58.2 |
| H905 | 15 | 3 | 3 | – | | – | *tadF2* (VF_A0228); G21D (77) | 3253899 | 96.7 | 103.8 | 94.4 |
| H905 | 15 | 3 | 4 | – | | – | *tadF2* (VF_A0228); G21D (58) | 3242749 | 97.1 | 103.3 | 94.7 |
| H905 | 15 | 3 | 5 | – | | – | *tadF2* (VF_A0228); G21D (25) | 2190771 | 95.9 | 67.5 | 59 |
| H905 | 15 | 4 | 1 | E43* | WGS (102) | – | – | 6651385 | 92 | 125.1 | 130 |
| H905 | 15 | 4 | 3 | E43* | WGS (111) | – | – | 4032373 | 96.4 | 135.9 | 120.4 |
| H905 | 15 | 4 | 4 | E43* | WGS (187) | – | – | 6122168 | 95.8 | 203.4 | 179.4 |
| H905 | 15 | 4 | 5 | E43* | WGS (90) | – | – | 3177817 | 96.7 | 100.8 | 90.6 |
| H905 | 15 | 5 | 1 | Δ1 bp @ 2325/2595nt | WGS (113) | – | – | 7166870 | 90.4 | 134.5 | 130.9 |
| H905 | 15 | 5 | 2 | Δ1 bp @ 2325/2595nt | WGS (94) | – | – | 3703946 | 96.7 | 118.6 | 108.3 |
| H905 | 15 | 5 | 3 | Δ1 bp @ 2325/2595nt | WGS (66) | – | – | 2828102 | 97.4 | 98.6 | 90.4 |
| H905 | 15 | 5 | 4 | Δ1 bp @ 2325/2595nt | WGS (109) | – | – | 4721575 | 97 | 158.9 | 143.8 |
| H905 | 2 | 6 | 1 | T195I | WGS (105) | – | *tadF2* (VF_A0228); G21D (28) | 2743693 | 94 | 83.3 | 73.6 |
| H905 | 15 | 6 | 3 | T195I | WGS (142) | – | *tadF2* (VF_A0228); G21D (49) | 5594771 | 97.5 | 191.7 | 175.3 |
| H905 | 15 | 6 | 4 | T195I | WGS (105) | – | *tadF2* (VF_A0228); G21D (37) | 3361206 | 96 | 115.9 | 101.4 |

[†]Individual characterized strain collection names assigned to isolates were derived from their ancestral lineage (e.g. MJ11), evolved passage (e.g. EP2), the population (e.g. 1), and isolate number (e.g. 1), which in the preceding example would give rise to strain collection name of MJ11EP2-1-1. Isolates in bold served as allelic *binK* representatives for further assays. mg: metagenome sequencing by pooling five isolates from a population.

[‡]The presence of mutations was determined from Illumina short read (100PE) whole genome sequencing (WGS), by allele-specific PCR (PCR), and/or by locus PCR-amplification, followed by Sanger sequencing (SS). '–' indicates that no mutations were identified at this locus by breseq (***Deatherage and Barrick, 2014***) in this isolate using WGS. 'n.d.' indicates that the presence of mutations at this locus was not determined.

[§]The number of reads supporting the mutation call from WGS data is provided. Mutations were called for sites with minimum coverage of 20 mappable reads. Mutations identified by Sanger sequencing (SS) of PCR-generated amplicons were confirmed from alignments of both forward and reverse reads. Coding genes reference *V. fischeri* ES114 locus tags.

inoculum for 10 squid), the *binK1* variant colonized multiple squid (***Figure 4—figure supplement 1***). The estimated selective advantage, based on the ratios of the growth rates (a measure of relative competitiveness) of wild-type bacteria and the *binK1* variant in light-organ populations of co-colonized squid, was independent of initial allele frequencies in the inoculum, consistent with a model of hard selection (***Figure 4B***, ***Figure 4—figure supplement 1***) (***Saccheri and Hanski, 2006***). The estimated selective advantage of the squid-adaptive *binK1* allele continued to increase by more than 60% between 24 and 48 hr in squid (24 hr: 1.1; 48 hr: 1.8) (***Figure 4B***). The competitive advantage conferred by *binK1* therefore extended beyond the initial colonization events (the 'initiation phase' during the first 24 hr) to include the period of competitive re-growth following the daily venting of

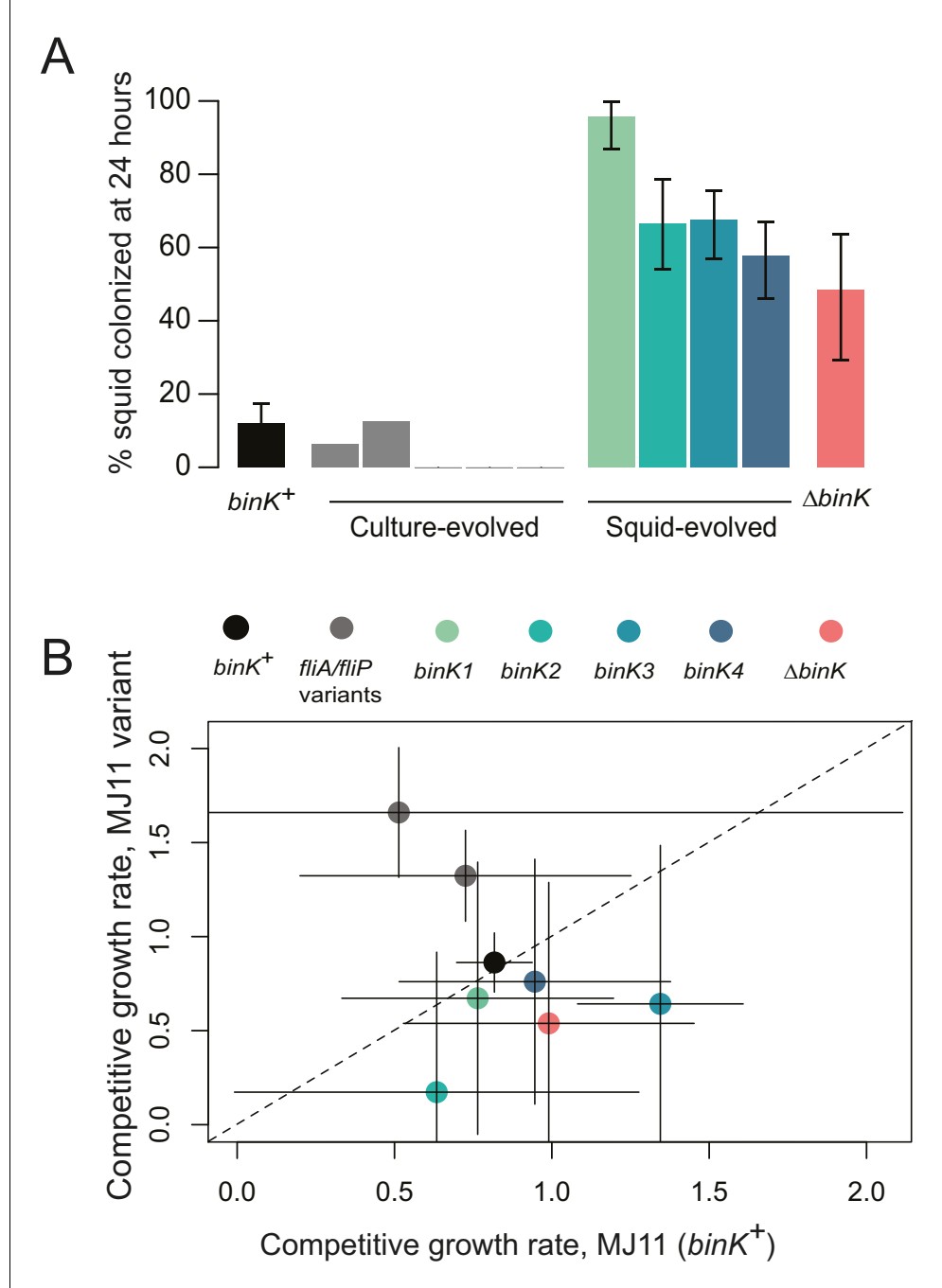

**Figure 3.** Evolved *binK* alleles enhanced host colonization and conferred a fitness tradeoff in non-host environments. (**A**) Symbiotic colonization efficiency of MJ11 and derivatives in squid. Percentage of squid colonized by culture-evolved (c1–c5) and squid-evolved (*binK1- binK4*, bolded isolates in *Table 2*) derivatives of MJ11. Three hours after a cohort of 10–20 squid were inoculated with 3000 CFU/mL of each MJ11 strain, the squid were separated into individual vials, and colonization percentages determined by detectable luminescence at 24 hr. Bars: 95% CI. (**B**) Growth rates of MJ11 and evolved strains during competition in laboratory culture. Average growth rates (realized Malthusian parameters) of *ΔbinK*, squid-evolved *binK* and culture-evolved flagellar mutants (*fliA* and *fliP* variants, see *Table 2*) following in vitro culture competition in minimal media with ancestral *binK*+ MJ11, estimated using CFU yields of each competitor recovered at regular intervals. Bars: 95% CI. The diagonal line indicates 1:1 growth. Please refer to *Figure 3—figure supplement 1* for data on the competitive abilities of *binK1* and *binK3* during colonization. Please refer to *Figure 3—figure supplement 2* for symbiotic yields (CFU) of ES114 and MJ11 strains after 24 and 48 hr.

*Figure 3 continued on next page*

*Figure 3 continued*

The following figure supplements are available for figure 3:

**Figure supplement 1.** Relative competitive ability of *binK1* and *binK3* variants to colonize squid.

**Figure supplement 2.** Growth of strain ES114 and strain MJ11 and its *binK* variants in squid light organs 24 or 48 hr after inoculation.

95% of the bacterial population (the 'maintenance phase'), when several different host sanctions are implicated (*Figure 1A–B*; *Figure 4A–B*). By contrast, squid-adaptive *binK* alleles reduced fitness relative to wild-type (*binK+*) in laboratory planktonic culture ($-0.18 > s > -1$), demonstrating a modest fitness cost for some alleles in the absence of hosts (*Figure 3B*).

Even given the extreme fitness advantage attained by the *binK1* variant growing within squid (*Figure 4B*), the repeated recruitment of *binK* variants among the few cells that initiated symbiosis is remarkable. Not only must the mutations confer exceptional host-selected advantages, but these rare variants must also survive extinction (*i.e.*, loss from the population as the result of genetic drift) during repeated host-imposed bottlenecks (*Nyholm and McFall-Ngai, 2004*; *Wollenberg and Ruby, 2009*). To examine how mutation timing, strength of selective advantage and population size influenced the ability of rare beneficial variants to attain a high frequency in populations passaged between squid, we modeled the evolutionary dynamics and probability of survival of individual variants within a population experiencing recruitment, growth, and repeated cycles of bottlenecking within a single squid over a theoretical range of selection coefficients, applying generalized population and growth parameters derived from native strain ES114 in the squid–*Vibrio* symbiosis (*Wollenberg and Ruby, 2009*; *Altura et al., 2013*; *Wahl and Gerrish, 2001*) (see Materials and methods) (*Figure 4C*). The model predicts that in order for beneficial variation to ensure survival during the extreme bottleneck imposed by the host during initial recruitment, mutants would have to arise early during population expansion and confer $s \sim 6$. Conversely, any beneficial variants arising in light organs during the maintenance of symbiosis, which is characterized by daily venting bottlenecks and re-growth, have increased survival odds even if they confer a lower selective advantage, but the probability of their occurrence is reduced because of the small effective population size (Materials and methods and *Figure 4C*). Thus, the model suggests that the mutants were most probably present in the starting inoculum and were recruited into symbiosis by members of the first squid cohort. Using a high-resolution measure of the *V. fischeri* ES114 genomic mutation rate (*Dillon et al., 2017*), we predict that as many as 185 individual mutations could have spontaneously arisen in *binK* (see Materials and methods) during growth of the inoculum (*Figure 4A*). Despite their low initial frequency, any new alleles that arose by the tenth generation of inoculum growth and ultimately conferred a high selective advantage in squid (i.e., $s > 1$) would be expected to survive the first host passage ~10% of the time (*Figure 4C*, red line). Incidentally, the observed survival of each *binK* allele amounted to 1 or 2 out of 10 experimental squid. Thus, the empirical estimates of the selective advantage conferred by *binK1* in the symbiotic environment are supported by theoretical estimates derived from a model of extraordinarily strong selection during repeated bottlenecks (*Wahl and Gerrish, 2001*).

## Host-adapted *binK* improved early colonization behavior, survival to oxidation and evasion of host immunity through enhanced cell-associated matrix production

The substantial fitness gain conferred by the *binK1* allele within the first 24 hr of colonization (*Figure 4B*) suggested that it enhanced the early colonization behaviors of MJ11 (*Figure 1A and B*) (*Nyholm and McFall-Ngai, 2004*). Syp mediates the aggregation of native strain ES114 in squid mucus and its overproduction enhances the aggregation ability of this same strain (*Brooks and Mandel, 2016*; *Nyholm and McFall-Ngai, 2003*; *Shibata et al., 2012*). Therefore, we evaluated whether aggregation of the squid-evolved *binK1* variant was altered. *binK1* improved aggregation at the entrance to light organs compared to wild-type MJ11 (*Figure 5A*, *Figure 5—figure supplement 1*).

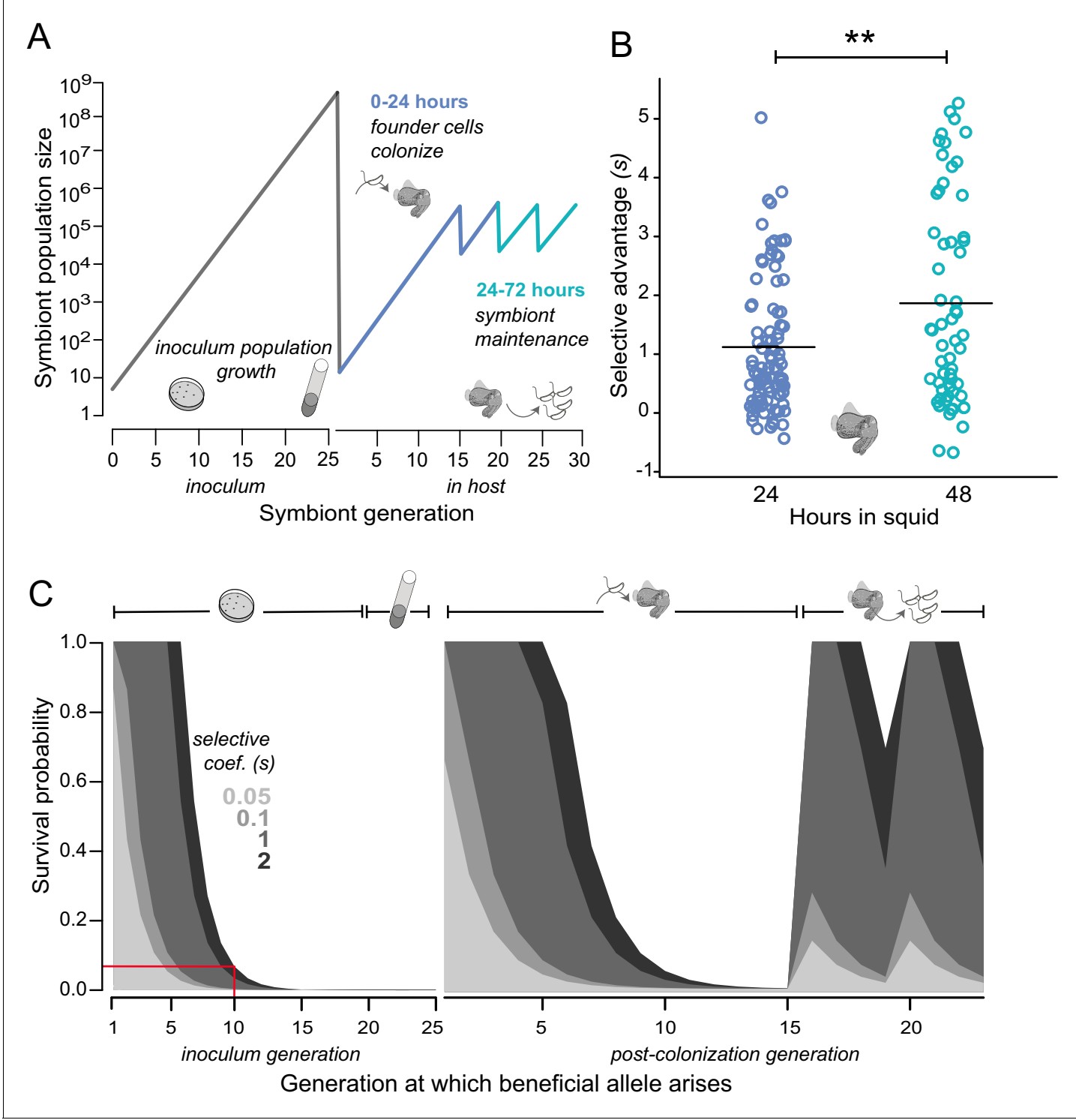

**Figure 4.** Empirical and modeled estimates of selective advantage in evolving *V. fischeri* symbiont populations. (**A**) Conceptual overview of symbiont population dynamics during growth in inoculum and following host colonization (black line), including daily host-imposed bottlenecks. (**B**) Comparison of the selection coefficients conferred by *binK1* in strain MJ11EP2-4-1 (harboring no other mutations) relative to *binK*[+] from co-inoculated squid light organs after 24 or 48 hr. The selective advantage (i.e., relative competitiveness) of the evolved allele increased significantly during this period from 1.1 to 1.8 (Fisher-Pitman permutation test, **p=0.0011). Each circle represents the selective advantage of each strain measured from the strain ratios recovered in an individual hatchling. Please refer to *Figure 4—figure supplement 1* for the effect of starting *binK1* frequencies and inoculum densities on estimates of selective advantage. (**C**) Modeled survival probabilities for new beneficial alleles arising in a growing symbiont population facing host-imposed bottlenecks. The gray shaded curves estimate the survival probability of new mutants following the subsequent population bottleneck, which

*Figure 4 continued on next page*

*Figure 4 continued*

depends on both the generation of growth in the inoculum or host in which they arise (x-axis) and the selective advantage (*s*) conferred by mutation (gray shading). Notably, beneficial variants that arise early in inoculum culture are likely to survive extinction at the subsequent bottleneck, and this probability of survival rapidly decreases even when conferring a large selective coefficient. On the basis of this model, for example, a mutation conferring a large selective advantage (*s* ~2) would have less than a 10% chance of surviving the subsequent colonization bottleneck if it arose during the tenth generation of inoculum growth (red line).

The following figure supplement is available for figure 4:

**Figure supplement 1.** Estimates of the selective advantage of the *binK1* allele during squid colonization across a range of starting frequencies and inoculum densities.

By contrast, it did not cause colony wrinkling (data not shown), a proxy for Syp-mediated biofilm production by strain ES114 (***Brooks and Mandel, 2016***; ***Shibata et al., 2012***). Still, *binK1* dramatically increased in vitro biofilm production compared to MJ11, as determined by surface adherence (***Figure 5B***), perhaps reflecting the presence of more complex biofilm matrices such as cellulose whose expression was enhanced by the *bink1* and Δ*binK* mutations (***Figure 5—figure supplement 2***, Appendix 1) (***Shibata et al., 2012***; ***Darnell et al., 2008***; ***Bassis and Visick, 2010***). To investigate the basis of increased biofilm formation by the *binK1* variant, we overexpressed genes encoding a repressor of Syp, *sypE* (***Morris and Visick, 2013***), and of cellulose, *binA* (***Figure 5—figure supplement 3***) (***Bassis and Visick, 2010***). Each regulator abolished the enhanced biofilm phenotype of the *binK1* variant, indicating that both matrix substrates contributed to this trait (***Figure 5B***). To test the role of Syp directly, we also introduced a Δ*sypK* mutation, which functionally eliminates Syp biofilm production by strain ES114 (***Shibata et al., 2012***). The mutation reduced biofilm by the *binK1* variant, indicating that the variant's improved biofilm production involved Syp production (***Figure 5B***).

Even as the increase in aggregation could confer a fitness gain by *binK* variants during the initiation phase of symbiosis, aggregation is a trait that is variable enough to call into question whether it could explain the dominance of *binK* variants. Improved aggregation alone would not cause the 60% increase in fitness observed during maintenance of the symbiosis (***Figure 4B***, ***Figure 3—figure supplement 2***). Furthermore, to our knowledge, no study has yet evaluated whether biofilm imparts symbiotic fitness beyond aggregation. Because of the potential that biofilm could confer survival in the face of environmental insults, we evaluated whether *binK1* impacted survival upon peroxide exposure, as oxidation is among the host's defensive arsenal (***Small and McFall-Ngai, 1999***; ***Visick and Ruby, 1998***) (***Figure 1A***). The *binK1* and Δ*binK* variants survived oxidation better than MJ11, and overexpression of the Syp repressor *sypE* or the cellulose repressor *binA* decreased survival (***Figure 5C***). Deletion of *sypK* in *binK* variants also reduced survival further, supporting the conclusion that Syp production confers resistance to oxidation (***Figure 5C***). Enhanced biofilm production and survival following peroxide exposure are correlated, suggesting that Syp and cellulose biofilm contribute to oxidative resistance conferred by *binK* variants.

During migration and upon reaching the squid light organ, potential symbionts must contend with host phagocytic, macrophage-like hemocytes which bind, engulf and destroy bacteria (***Figure 1A***) (***Nyholm and McFall-Ngai, 1998***). The ability of squid hemocytes to bind preferentially to non-symbiotic bacterial species is well established, but differential recognition among *V. fischeri* has only been reported for the native strain ES114 and its genetic variants (***Nyholm et al., 2009***). Therefore, we evaluated whether squid hemocytes preferentially target non-symbiotic MJ11, and whether the altered biofilm capacity conferred by *binK1* promoted evasion of the host's innate immune system (***Figure 6***, ***Figure 6—figure supplement 1***). Juvenile squid hemocytes bound wild-type MJ11 to a greater extent than they did the native strain ES114, and this binding was comparable to that observed with other species of bacteria, such as *V. harveyi* (***Figure 6***). In contrast, the *binK1* variant resisted host hemocyte binding at a level that was comparable to squid-native strain ES114 (***Figure 6***). Overexpression of either *sypE* or *binA* reduced immune evasion by ES114, and *sypE* also significantly reduced immune evasion by the squid-adaptive *binK1* variant, demonstrating that production of Syp and cellulose extracellular matrices mediated this trait. These results provide the first experimental evidence that Syp and cellulose production by native and non-

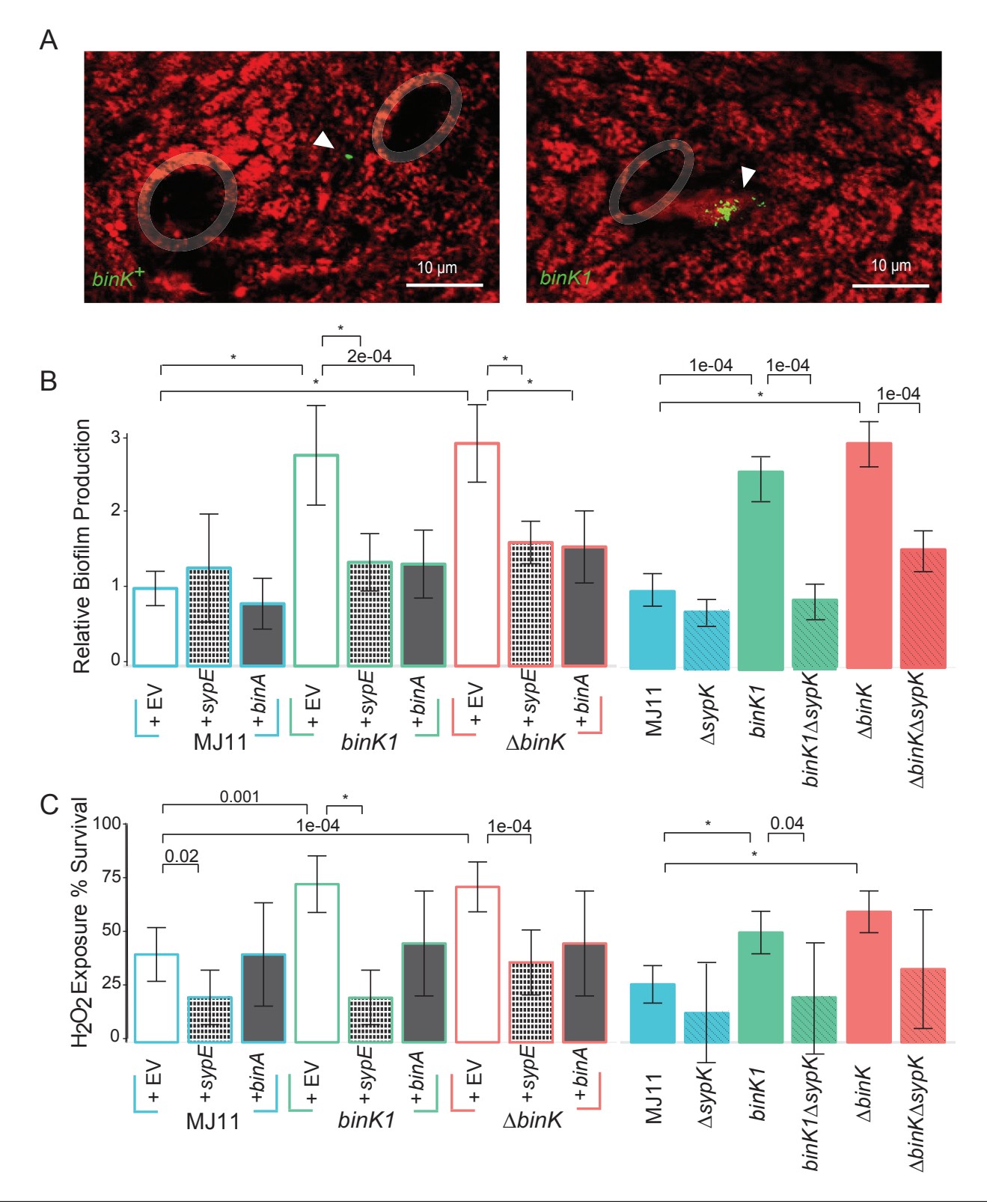

**Figure 5.** Host-adapted *binK1* improved initiation phenotypes through enhanced biofilm. (**A**) *V. fischeri* MJ11 aggregate formation near light-organ ducts. Host tissue stained with CellTracker Orange. Symbionts carry GFP plasmids (pKV111) (***Nyholm et al., 2000***). Micrographs show representative *V. fischeri* aggregates following the dissection of 30 newly hatched animals incubated with each strain. Aggregates were visualized between 2 and 3hr after of inoculation using a Zeiss LSM 510 Meta laser-scanning confocal microscope. Please refer to ***Figure 5—figure supplement 1*** for additional

*Figure 5 continued on next page*

*Figure 5 continued*

views of aggregate formation. (B) Biofilm production (crystal violet staining relative to MJ11) by wild-type MJ11(*binK*+), squid-adaptive *binK1* and Δ*binK* variants in the presence of either empty vector (EV, pVSV105) (white fill), Syp biofilm repressor *sypE* (pCLD48) (hatched fill), or cellulose repressor *binA* (pRF2A3) (gray fill). n = 12–16 biological replicates. See *Figure 5—figure supplement 2* for evidence of increased cellulose in *binK* variants, and *Figure 5—figure supplement 3* for biofilm repressor schematic. Followed by influence of a *sypK* deletion on biofilm production of MJ11 and *binK* variants. n = 10 biological replicates. (C) Binomial mean of survival following exposure to hydrogen peroxide of wild-type MJ11(*binK*+), squid-adaptive *binK1* and Δ*binK* variants in the presence of either empty vector (EV, pVSV105) (white fill), *sypE* (pCLD48) (hatched fill), or *binA* (pRF2A3) (gray fill). n = 20–50 biological replicates. Followed by influence of a *sypK* deletion (diagonal line overlay) on population survival of MJ11 and *binK* variants (color fill). n = 15–106 biological replicates. Error bars 95% CI. Significant p values (p<0.05) are indicated above each comparison. *p<2.2e-16. Although the effects of overexpression of *binA* and deletion of *sypK* on oxidative resistance in the Δ*binK* variant followed the same trends as these genes in *binK1*, the reductions were only marginally significant (p=0.051 and 0.15, respectively). Please refer to *Figure 5—figure supplement 2* for transcriptomic evidence of reduced expression of two cellulose loci in the Δ*binK* mutant. A schematic of the impact of the BinA and SypE repressors on biofilm substrates is available as *Figure 5—figure supplement 3*.

The following figure supplements are available for figure 5:

**Figure supplement 1.** In vivo aggregation behavioral changes conferred by evolved *binK1* variant.

**Figure supplement 2.** Transcriptional shifts associated with *binK* variants.

**Figure supplement 3.** Schematic of regulation by the biofilm repressors SypE and BinA.

native *V. fischeri* strains contribute to host hemocyte response. In addition, these findings demonstrate that, by altering biofilm substrate production, *binK1* could improve the survival of MJ11 during multiple host-imposed selective checkpoints.

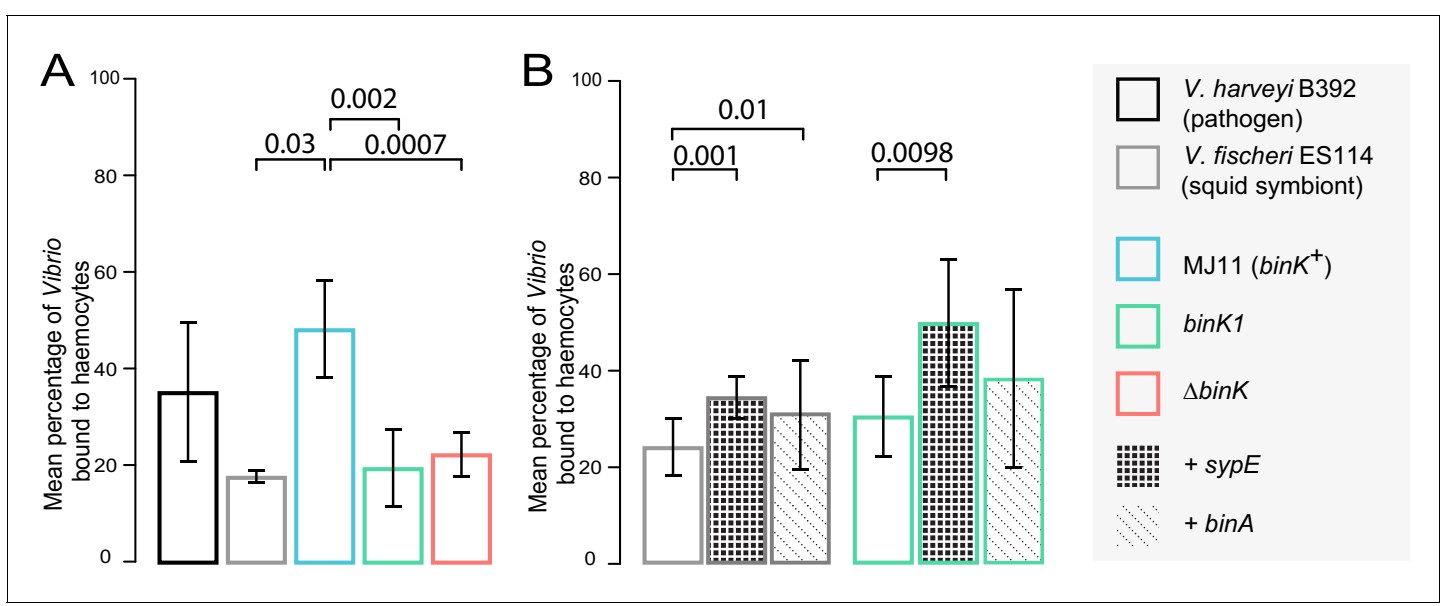

**Figure 6.** Biofilm production by squid-adaptive *binK1* variants mediates hemocyte evasion. (A) Relative efficiency of squid hemocyte binding of GFP-labelled *V. fischeri* strains including: squid-native symbiont ES114, *binK*+ MJ11, Δ*binK* MJ11 (RF1A4), *binK1* MJ11, and shellfish pathogen *V. harveyi* B392. (B) Relative efficiency of squid hemocyte binding of squid-native symbiont ES114 and squid-adapted *bink1* MJ11 carrying the empty vector (pVSV104), *sypE* (pRF2A1) or *binA* (pRF2A4). N = 30–52 hemocytes quantified per strain. Error bars: 95% CI. Significant p-values (p<0.05) are indicated above each comparison. Please refer to *Figure 6—figure supplement 1* for micrographs of *Vibrio*–hemocyte interactions.

The following figure supplement is available for figure 6:

**Figure supplement 1.** In vitro response of squid hemocytes to wild, squid-evolved and mutant *Vibrio*.

## Both Syp and cellulose contributed to enhanced squid colonization efficiency by *binK* variants

Given the demonstrated importance of Syp to colonization initiation (*Shibata et al., 2012*), we predicted that enhanced Syp production by *binK* derivatives improved colonization (*Figure 3A*). Although both Syp and cellulose conferred several phenotypes that are important to the symbiosis (*Figures 5B,C* and *6*), a role for cellulose during colonization processes has yet to be demonstrated. Here, repression of either Syp (through expression of *sypE*) or cellulose (through expression of *binA*) significantly reduced colonization efficiency by MJ11 and its *binK* derivatives (*Figure 7A*). However, *sypE* impaired colonization by Δ*binK* to a greater extent than did *binA*. This suggested to us that Syp may play a greater role than cellulose in colonization, in agreement with the hemocyte evasion results (*Figure 6B*). Alternatively, *sypE* could produce other regulatory effects (*Shibata et al., 2012*; *Bassis and Visick, 2010*; *Ray et al., 2015*; *Miyashiro et al., 2014*). To address the contribution of Syp to improved colonization more directly, we evaluated the impact of a *sypK* deletion, which eliminates colonization by the native symbiont (*Shibata et al., 2012*). Loss of *sypK* had no discernable effect on the colonization of MJ11, presumably because Syp is already under-produced (*Mandel et al., 2009*), but as expected, it significantly reduced colonization by both *binK1* and Δ*binK* variants (*Figure 7B*). Notably, deletion of *sypK* only modestly impaired colonization (25% reduction) by the *binK1* variant, suggesting that Syp is not the only contributor to its enhanced colonization. Elimination of *sypK* had a greater impact on colonization by the Δ*binK* mutant than by the *binK1* variant, reducing its colonization to wild-type levels, which could reflect the greater fitness cost associated with the Δ*binK* allele (*Figure 3A and B*) or might allude to unique functions associated with the evolved *binK1* allele. Together, these results suggest that both Syp and cellulose contribute to enhanced colonization efficiency in the *binK1* and Δ*binK* variants.

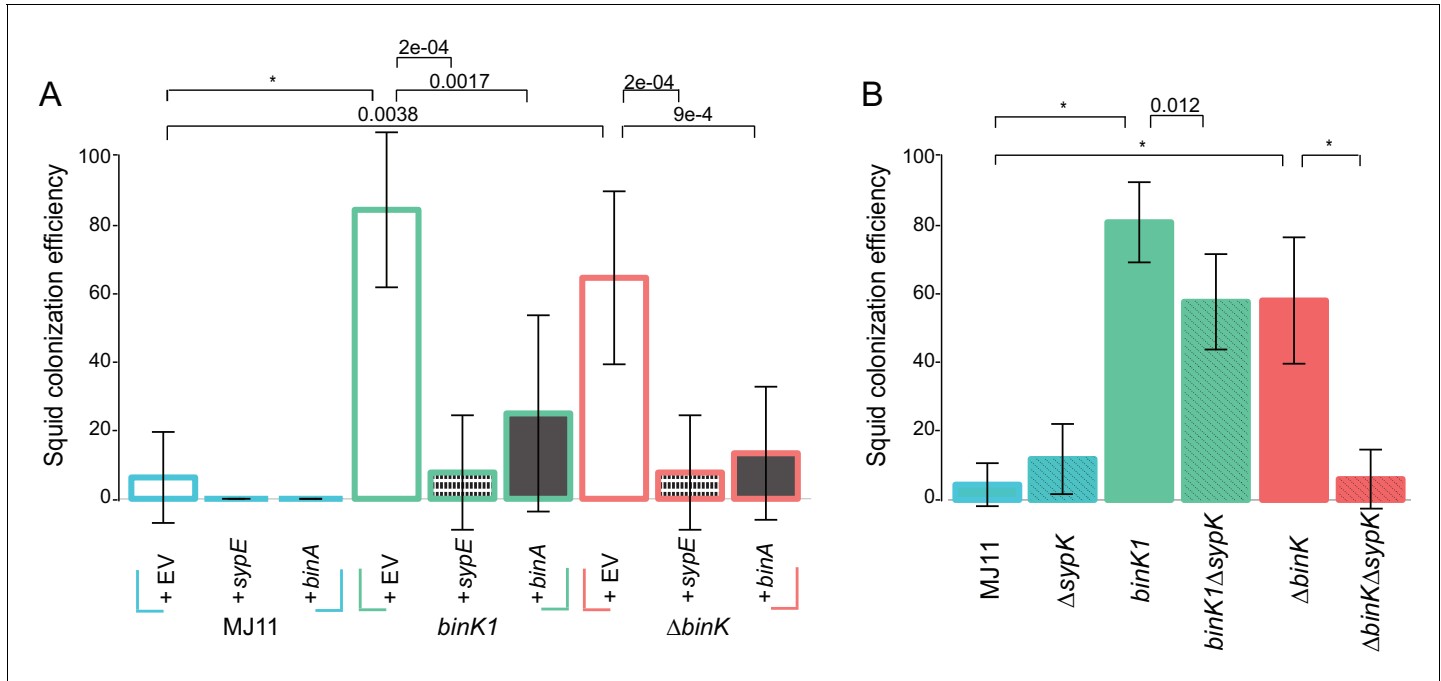

**Figure 7.** Contribution of Syp and cellulose to improved squid colonization by *binK* variants. (**A**) Colonization efficiency (% colonized squid at 24 hr) by wild-type MJ11 (*binK*+), squid-adaptive *binK1* and Δ*binK* variants in the presence of empty vector (EV, pVSV105) (white fill), the Syp repressor *sypE* (pCLD48) (hatched fill), or the cellulose repressor *binA* (pRF2A3) (gray fill). n = 15–20 biological replicates. (**B**) Influence of a *sypK* deletion on colonization efficiency of MJ11 and *binK* variants. n = 31–52 biological replicates. Error bars: 95% CI. Significant p-values (p<0.05) are indicated above each comparison. *p<2.2e-16.

## Squid-adapted *binK* reduced luminescence by attenuating quorum sensing

Bioluminescence serves as the currency of this symbiosis, and yet the correlation of excessive bioluminescence with poor symbiotic ability suggests that luminescence intensity is a phenotype shaped by host selection (*Lee and Ruby, 1994a*; *Nishiguchi et al., 1998*; *Visick et al., 2000*). Squid-adapted derivatives of MJ11 – where the wild-type ancestor is ≥1,000 fold brighter than native symbiont strain ES114 (*Schuster et al., 2010*) – evolved a delay in luminescence induction compared to their ancestors. To determine whether quorum-sensing thresholds had been altered by *binK* mutations, we quantified the production of AinS-synthesized C8-HSL and LuxI-synthesized 3-oxo-C6-HSL signals and the concurrent luminescence production by wild-type MJ11 and by *binK1*, and *ΔbinK* variants during the period of induction ($OD_{600}$ 1.1) (*Figure 8*). For all three strains, luminescence correlated with 3-oxo-C6-HSL concentration (*Figure 8A*) ($r^2$ = 0.857, p=6.4×10$^{-13}$) and not C8-HSL concentration ($r^2$ = 0.105, p=0.1). When compared to the wild-type, both the *binK1* and the *ΔbinK* variant alleles reduced 3-oxo-C6-HSL production and the corresponding luminescence by an order of magnitude (*Figure 8*). These significant differences were not caused by MJ11's attaining a higher cell density (2.0 × 10$^8$ CFU•ml$^{-1}$•$OD_{600}$$^{-1}$), as both the *binK1* and *ΔbinK* derivatives produced slightly higher CFU (*Figure 8B*) (3.2 × 10$^8$ CFU•ml$^{-1}$•$OD_{600}$$^{-1}$ and 3.7 × 10$^8$ CFU•ml$^{-1}$•$OD_{600}$$^{-1}$, respectively) (*Figure 8B*). Although there was a modest (<2 fold) increase in the molar concentration of C8-HSL in *ΔbinK* mutant supernatants, which could inhibit light production through competitive inhibition of LuxR-binding to its cognate 3-oxo-C6-HSL signal (*Kuo et al., 1996*; *Schaefer et al., 1996*), there was no discernable difference in C8-HSL production when controlling for the higher cell counts produced by the *ΔbinK* mutant compared to wild-type MJ11 (p=0.82) (*Figure 8B*). These findings are in agreement with previous biological assays and demonstrate that the *binK1* mutation alters quorum sensing and raises the threshold for quorum-sensing activation of luminescence (*Schuster et al., 2010*).

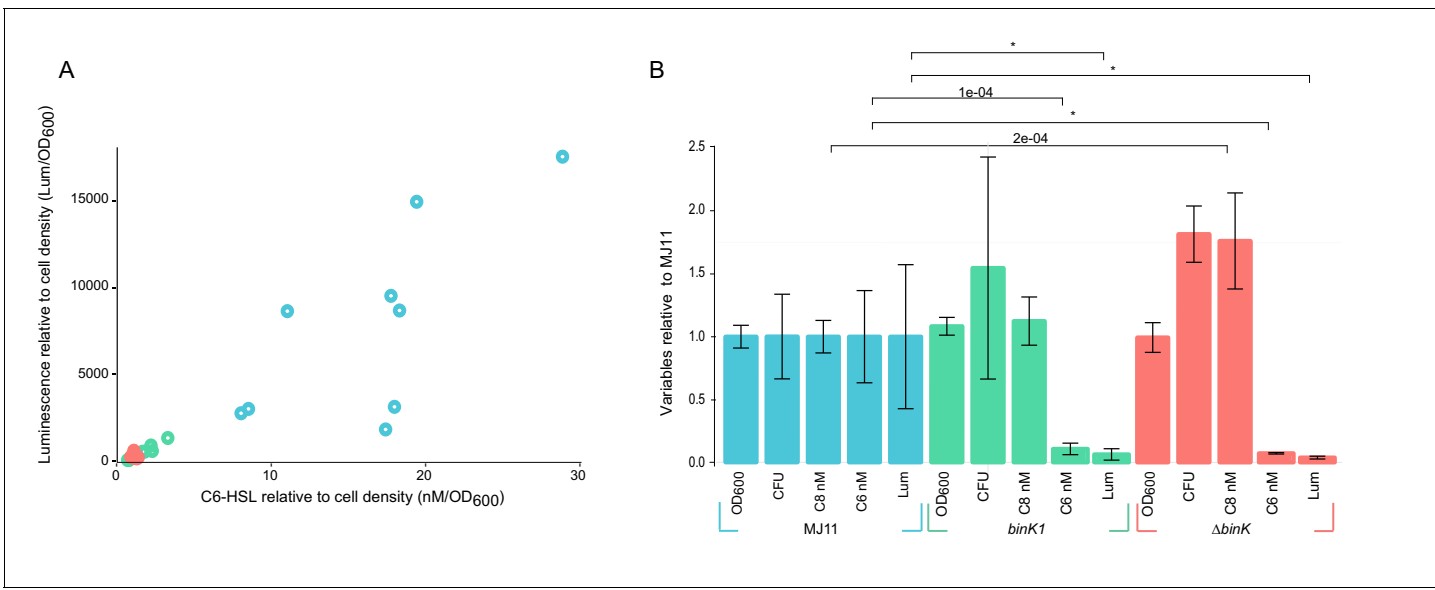

**Figure 8.** Host-adapted *binK1* attenuates quorum-sensing regulation of luminescence. (**A**) Supernatant concentrations (nM/$OD_{600}$) of *N*-(3-oxohexanoyl) homoserine lactone (C6-HSL), as quantified against synthetic standards (*Schaefer et al., 2000*; *Pearson et al., 1994*; *Duerkop et al., 2007*) and corresponding luminescence (Lum/$OD_{600}$) of 10 independent cultures each for wild-type MJ11, *binK1* and *ΔbinK* derivatives during quorum-sensing induction of luminescence determined from cultures grown to early log (Average $OD_{600}$ 1.1, range 0.9–1.4,). (**B**) Average cell density as measured by absorbance ($OD_{600}$), colony-forming units (CFU)/mL/$OD_{600}$, *N*-(3-oxohexanoyl) homoserine lactone (C6) nM concentration, *N*-octanoyl homoserine lactone (C8) nM concentration, and luminescence (Lum)/1 mL culture for ten biological replicates of each variant relative to wild-type MJ11. Error bars: 95% CI. Significant p-values (p<0.05) are indicated above each comparison. *p<2.2e-16.

## Host adaptation produced dominant *binK* alleles suggestive of altered function

Comparisons of the squid-evolved *binK1* variant and *ΔbinK* mutant, especially exemplified by colonization efficiency (*Figure 7B*), transcriptional profiles and changes in metabolic activity that were convergent with the native symbiont (*Figure 5—figure supplement 2*, Appendix 1, Appendix 2), suggested that squid selection did not favor outright loss of BinK function in MJ11. The evolved *binK1* and null *ΔbinK* variants did not differ significantly in biofilm production and exhibited similar biofilm-linked traits of oxidation survival and hemocyte evasion (*Figure 5C and 6*). Yet, the squid-adapted *binK1* variant significantly outperformed the null mutant in culture competition with *binK⁺* (*Figure 3B*). This enhanced fitness could be due to the maintenance of partial function or to regulatory effects that are unique to the evolved allele. To investigate this further, we assessed the impact of multi-copy expression of wild-type and *binK1* alleles. Ancestral *binK⁺* complemented adaptive behaviors conferred by the *binK1* and *ΔbinK* mutants, including the abilities to form biofilm and to colonize squid, as would be expected if wild-type BinK function impaired these traits (*Figure 9*). Multi-copy expression of *binK1* modestly reduced biofilm production by the *ΔbinK* mutant, suggesting that partial function was maintained by this allele, but it also unexpectedly enhanced biofilm production by MJ11, implying altered function (*Figure 5B and 11*). Finally, *binK1* significantly enhanced colonization by all variants, even in the presence of a single genomic copy of the wild-type allele, proiding evidence that *binK1* is dominant and consistent with its altered function. Even if reduced activity of BinK was sufficient to confer some adaptive traits (*Figures 5–8*), these results suggest that improved symbiosis could also arise through phenotypes conferred by alteration of its function (*Figures 9–11*).

## Discussion

In theory, the large population sizes and genetic diversity within bacterial species may enable symbiotic lifestyles with eukaryotic hosts to evolve rapidly (*Fisher, 1930*). While the processes leading to pathogen emergence have been intensely studied, much less is known regarding the genetic changes that drive adaptation to novel host niches in nonpathogenic bacteria (*Jansen et al., 2015*; *Ochman and Moran, 2001*; *Kwong and Moran, 2015*; *Guan et al., 2013*). In pathogens, mobile elements encoded on pathogenicity islands are often cited as the cause of repeated and rapid evolution of host associations, but these elements alone rarely provide bacteria with the ability to colonize hosts (*Reuter et al., 2014*). Further, the selective pressures exerted by new hosts may require synchronized phenotypic changes, limiting the number of adaptive 'solutions' available to a microbial genome that is constrained by regulatory structure. Here, rapid adaptation to squid symbiosis occurred in multiple parallel experimental lineages through convergent mutations in a single gene, the *binK* sensor kinase. These mutations altered multiple functions that are known to contribute to the native symbiosis between strain ES114 and squid (*Figure 10*), suggesting that that the regulatory circuits of *V. fischeri* may have been pre-wired to coordinate diverse symbiotic traits. Many of the BinK-regulated behaviors have established crucial roles in symbiotic association, including quorum-sensing activation of bioluminescence and Syp-mediated aggregation, (*Nishiguchi et al., 1998*; *Brooks and Mandel, 2016*; *Nyholm and McFall-Ngai, 2003*; *Shibata et al., 2012*; *Visick et al., 2000*; *Yip et al., 2005*), but we provide the first experimental evidence that two different *binK*-regulated cell-associated matrix substances, Syp and cellulose, modulate host innate immune interactions that could contribute to strain discrimination during the selection of symbiotic partners.

The convergent paths to adaptation taken by independent lineages evolving experimentally through squid reveals that squid hosts exert hard selection on colonizing bacteria, driving the evolution of fitter, symbiotic genotypes. A model of the population-genetic dynamics of bacterial colonization suggests that in order to survive extinction during the host-imposed bottlenecks, *binK* alleles must confer a massive selective advantage in symbiotic association and must arise early during population growth, most probably— prior to host recruitment—rather than later during symbiotic maintenance (*Figure 4A and C*). This prediction is consistent with the improved initiation capacity of evolved variants (*Figures 1*, *3A*, *7* and *9*) and explains their detection in the first few squid passages (*Table 2*). These mutants would not be expected to rise to detectable frequency considering that alleles that confer enhanced fitness in squid are deleterious in broth culture (*Figure 3B*). The success of *binK* mutations, sweeping from undetectable frequency in the ancestral inoculum to

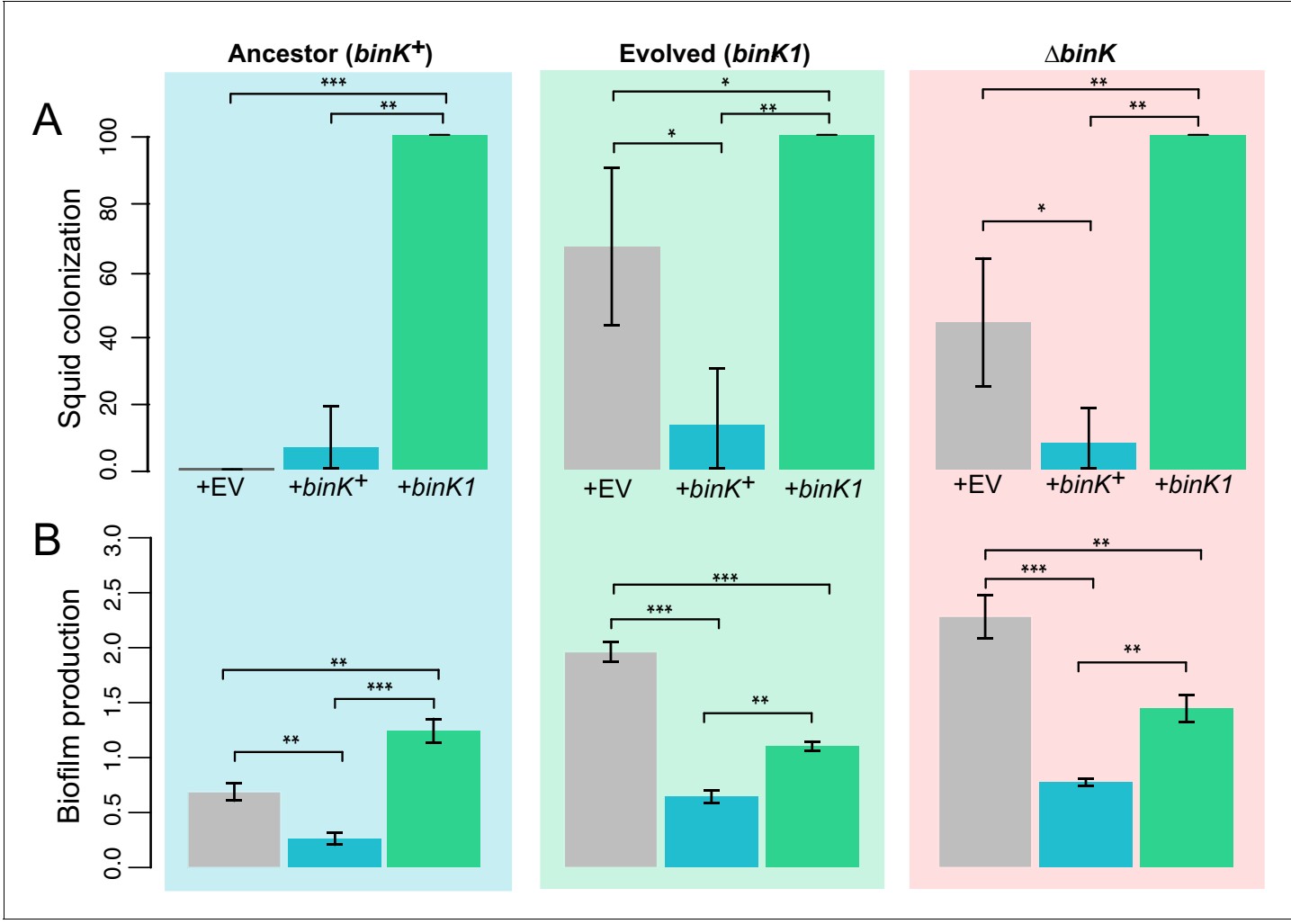

**Figure 9.** Effect of *binK* on squid colonization and biofilm production. (**A**) Improvement in colonization by multi-copy *in trans* expression of the evolved *binK1* allele and decreased colonization by expression of the ancestral *binK+* allele. Colonization assessed by percentage of squid that are luminous after 24 hr. Error bars: 95% CI. N = 15–25. (**B**) Increased biofilm production resulting from *in trans* expression of the *binK1* allele, and decreased biofilm production resulting from expression of the ancestral *binK+*. Comparisons of biofilm production in control-plasmids (pVSV105= EV) with that in multi-copy plasmids carrying *binK* suggest an inhibitory role for BinK in biofilm production, presumably alleviated by the dominance of the *binK1* allele. Biofilm production was quantified by absorbance of crystal violet at $A_{550}$. Background color depicts strain background in which multicopy plasmid effects were measured, mirroring those used throughout where blue is wild-type MJ11, green is the evolved *binK1* variant and salmon is the Δ*binK* derivative. Error bars: 95% CI; non-overlap indicates significance. N = 7–8. Significant p-values (p<0.05) are indicated above each comparison. *p<0.05, **p<0.005, ***p<0.005.

fixation in as little as ~50 generations, was only realized when under strong squid host selection. Estimated selective coefficients for the *binK1* allele of MJ11 ranged as high as *s* = 5.3 when determined empirically, similar to estimates obtained by population modeling (s ~6) (see Materials and methods, *Figure 4*). Selective coefficients above one are rarely reported from nature; however, these are consistent with the stringent selection pressures imposed on pathogens as they colonize new hosts (*Morley et al., 2015*; *Bedhomme et al., 2012*; *Thurman and Barrett, 2016*). This enormous selective advantage is also consistent with the observation that ancestral populations with lower mean fitness (such as strains MJ11 and H905) are more likely than fitter populations (such as WH1, EM17 and ES114) to make a major adaptive leap (*Lenski and Travisano, 1994*). That is, due to their distance from optimal fitness (e.g., 100% colonization), less fit ancestors are poised to benefit more from mutations of greater selective advantage (*Orr, 2000, 2003*; *Wielgoss et al., 2013*). Thus, even though elimination of BinK function also increases competitive fitness by ES114 (*Brooks and*

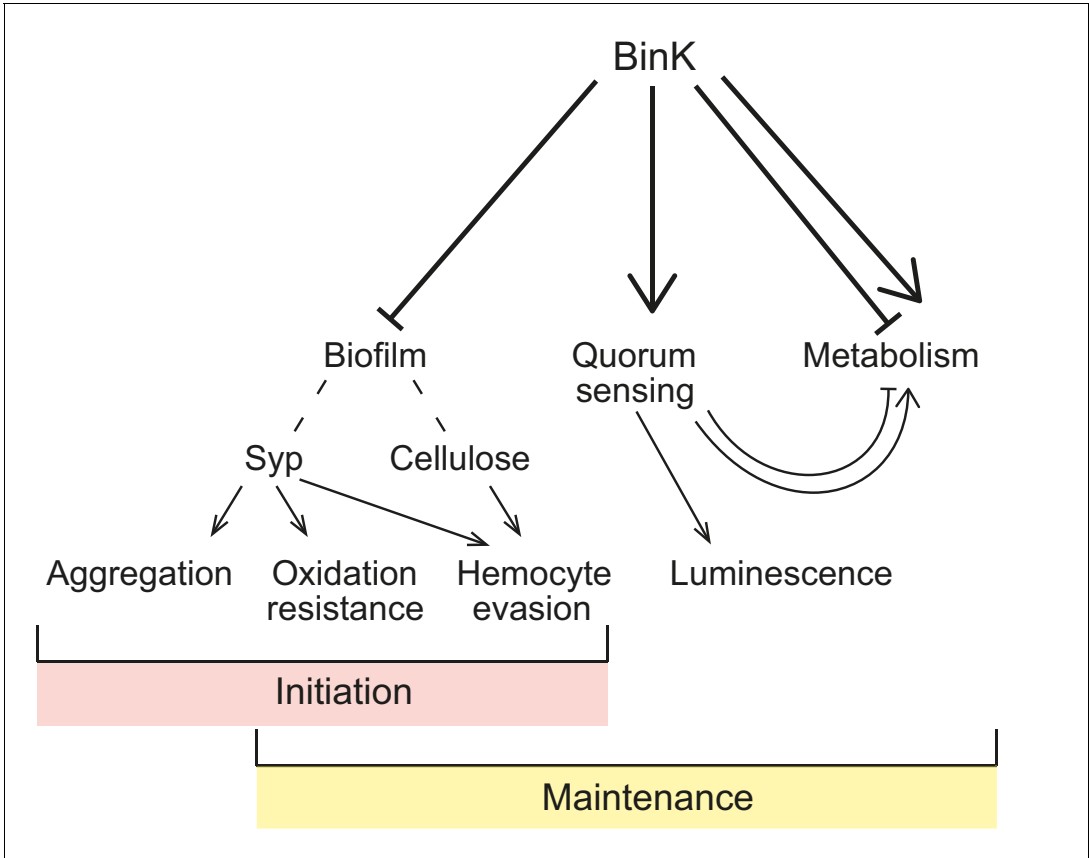

**Figure 10.** Model of BinK regulation of traits adaptive during squid symbiosis. Arrows originating from BinK point to characteristics that are activated or enhanced, and blocked lines point to those that are repressed or blocked by BinK. Hashed lines point to polysaccharides that contribute to biofilm.

*Mandel, 2016*), it is not surprising that *binK* mutations did not evolve in populations derived from ancestors with greater starting fitness, such as EM17, WH1, and ES114, as it is unlikely that these mutations could confer a selective advantage sufficient to survive extinction (*Figure 2A and 4C*, *Table 2*). The high predicted selective advantages of *binK* mutants evolved from squid-maladapted strains MJ11 and H905 support the theory that adaptation from unfit ancestors may initially proceed by large leaps, as opposed to incremental changes of small effect (*Wiser et al., 2013*).

Requisite to successful symbiosis with squid is the ability of bacteria to bypass host barriers during initiation: symbionts first aggregate and then migrate though ducts that are policed by hemocytes and eventually reach the oxidative light organ interior (*Figure 1*) (*Nyholm and McFall-Ngai, 2004*). The poor colonization capacity of MJ11 has been attributed to its lack of *rscS*, a horizontally acquired regulator in the same hybrid histidine kinase family as *binK* (*Figure 2—figure supplement 1*). RscS that activates Syp polysaccharide and allows the native symbiont ES114 to overcome the squid initiation barrier (*Figure 1A*) (*Yip et al., 2006*; *Mandel et al., 2009*). Despite its conserved function as a repressor of Syp in ES114 (*Brooks and Mandel, 2016*), BinK does not impede symbiosis in that strain, perhaps owing to the activity of RscS (*Yip et al., 2006*). But, notably, strain H905—a close relative to ES114 isolated from the squid habitat and containing *rscS*—is symbiotically impaired and also evolved convergent mutations in *binK* during our experiments (*Perry, 2009*) (*Figure 2—figure supplement 1*, *Table 2*). This suggests that its colonization deficiency stems from regulatory constraints on Syp production, from ineffective integration of the horizontally acquired RscS regulator with existing regulatory circuitries, or from the evolution of attributes relating to a planktonic lifestyle which impair its ability to access squid light organs (*Lee and Ruby, 1994a*). Here, MJ11 adapted to experimental squid symbiosis through enhancement

not only of Syp (*Figures 5–7*), a proposed mechanism for symbiotic evolution in the native symbiont ES114 (*Mandel et al., 2009*), but also by producing additional matrix components such as cellulose (*Figure 5—figure supplement 2*, Appendix 1), both of which improved colonization (*Figure 7*).

The finding that biofilm phenotypes conferred by evolved *binK* alleles improved survival of host defenses (*Figure 5 and 6*) expands our understanding of the nature of host selection, and provides important context for how biofilm can confer enhanced fitness upon individuals. Symbiotic microbes commonly secrete exopolysaccharides or glycosylated compounds to produce biofilm capsules that confer protection against macrophages, antibiotics or toxic substances, and that promote adhesion to epithelial surfaces (*Nizet and Esko, 2009*; *Sengupta et al., 2013*; *Williams et al., 2013*; *Hsieh et al., 2003*). Yet beyond its role in aggregate formation, it was not known whether biofilm contributed to squid colonization (*Yip et al., 2006*). The *binK1* allele enabled immune evasion by reducing the attachment of host macrophage-like hemocytes to a level comparable with that of squid-native strain ES114 and by enhancing survival when exposed to oxidation (*Figure 5*). Both immune evasion and biofilm production were suppressed by overexpression of either the *sypE* or *binA* repressors, which indicates that these traits are mediated by Syp and cellulose production (*Figure 5*). Squid immune response is mitigated by *V. fischeri* lipopolysaccharide and other microbe-associated molecular patterns (MAMPs) (*Nyholm et al., 2009*; *Nyholm and McFall-Ngai, 1998*; *Koropatkin et al., 2012*; *Koropatnick et al., 2004*; *Foster et al., 2000*), but this study provides the first evidence that Syp contributes to host immunomodulation by *V. fischeri*. The genes for Syp share little similarity with those encoding the capsular polysaccharide common to immunomodulating *Vibrio* species and other pathogens (*Shibata et al., 2012*; *Yildiz and Visick, 2009*), but the Syp polysaccharide may nonetheless serve a role analogous to that of the polysaccharide ligands of mammalian macrophage receptors produced by gut symbionts, which also exhibit immunosuppressive activity that reduces host inflammatory response (*Mazmanian et al., 2008*; *Chu and Mazmanian, 2013*; *Jones et al., 2014*). Recent evidence in *Vibrio parahaemolyticus* suggests that the use of Syp is potentially widespread among host-associated *Vibrio*, mediating virulence and epithelial colonization (*Ye et al., 2014*) as well as evasion of host innate immunity (*Hsieh et al., 2003*; *Vuong et al., 2004*). The pleiotropic effects of Syp on symbiotic competence suggest why single *binK* mutations provide such benefit to squid-naïve *V. fischeri*. Further, they reveal a critical role for cell-associated polysaccharides in the squid–*Vibrio* interaction, not only mediating group behaviors that improve initiation but also contributing to partner selection on an individual cell basis.

Not only do evolved *binK* alleles increase fitness during the first 24 hr of colonization, they also further enhance fitness between 24 and 48 hr post-colonization during the maintenance phase of symbiosis (*Figure 4B*, *Figure 3—figure supplement 2*) when the squid selects on symbiont luminescence intensity and resource utilization (*Graf and Ruby, 1998*; *Soto et al., 2014*; *Schuster et al., 2010*; *Visick et al., 2000*; *Septer et al., 2013*; *Soto and Nishiguchi, 2014*) (*Figure 1A*). Although luminescence could be directly under host selection (*Figure 8A*) (*Visick et al., 2000*; *Whistler and Ruby, 2003*), selection could alternatively favor the altered quorum threshold underlying reduced luminescence in *binK* variants (*Figure 8B*). Mutations in *luxO* and *litR*—which, like *binK* variants, attenuate quorum sensing—enhance competitive fitness either in culture or in squid (*Fidopiastis et al., 2002*; *Kimbrough and Stabb, 2015*). Impaired quorum sensing by other species also enhances competitive growth because of the subsequent de-repression of metabolic functions such as carbohydrate uptake and utilization, and the perturbation of fatty acid and carbohydrate biosynthesis (*Davenport et al., 2015*; *An et al., 2014*). Transcriptomics analysis indicated that similar changes occurred in the Δ*binK* mutant (*Figure 5—figure supplement 2*; Appendix 1). Quorum-regulated metabolic pathways that serve as 'private goods' could be targets of selection if they facilitated the utilization of host-provisioned resources that support symbiont growth in juvenile squid (Appendices 1 and 2) (*Graf and Ruby, 1998*; *Pan et al., 2015*; *Wier et al., 2010*; *Miyashiro et al., 2011*; *Mandel et al., 2012*; *Dandekar et al., 2012*), thereby contributing to a sustained selective advantage following initial colonization (*Figures 1A* and *4B*).

The synchronized changes attained through amino acid substitutions in an existing sensor kinase highlight how the conserved but malleable components of signal transduction systems make them key mediators of adaptive evolution (*Figure 2C*). During bacterial evolution, sensory transduction pathways may serve as pliable targets because of the modularity of their components (*Vogel et al., 2004*; *Pasek et al., 2006*). Conserved phosphorelay and accessory domains (e.g. *Figure 2C*) are shared across numerous pathways and facilitate flexible partner interactions, known as 'cross-talk'

(*Capra and Laub, 2012*). Sensor histidine kinases are effective targets of selective regimes in part because of their dual kinase and phosphatase capabilities, as well as their ability to augment partner interactions through these shared modules which can allow rapid rewiring of networks (*Capra and Laub, 2012*; *Taylor et al., 2015*; *Rowland and Deeds, 2014*). The array of phenotypes effected in *binK* variants (*Figures 5*, *6* and *8*) implies that BinK sensor kinase may participate in more than one signal transduction pathway (*Nyholm and McFall-Ngai, 2004*; *Yip et al., 2006*; *Miyashiro and Ruby, 2012*). Phenotypic changes could be caused by altered interaction with a number of regulators with phosphorelay modules that are already described both for Syp polysaccharide (*Brooks and Mandel, 2016*) and for the quorum-sensing pathway that controls luminescence (*Miyashiro and Ruby, 2012*; *Whistler et al., 2007*), although this does not eliminate the possibility that there are unidentified partner(s) that mediate these effects. Regardless, evolved BinK enacted global effects by intersecting with pre-existing circuitry, which was presumably shaped by varying interactions with environments including hosts during *V. fischeri* adaptive evolution (*Gao and Stock, 2013*; *Mitrophanov and Groisman, 2008*).

This study demonstrates that some strains of *V. fischeri* can evolve by leaps in host range that result from single mutations of large effect. That simple point mutations in a regulator can evoke such broad consequences reveals that disparate traits that are important for symbiosis initiation and maintenance are already co-regulated. Such preexisting coordination is almost certainly an evolved ability, perhaps reflective of a history of selection and 'tinkering' while fluctuating between the non-host and host-associated environments in which these bacteria naturally reside (*Lee and Gelembiuk, 2008*; *Jacob, 1977*). The immense populations of *Vibrio* species should, in theory, empower natural selection to refine even subtle traits, promoting the ability to adapt to uncertain conditions through appropriate regulation with remarkable efficacy (*Dillon et al., 2017*). Viewed in this light, this study suggests that the exceptional adaptability of certain bacteria such as *Vibrio* in forming novel intimate associations with various host organisms may be possible in part due to the structure of existing regulatory pathways formed during thousands of past transient interactions. Such parsimonious reconciliation of genomic constraints with host selection pressures is likely paramount in shaping emerging symbioses.

## Materials and methods

### Bacterial strains, plasmids, and culture conditions

Strains and plasmids are listed in *Table 1*. Wild-type *Vibrio fischeri* including strain MJ11 (isolated from the fish *Monocentris japonica* [*Haygood et al., 1984*]) and its derivatives, as well as squid symbiont ES114, were routinely grown at 28°C in either liquid seawater-tryptone broth (SWT) or Luria Bertani broth with added salt (LBS) with shaking at 200 rpm, or on LBS medium with 1.5% agar (LBS agar) (*Graf et al., 1994*). *Escherichia coli* strains were routinely grown in Luria-Bertani (LB) broth (*Sambrook et al., 1989*) or in brain heart infusion medium (BHI) (Difco) at 37°C. When required, media were supplemented with antibiotics at the following concentrations: for *V. fischeri*, chloramphenicol (Ch) at 2.5 μg/ml, kanamycin (Km) 100 μg/ml and erythromycin (Em) at 5 μg/ml; for *E. coli*, Ch at 25 μg/ml, Km at 50 μg/ml, and Em at 150 μg/ml (for BHI media). For maintaining selection in seawater, these antibiotics were used at half this concentration. When applicable, agar plates were supplemented with 40 mg of 5-bromo-4-chloro-3-indolyl-β-galactopyranosidase (X-gal)/ml for visualization of β-galactosidase activity. For biofilm quantification, bacteria were grown in liquid seawater-tryptone broth with added salt (SWTO) (*Bose et al., 2007*). To generate transcriptomic libraries, bacteria were grown in 3 mL SWTO supplemented with 0.5 mM N-acetyl-D-glucosamine. Bacteria were also grown in variations of HEPES minimal medium (HMM) (*Ruby and Nealson, 1977*), a seawater-based defined minimal medium with 1x artificial sea water (ASW: 50 mM $MgSO_4$, 10 mM $CaCl_2$, 300 mM NaCl, 10 mM KCl), 0.333 mM $K_2HPO_4$, 18.5 mM $NH_4Cl$, and 0.0144% casamino acids, buffered with 10 mM Hepes with a suitable carbon source. Other buffers were substituted and additional nutrients supplemented as follows: for in vitro competition, the medium was supplemented with 0.53 mM glucose; for siderophore assessment in reduced iron conditions (*Payne, 1994a*), the medium was buffered with 100 mM Pipes (pH 6.8), casamino acids were increased to 0.3%, and the medium was supplemented with 32.6 mM glycerol; and for qualitative detection of siderophores, this medium was additionally supplemented with 1.5% Difco

bacto-agar and 10% chrome azurol S-hexadecyltrimethylammonium bromide assay solution (CAS – HDTMA) (*Lee and Ruby, 1994a*; *Payne, 1994a*; *Boettcher and Ruby, 1990*; *Graf and Ruby, 2000*). Plasmids were conjugated between *E. coli* and *V. fischeri* as previously described (*Stabb and Ruby, 2002*).

## Recombinant DNA techniques and PCR

Integrated DNA Technologies (Coralville, IA) synthesized the oligonucleotide primers listed in *Table 4*. Routine PCR was performed using AccuStart II PCR Supermix (Quanta, Houston, TX). Phusion High Fidelity DNA polymerase (New England Biolabs, Ipswich, MA) was used for cloning and to produce templates for sequencing reactions. PCR cycling was performed according to the manufacturer's protocol in an Eppendorf Mastercycler or Master Cycler Nexus (Eppendorf, Hamburg, Germany). Annealing temperatures used for primers were determined by subtracting 2°C from the melting temperatures (Tm) determined by Premiere Biosoft's Netprimer. The lowest annealing temperature of the primers in the reaction was used during PCR (*Table 4*).

Standard molecular methods and manufacturer protocols were used for transformations, restriction enzyme digests, ligations, gel electrophoresis, and PCR. Restriction enzymes were purchased from New England Biolabs (Beverly, MA), and T4 DNA Ligase was from Invitrogen (Waltham, MA). Gel isolation and extraction of DNA from restriction digests were done using the Qiagen QIAquick Gel Extraction Kit (Qiagen, Valencia, CA). Plasmids for recombinant work and for sequencing were purified using Zymo Research Zyppy Plasmid Mini Prep (Irvine, CA). Genomic DNA used in PCR reactions was isolated by the phenol/chloroform extraction method (*Wilson, 2001*). Routine PCR amplifications were performed with AccuStart II (Quanta Bio, Beverly, MA).

## Genome sequencing and analysis

Genomic DNA was extracted from mid-log cultures grown in LBS using the Promega Wizard Genomic DNA Purification Kit (Madison, WI). The genomes of *V. fischeri* strains EM17, WH1 and H905 were sequenced de novo using single-molecule sequencing (Pacific Biosciences) and assembled using HGAP at the Icahn School of Medicine. Gene models for de novo genomes were predicted and annotated using Prokka with strain ES114 serving as the reference (*Seemann, 2014*). For all strains derived from experimental evolution (both squid and culture experiments), genomic libraries were prepared on isogenic clones following a modified high-throughput Nextera library construction protocol (*Baym et al., 2015*) and were sequenced using the Illumina Hi-Seq 2500 platform at the University of New Hampshire or the New York Genome Center. Nextera PE adapter sequences were removed from raw reads using Trimmomatic (*Bolger et al., 2014*) with the clip settings as follows: ILLUMINACLIP = 2:40:15 LEADING:2 TRAILING:2 MINLEN:25 (*Macmanes, 2014*). Processed reads were aligned and analyzed against their respective strain reference (ancestral) genome to identify mutations, using default settings in breseq (*Deatherage and Barrick, 2014*) for single isolate genomes and using the '—polymorphism' setting for libraries constructed from pooled isolate gDNA. On average, 99% of the processed reads from each isolate mapped to their reference genome, resulting in an average chromosomal coverage of 95x per isolate (*Table 2*) for MJ11. Mutations were called only for regions covered by a minimum of 20 reads. To identify which mutation calls reflected true evolutionary change as opposed to errors in the PacBio or NCBI reference genome, we compared each putative call across all genomes derived from the same ancestor. Potential mutation calls for strain ES114 were cross-referenced with known variants (*Foxall et al., 2015*). Any mutation calls that were shared amongst at least 50% of independently evolved strain genomes were assumed to reflect ancestral genotype and thus discarded. All mutations in the *binK* locus identified by breseq were subsequently confirmed by targeted PCR amplification and Sanger sequencing by using primers A0397 F3 and A0397 R4 for amplification and to sequence *binK1* and *binK2*, and primers A0397 F8 and A0397 R9 for amplification and A0397 F3 and A0397 R6 to sequence *binK3* and *binK4* (UNH and GeneWiz).

## Phylogenetic relationships among *V. fischeri*

Nucleotide sequence from published Vibrionaceae genomes (*Vibrio parahaemolyticus*, *Aliivibrio salmonicida*, *A. logei*, and *V. fischeri* strains ES114, MJ11, SR5, ZF-211; *Table 3*) and newly generated genomes (*V. fischeri* strains H905, EM17, SA1, CG101, VLS2, PP3, WH1, WH4) were analyzed in

REALPHY and RAxML to infer whole-genome maximum likelihood phylogeny under the GTRGAMMA model of nucleotide substitution (*Bertels et al., 2014*). Node support was estimated by running 1,000 bootstrapped analyses.

## Squid colonization and experimental evolution of *V. fischeri*

Squid colonization was conducted as previously described (*Whistler and Ruby, 2003*). Squid were bred from adults collected from Maunalua Bay, HI with the original adults collected and bred in December 2006, and subsequent cohorts collected intermittently from the same location between 2007 and 2016. Squid were routinely held in 32 ppt Instant Ocean (IO) (Blacksburg, VA) in diH$_2$O water. For determining colonization efficiency, a cohort of squid was placed in bacterial inoculum derived from mid-log (OD$_{600}$ 0.2) SWT broth cultures diluted in filtered IO. The luminescence of squid individually housed in 4 mL IO was monitored daily, and bacterial colonization was determined by plating dilutions of homogenized squid following freezing at −80°C. For starting capacity measurements, squid were exposed to inoculum for 3 hr (ES114, EM17, and WH1) or overnight (H905 and MJ11) at increasing concentrations of bacteria (from 3,000 to 20,000 CFU/mL), until 90% of squid became colonized as determined by luminescence detection at 24 and 48 hr post colonization, and direct plating of light-organ homogenates at 48 hr post colonization. Colonization experiments were completed with at least 10 replicate squid, included aposymbiotic control squid, and were repeated a minimum of three times.

Strains MJ11, EM17, WH1, H905, and ES114 were evolved using squid hosts as previously described (*Schuster et al., 2010*). Briefly, 10 aposymbiotic hatchling squid were inoculated in an ancestral population of each strain (20,000 CFU/ml in 50 ml filtered IO for H905 and MJ11, 6,000 CFU/ml for WH1, and 3,000 CFU/ml EM17 and ES114). Following overnight incubation, squid were isolated and rinsed in filtered IO. Squid with detectable luminescence after 48 hr served as the founder passage for each parallel replicate population. At 96 hr following initial inoculation, squid hosts were preserved at −80°C while their seawater containing ventate was used to inoculate a new passage of aposymbiotic squid. Half of the ventate was preserved by freezing in 40% glycerol at −80°C. Serial passaging with 1 ml ventate combined with 1 mL fresh IO was initiated with a hatchling squid held overnight to confirm that they were uncolonized on the basis of luminescence measurements. Passaging continued in this manner for a total of 15 host squid per experimental lineage (see *Figure 1C*).

Isolates from various passages of the evolutions were recovered and stored from archived ventate. Ten microliters of the ventate were plated onto SWT agar and incubated at 28°C, and representative colonies that were phenotypically similar to *V. fischeri* were quadrant streaked for isolation on LBS agar. Isolated colonies were grown in LBS liquid media and preserved by freezing in 40% glycerol at −80°C for subsequent analysis. For isolates whose identity as *V. fischeri* was suspect due to morphological differences, luminescence was measured from SWT cultures, and the strain diagnostic *gapA* gene was amplified and sequenced using primers gapA F1 and gapA R1 (*Table 4*) for confirmation (*Nishiguchi et al., 1998*).

## BinK orthology and hybrid histidine kinase phylogeny

To construct a gene tree for hybrid histidine kinase genes across *V. fischeri* strains and *Vibrio* relatives, each of the gene models from the complete genomes listed in *Table 4* were queried with the PFAM Hidden Markov Models for HATPase C (PF02518), HisKA (PF00512), and REC (PF00072) domains using hmmer. Sequences containing all of these conserved domains were then aligned in MAFFT (*Katoh et al., 2002*). A maximum likelihood topology was inferred using RAxML (*Stamatakis, 2006*) under the PROTGAMMAWAG model of amino acid substitution, following model selection using the Bayesian Information Criterion with IQ-TREE (*Nguyen et al., 2015*). Gene families were annotated based on consensus among strain ES114, *Vibrio parahaemolyticus*, and *E. coli* annotations identified using the BLAST algorithm (*Camacho et al., 2009*).

## Allele identification

Isolates from the second squid ventate from replicate MJ11 population four were screened for *binK* and *binK1* alleles using forward primer A0397 F5* and allele-specific reverse primers A0397 WT+ R and A0397 4+ R for *binK* and *binK1*, respectively (*Table 4*). The presence or absence of amplicons

**Table 3.** Genomes used in phylogenetic analyses. This table lists GenBank accessions for nucleotide genomes used in strain phylogeny and source for gene models used in hybrid histidine kinase phylogeny.

| Strain | NCBI accession/de novo | Prokka/NCBI gene models |
|---|---|---|
| *Escherichia coli* | NC_000913 | NCBI |
| *Aliivibrio wodanis* | LN554846-51 | NCBI |
| *A. salmonicida* | NC_011311–6 | NCBI |
| *A. logei* | NZ_AJYJ00000000 | Prokka |
| *Vibrio furnissii* | NC_016602, NC_016628 | NCBI |
| *Vibrio parahaemolyticus* | NC_004603–5 | NCBI |
| *Vibrio fischeri SR5* | NZ_AHIH00000000 | Prokka |
| *Vibrio fischeri ES114* | NC_006840–2 | NCBI |
| *Vibrio fischeri MJ11* | NC_011184–6 | NCBI |
| *Vibrio fischeri EM17* | De novo | Prokka |
| *Vibrio fischeri WH1* | De novo | Prokka |
| *Vibrio fischeri ZF211* | AJYI01 | Prokka |
| *Vibrio fischeri WH4* | De novo | Prokka |
| *Vibrio fischeri SA1* | De novo | Prokka |
| *Vibrio fischeri CG101* | De novo | Prokka |
| *Vibrio fischeri H905* | De novo | Prokka |
| *Vibrio fischeri PP3* | De novo | Prokka |
| *Vibrio fischeri VLS2* | De novo | Prokka |

was evaluated against controls including MJ11 (*binK*[+]), *binK1* variant MJ11EP2-4-1 and Δ*binK* variant RF1A4. PCR amplification was conducted following denaturation at 95°C for 30 s followed by annealing at 53°C for 15 s, and elongation at 72°C for 50 s. To confirm the identity of alleles, the *binK* region in five isolates was amplified by PCR using A0397 F10 and A0397 R13, and unconsumed dNTPs and primers were removed using ExoSAP-IT (Affymetrix Santa Clara, CA) before Sanger-sequencing at Genewiz (Cambridge, MA) using primers A0397 F3 and A0397 R4 (*Table 4*). Results were aligned with reference MJ11_A0397 using Lasergene Software programs (DNASTAR, Inc. Madison, WI) and the presence of *binK1* in the evolved isolates was confirmed.

## Δ*binK* mutant generation

The MJ11 Δ*binK*::Em[R] (RF1A4) strain was generated by marker exchange mutagenesis using a construct produced by Splicing and Overlap Extension PCR (*Horton et al., 1990*). Briefly, the primer pairs HKSoeA F (SalI) and HKSoeA2 R, HKSoeB2 F and HKSoeB2 R, and HKSoeC2 F and HKSoeC R (KpnI), and the Phusion High Fidelity DNA polymerase were used to amplify the genomic region upstream and downstream of *binK* from MJ11 genomic DNA, using Em[R]colonies and pEVS170 plasmid DNA as the templates (*Tables 1* and *4*) (*Lyell et al., 2008*). The purified amplicons were then fused using Expand Long Template polymerase (Roche) where *binK* was replaced by an Em[R] cassette. This purified product was cloned into pCR2.1 TOPO and transformed into TOP10 cells (Invitrogen, Waltham, MA), following the manufacturer's protocol. Putative clones were sequenced by the Sanger method with primers M13 F, M13 R, TnErm4, and TnErm5 (*Table 4*) at the Hubbard Center for Genome Studies at the University of New Hampshire before the fragment was sub cloned into the suicide vector pEVS79, which was used for allelic exchange (*Stabb and Ruby, 2002*). Whole genome re-sequencing (illumina HiSeq) confirmed that the gene was replaced in MJ11 mutant RF1A4.

**Table 4.** DNA oligonucleotide primers used in this study.

| Primer name | Primer DNA sequence (5′−3′) | Annealing temperature | Source |
| --- | --- | --- | --- |
| A0397 F5 | AAGAGTCATGGTATACATCGG | 51°C | This study |
| A0397 F5* | TGTAGCTGATGAGACTTTGCG | 56°C | This study |
| A0397 F8 | TCATTGAAAGGTTTAATCGGTGT | 57°C | This study |
| A0397 R11 | CACTTTATGGATGATCTTCGCT | 56°C | This study |
| A0397 F3 | GCTGATGAGACTTTCGCTC | 52°C | This study |
| A0397 R4 | GGCTGATTAGATCATCCTGC | 54°C | This study |
| A0397 F12 | CAGAAGCACTAAATCATGTGAG | 52°C | This study |
| A0397 R9 | TCTGACATGCCAATAATGCCAT | 59°C | This study |
| MJ11A0397 R KpnI | GGTACCCCGAAATTAACGACCAT | 50°C | This study |
| MJ11A0397 F SalI | GTCGACAAATAGAAACACTAACCAC | 50°C | This study |
| HKSoeA F (SalI) | GTCGACAATGTAGAAGTGGTAGAACGC | 50°C | This study |
| HKSoeA2 R | GTTTCCGCCATTCTTTGTGGTTAGTGTTTCT3 | 50°C | This study |
| HKSoeB2 F | AGAAACACTAACCACAAAGAATGGCGGAAAC | 50°C | This study |
| HKSoeB2 R | GCACCGACACTCATCAATTCGATATCAAGCT | 50°C | This study |
| HKSoeC2 F | AGCTTGATATCGAATTGATGAGTGTCGGTGC | 50°C | This study |
| HKSoeC R (KpnI) | GGTACCAGCGGCAATAGAATCAGTC | 50°C | This study |
| TnErm4 | AATGCCCTTTACCTGTTCC | 53°C | This study |
| TnErm5 | CATGCGTCTGACATCTATCTGA | 55°C | This study |
| A0397 R13 | GTACACCCGAAATTAACGACCA | 59°C | This study |
| A0397 F10 | CAGAGTTATGGGGTTGCTGAGT | 58°C | This study |
| A0397 WT+ R | GTCCCACCAAATTGACG | 53°C | This study |
| A0397 4+ R | GTCCCACCAAATTGACA | 53°C | This study |
| sypE RF F2 | GCAGGTTATGTGCGAGG | 52°C | This study |
| gapA F1 | GCCGTAGTGTACTTCGAGCG | 55°C | 31 |
| gapA R1 | CCCATTACTCACCCTTGTTTG | 55°C | 31 |
| PrRF9 | AAGCTTATTGGGAATACGGATACCTG | 53°C | This study |
| PrRF10 | CATATGCACATCTTCTAACCATTGCTG | 53°C | This study |
| PrRF19 | TGTCAGTATCACTCCCCTTCAC | 55°C | This study |
| PrRF20 | AGCAGACAGTTTTATTGTTCATTGTTTCACCTCATTTAA | 50°C | This study |
| PrRF21 | TTAAATGAGGTGAAACAATGAACAATAAAACTGTCTGCT | 50°C | This study |
| PrRF22 | TTTCCTGTTTGTTCTTTTTTAGAAAAACTCATCGAGCA | 50°C | This study |
| PrRF23 | TGCTCGATGAGTTTTTCTAAAAAAGAACAAACAGGAAA | 50°C | This study |
| PrRF24 | GTTCCTTCTACAAGTCCTATTCC | 53°C | This study |
| PrRF36 | ATCCATTGTAATAGTGCTGC | 53°C | This study |
| PrRF52 | AATAAGTCCATTTCGTTCTGC | 54°C | This study |
| PrRF53 | AAGCGGAAGTAGCGAAAAC | 54°C | This study |
| VSV105InF | GCCTGGGGTGCCTAATG | 56°C | This study |
| KanINF | ATACAAGGGGTGTTATGAGCC | 55°C | This study |
| KanINR | CAAGTCAGCGTAATGCTCTGC | 56°C | This study |

## ΔsypK mutant generation

The Δ*sypK::aphA1* mutant strains RF1A5, RF1A6, and RF1A7 were generated by marker exchange mutagenesis using a construct produced by Splicing and Overlap Extension PCR (*Horton et al., 1990*). Briefly, the primer pairs PrRF19 and PrRF20, PrRF21 and PrRF22, and PrRF23 and PrRF24, and the Phusion High Fidelity DNA polymerase were used to amplify the genomic region upstream and downstream of *sypK* from MJ11 genomic DNA, and using $Km^R$ colonies and pVSV103 plasmid DNA as the template (*Tables 1* and *4*) (*Dunn et al., 2006*). The purified amplicons were then fused using Expand Long Template polymerase (Roche) where *sypK* was replaced by a $Km^R$ cassette. This purified product was cloned into pCR2.1 TOPO and transformed into TOP10 cells (Invitrogen, Waltham, MA), following the manufacturer's protocol. Putative clones were sequenced by the Sanger method with primers M13 F, M13 R, KanINF, KanINR (*Table 4*) at Genewiz in South Plainfield, NJ before the construct, RF2B7, was used for allelic exchange with a modified chitin competence protocol (*Brooks et al., 2015*). Briefly, *V. fischeri* cells were grown in minimal media with a chitin derivative (n-acetyl glucosamine) until they reached $OD_{600}$ 0.2. Cultures were incubated with 10 µg/mL of pRF2B7 linearized by up to five cycles of freeze-thawing. After incubation with DNA fragments for allelic exchange, cells were recovered, plated onto LBS+Km plates and screened by PCR for incorporation of Δ*sypK::aphA1* fragment using primers PrRF36 and KanINR2 (*Table 4*).

## Transcriptome sequencing and analysis

Single colonies of *V. fischeri* MJ11 and two of its derived strains, squid-evolved *binK1* strain (MJ11EP2-4-1) and MJ11 mutant Δ*binK* (RF1A4), were grown in quadruplicate until they had an $OD_{600}$ of 0.25 (Biophotometer; Eppendorf AG, Hamburg, Germany) in order to capture populations prior to detectable biofilm activity or flocculation and to minimize effects of spontaneous suppression due to growth defects of *binK* variants. Cells were pelleted and flash frozen. RNA was extracted following the protocol for the Quick-RNA MiniPrep kit (Zymo, Irvine, CA). Ribosomal RNA was depleted using the RiboZero kit (Illumina). mRNA libraries were constructed using the TruSeq Stranded mRNA library prep kit (Illumina) and sequenced using the HiSeq 2500 at New York Genome Center. Quality-trimmed reads were mapped onto the MJ11 reference genome using bowtie2 (*Langmead and Salzberg, 2012*) and quantified using RSEM (*Li and Dewey, 2011*). Differential expression between strains was assessed using edgeR (*Robinson et al., 2010*) with a significance threshold of FDR < 0.05.

## Plasmid construction

*binK* and *binK1* alleles were cloned into pVSV105 (*Dunn et al., 2006*) following amplification of MJ11 and *binK1* genomic DNA with forward primer MJ11A0397 F SalI and reverse MJ11A0397 R KpnI (*Table 4*). The 2.977 Kb product was cloned into pCR2.1 TOPO (Invitrogen) following the manufacturers' instructions. The constructs were sequenced using M13F, M13R, A0397 F3, A0397 F5, A0397 F8, A0397 F12, A0397 R4, A0397 R9, and A0397 R11 (*Table 4*), and aligned to their respective references to ensure that there were no mutations. The inserts were sub cloned from pCR2.1 TOPO into pVSV105 following digestion using the restriction enzymes SalI and KpnI, and ligation using T4 DNA ligase. Ligation reactions were transformed into chemically competent DH5αλ*pir* cells (*Herrero et al., 1990*). Cell lysates of $Ch^R$ colonies were directly screened for correct insert harboring plasmids by PCR using M13F and A0397 R4. Positive clones harbored pRAD2E1(*binK*$^+$) and pRF2A2(*binK1*).

binA was cloned into pVSV105 (*Dunn et al., 2006*) following amplification of MJ11 genomic DNA with forward primer PrRF9 and reverse PrRF10 (*Table 4*). The 2.053 Kb product was cloned into pCR2.1 TOPO (Invitrogen) following the manufacturers' instructions. The TOPO constructs were sequenced using M13F, M13R, PrRF9, PrRF10, PrRF52 and PrRF53 (*Table 4*), and aligned to the genomic sequence in MJ11 using the DNA Star software package (https://www.dnastar.com/) to ensure that no mutations were generated during cloning. The inserts were sub-cloned following digestions with XhoI and NdeI and SalI and NdeI digestions of pVSV105, and ligation using T4 DNA ligase. Ligation reactions were transformed into chemically competent DH5αλ*pir* cells. Cell lysates of $Ch^R$ were directly PCR screened for insert-harboring plasmids by PrRF9 and VSV105InF (*Table 4*). Positive clones harbored pRF2A3 (*binA*$^+$) (*Table 1*).

To make Km$^R$ constructs compatible with pKV111 for hemocyte assays, the *sypE* SphI and SacI fragment was sub-cloned from pCLD48 into SphI and SacI digested pVSV104 (*Stabb and Ruby, 2002*). Following transformation into chemically competent DH5α*λpir* cells, the cell lysates of Km$^R$ colonies were directly screened for *sypE* insert using M13F and sypE RF F2 (*Table 4*). Positive clones harbored pRF2A1 (*Table 1*). The *binA* Sph1 and SacI fragment was sub-cloned from TOPO 2.1 into pVSV104 digested with SphI and SacI (*Stabb and Ruby, 2002*). Cell lysates of Km$^R$ colonies were directly screened for *binA* insert using VSV105InF and PrRF9 (*Table 4*). Positive clones harbored pRF2A4 (*Table 1*).

To mark bacteria for direct competition, the *lacZ*-expressing plasmid pVSV103 (*Dunn et al., 2006*), which confers a blue colony on media containing X-gal and confers kanamycin resistance, was used along with a derivative of this plasmid (pCAW7B1) in which *lacZ* was inactivated by removal of an internal 624-bp fragment by digestion with HpaI followed by self-ligation.

## Bacterial competition *in vivo*

Estimates of Malthusian growth rates and fitness for MJ11 strains were calculated by measuring relative abundances of marked strains in squid hatchings that were co-inoculated with varying ratios of each strain (Altered Starting Ratio method *sensu* [*Wiser and Lenski, 2015*]). Strains were marked with either an intact version of the plasmid pVSV103 (*Dunn et al., 2006*) or pCAW7B1 that contains *lacZ*, which harbors a 200-amino-acid deletion that renders LacZ unable to produce blue pigment in colonies (*Table 1*). Squid were inoculated overnight in 50 ml IO containing 25 μg/ml Km and stored at −80°C after 24 or 48 hr (n = 98 and 59, respectively) following initial inoculum exposure if detectably luminous. Inoculations spanned 17 experiments, which contained inoculums with reciprocally marked strains in order to control for potential plasmid effects, ranging both in total cell density (from 1,600 to 26,600 CFU/mL) and in relative strain frequency (from ~1 binK1 per 10,000 binK+ up to approximately equal proportions). To estimate CFU abundance for each strain in squid light organs, we counted blue and white colonies after 72 hr of plating squid homogenates onto SWT plates containing 50 μg/ml Km and 1.5 mg/ml X-gal.

To calculate the selective coefficient (*s*) associated with the evolved variant during competition with the ancestral genotype in squid, we use the derivation in *Chevin (2011)*. First, Malthusian growth rates (*M*) (*Fisher, 1930*) were estimated by taking the natural-log of the ratio of the CFU estimate from each co-colonized light organ to the starting inoculum concentration (i.e., starting density) (*Lenski and Travisano, 1994*; *Lenski et al., 1991*). The standard plating method to quantify symbionts from squid light organs can detect as few as 15 CFU (*Ruby and Asato, 1993*). Then the relative growth rate difference (*s$_{GR}$*) was used to calculate the selection coefficient:

Relative growth rate difference, $s_{GR} = (M_{Evo} - M_{Anc})/ M_{Anc}$

Selection coefficient, $s = s_{GR} / ln2$

Spearman rank correlation tests were then used to test for relationships between Malthusian growth rates and either starting frequency or starting density of inocula. Significant differences in growth rate at either 24 or 48 hr between ancestral and evolved binK1 strains were assessed using exact Fisher-Pitman permutation tests through the 'oneway_test' method in the R 'coin' package (*Hothorn et al., 2008*). Significant differences in competitive colonization by evolved variants *binK1* and *binK3* (mutations in HATPaseC or HAMP domains, respectively) were assessed with a permutation t-test in the R package 'DAAG' using the method 'onet.permutation' with 9,999 simulations (*Maindonald and Braun, 2015*).

## Bacterial competition *in vitro*

Malthusian growth rates were estimated similarly to in vivo competitions in which fitness for MJ11 strains was determined following co-inoculation of 150 μl with a single colony from each strain marked with either pVSV103 (*Dunn et al., 2006*) or pCAW7B1. Cultures were grown statically at 28°C and, at 2 hr intervals, a new culture was founded by serial 1/10 dilution into fresh media in a 96-well polystyrene microplate (Corning). At each passage, 20 μl of each competition was diluted, and plated onto SWT plates containing 50 μg/ml Km and 1.5 mg/ml X-gal. The total number of blue and white colonies apparent after 72 hr of growth was determined and used for calculations of realized Malthusian parameters. Strain competitions were each conducted with eight replicates and repeated twice. Differences in growth rate (Malthusian parameter, described above and in

*Fisher (1930)* were assessed for significance using exact Fisher-Pitman permutation tests through the 'oneway_test' method from the R package 'coin' (*Hothorn et al., 2008*).

## Theoretical estimation of selective advantage and mutation probability in BinK

### Selection coefficient modelling

The analytical approximation developed in *Wahl and Gerrish (2001)* was used to estimate the range of selection coefficients required for a novel beneficial variant to overcome the extinction risk in a population exposed to frequent bottlenecking:

$$V(t,s) \cong 1 - \left( \frac{\ln 2}{2^{t-1}} s\tau \right)$$

Where, $V(t,s)$ represents the probability of extinction given selective coefficient ($s$) and generation ($t$) of growth in which the variant arises. This risk is determined by the number of generations between bottlenecks ($\tau$), selective advantage ($s$), and the generation of arrival ($t$). In the context of the squid–*Vibrio* colonization dynamic, the following values were applied towards these parameters: for the initial host colonization bottleneck following inoculum growth, $\tau$ was 25 generations; for the subsequent venting bottlenecks experienced by symbiont populations, $\tau$ was four generations.

To estimate the minimum selection rate ($r$) conferred by a new rare variant capable of successfully colonizing a host (i.e., comprising one of the ~10 initiating cells [*Wollenberg and Ruby, 2009*; *Altura et al., 2013*]), first we predicted the number of non-synonymous mutations that would accumulate in the *binK* locus during growth of the ancestral population under neutral evolution using the estimated mutation rate for *V. fischeri* (*Dillon et al., 2017*): this was ~325 assuming ~25 generations of cell division to form a final population size of $2.4 \times 10^8$. Then, using the method of *Lenski and Travisano (1994)* for estimating fitness differences in declining populations, selection rates were estimated for the rare variant using the Malthusian parameters (*Fisher, 1930*):

$$M(\text{rare variant}) = \ln(1/325)$$
$$M(\text{wild-type}) = \ln(9/2.4 \times 10^8)$$
$$r = M(\text{rare variant}) - M(\text{wild-type}) = 5.6 \text{ natural logs}$$

Using these approximations, selection coefficients for variants arising during the inoculum's growth phase must be much larger than one in order to attain a reasonable chance of surviving the colonization bottleneck. Conversely, during the venting-regrowth periods,although the probability of a new mutation arising is low, given how comparatively few generations occur during daily re-growth, beneficial alleles with coefficients as low at 0.5 may regularly survive (*Figure 3C*).

A caveat to this approach is that the applied model did not incorporate sub-population dynamics that could result from nuances in the topology of an individual squid's light organ, rather it applies generalized population and growth parameters of a single evolving population through one experimental squid, using data derived from native strain ES114 in the squid–*Vibrio* symbiosis (*Wollenberg and Ruby, 2009*; *Altura et al., 2013*; *Wahl and Gerrish, 2001*). While such population subdivision could potentially facilitate genetic variation among symbionts, it does not affect the estimated selective coefficient of evolved alleles.

### BinK mutation probability modelling

To estimate the probability of a neutral mutation occurring within the *binK* locus during either the inoculum growth phase or during growth cycles in the host, the following parameters were used. References are provided for any parameters based on previously published estimates.

| Parameter | Estimate | Source |
|---|---|---|
| Genome mutation rate | $2.08 \times 10^{-8}$ bp$^{-1}$division$^{-1}$ | *Dillon et al. (2017)* |
| Genome size of MJ11 | 4,323,877 bp | NCBI |
| Available non-synonymous *binK* positions (approximately 2/3 of codon positions) | 2,595 *2/3 | |

| $N_0$ (Inoculum starting population) | 5 cells | |
|---|---|---|
| $N_{inoc}$ (max. population of inoculum prior to dilution) | $2.4 \times 10^8$ cells | |
| $N_{col}$ (V. fischeri founder population size) | 12 (2–3 cells per crypt) | *Nyholm et al. (2000)*; *Wollenberg and Ruby, (2009)*; *Altura et al. (2013)* |
| $N_{host}$ (Juvenile light organ V. fischeri population capacity) | $5 \times 10^5$ cells | *Koch et al. (2014)* |

To place the empirical observations in the context of expectations using the model of *Wahl and Gerrish (2001)*, we predict that mutants carrying a selective advantage of $s \sim 2.8$ would have originated within the first 10 generations of inoculum growth, with the probability of any non-synonymous mutation in the locus occurring within the first 10 generations of inoculum growth being 0.004 (under Poisson). However, the recovery of four distinct *binK* alleles suggests that selection could be much greater than this empirical estimation. Although quantification of the selective advantage is central to understanding the dynamics of natural selection during evolution, obtaining accurate estimates is made more difficult as fitness differentials diverge and become extreme (*Wiser and Lenski, 2015*). We suspect that empirical estimates of *s* using competitive co-inoculations may vastly underestimate the strength of selection in this system, not only because of the extreme and diverging fitness differential between ancestor and evolved strains but also because of the difficulty imposed by the recovery and the challenges of accurate enumeration of rare genotypes.

## Bacterial aggregation

Assessment of the capacities of MJ11 and the *binK1* variant to form cell aggregates in the squid mucus prior to entry through the ducts was conducted as previously described (*Nyholm and McFall-Ngai, 2003*). Briefly, 1.5 hr after newly hatched squid were inoculated with $\sim 10^5$ CFU/ml GFP-labeled strains of interest (harboring pKV111 [*Nyholm et al., 2000*]), squid were incubated in 1 uM CellTracker Orange (Invitrogen) for 30 min, anesthetized in isotonic magnesium chloride and dissected by removing the mantel to expose the intact light organ. Dissected animals were then promptly imaged at 20X and 40X using a Zeiss laser scanning confocal microscope 510. N = 15–20 squid tested per strain.

## Biofilm quantification

Biofilm production was quantified using a standard assay with minor modifications (*O'Toole, 2011*). Briefly, a colony of bacteria from an agar plate was inoculated into either 150 µl (in a Costar 96-well plate) or 2 mL (in a 15 mm glass tube) of SWTO and grown shaking at 200 rpm for 17 hr at 28°C. The biofilm that remained after expulsion of liquid, rinsing, and heat fixation at 80°C for 10 min was stained with 0.1% crystal violet and then decolorized in a volume of 200 µl for assays in plates or 2 mL for tube assays. Biofilm production was determined by absorbance at 550 nm using a Tecan Infinite M200 plate reader. Experiments were performed in triplicate and contained 3–5 biological replicates per treatment. Differences in means were evaluated for significance using a two-sample Fisher-Pitman permutation test conducted using the exact distribution with the 'oneway_test' method from the package coin in R (*Hothorn et al., 2008*).

## Hydrogen peroxide survival

Strains were grown in LBS media at 28°C with shaking at 200 rpm until cultures reach an $OD_{600}$ between 1 and 1.5, the cultures were normalized to an $OD_{600}$ of 1.0 by dilution and 5 µl was subject, in triplicate, to exposure to hydrogen peroxide at different concentrations (ranging from 0.02% to 0.18%) in 200 µl of LBS media in a 96-well Costar polystyrene plate. The minimum concentrations of hydrogen peroxide that restricted all growth (MIC) of wild-type MJ11 and ES114 after over-night incubation was determined for every batch of hydrogen peroxide. Experimental concentrations ranged from 0.02% to 0.18%. Differences in strain survival (binomial outcomes) of at least three combined experiments that contained 106 replicates of strains without plasmids, 15 replicates of $\Delta sypK$ variants that were assayed in conjunction with control strains that lacked the mutation (MJ11, *binK1*, $\Delta binK$) and 50 replicates of strains with plasmids were evaluated for significance using exact Fisher-Pitman permutation tests with the 'oneway_test' method from the R package 'coin' (*Hothorn et al., 2008*). The plasmid harboring pRF2A3 (*binA*) was assayed 20 times in the in same

experiment as control strains that harbored pVSV105 and pCLD48 (*sypE*), which was evaluated in the same way.

## Host hemocyte binding of bacteria

Squid macrophage-like hemocytes were isolated from aposymbiotic hatchling squid using glass adhesion and then stained with Cell Tracker Orange (Invitrogen) suspended in Squid-Ringers, prior to exposure to GFP-labeled *V. fischeri* cells following a previously detailed protocol (*Nyholm et al., 2009*; *Collins and Nyholm, 2010*), with modifications communicated by Dr Bethany Rader. Hemocytes were exposed for one hour to *V. fischeri* strains ES114, MJ11 (*binK+*), MJ11EP2-4-1 (*binK1*) or non-symbiotic *Vibrio harveyi* B392, carrying the GFP plasmid pKV111 (*Nyholm et al., 2000*). To test for the effect of Syp biofilm on hemocyte binding, additional assays were conducted using GFP-labeled strains carrying either control plasmid (pVSV104), *sypE* expression plasmid (pRF2A1), or *binA* expression plasmid (pRF2A4) in addition to GFP plasmid (pKV111) (*Nyholm et al., 2000*) (*Table 1*). Following exposure, hemocyte response to bacteria was visualized at 63x magnification by confocal microscopy and differential interference contrast using a Zeiss LSM 510. Hemocyte binding was quantified by enumeration of bound *Vibrio* relative to total *Vibrio* within a 60 μm radius surrounding each cell. A minimum of 30 hemocyte interactions were quantified per strain. Significant differences in mean proportional binding across strains were detected using a permutation-based test of independence in the R package 'coin' ('independence_test' method, using the exact distribution) (*Hothorn et al., 2008*).

## Siderophore production

Siderophore was measured qualitatively as an orange halo appearing around cells cultured on CAS agar (*Graf and Ruby, 2000*) or from cell free supernatants after 17 hr of growth under iron limited conditions using a chrom-azurol S liquid assay (*Lee and Ruby, 1994a*; *Payne, 1994b*). Colorimetric reduction in $OD_{630}$ was measured in a Tecan Infinite M200 plate reader and % siderohore units were calculated and normalized by cell density (*Lee and Ruby, 1994a*). Siderophore units were below the detection limit for MJ11 and its *binK1* derivative but not ES114.

## Luminescence, homoserine lactone, and cell density determination

Luminescence, cell density and homoserine lactones were quantified from *V. fischeri* MJ11 and variants grown in a starting volume of 15 mL SWT broth culture in a 125 ml flask, which incrementally decreased in volume with sampling. Luminescence produced by the equivalent of 1 mL of culture was quantified on cells diluted up to 1:1000, to ensure that measurements were within the range of detection, with a Turner 20/20 luminometer (Turner Designs, Sunnyvale, CA). Concurrently, the optical density ($OD_{600}$) was determined with a Biophotometer (Eppendorf AG, Hamburg, Germany), with cells diluted into medium. In parallel, colony forming units were determined by standard serial dilution and plating on LBS agar. Published methods were used for the purification and quantification of *N*-(3-oxohexanoyl) homoserine lactone (3-oxo-C6-HSL) and *N*-octanoyl homoserine lactone (C8-HSL) (*Schaefer et al., 2000*; *Duerkop et al., 2007*). Briefly, acyl-HSLs were extracted twice with an equal volume of acidified ethyl acetate from cell-free supernatants of MJ11 and derivatives sampled at a several $OD_{600}$ levels—representing mid-log ($OD_{600}$ ~0.7 and 1.0), late-log (~1.7), early stationary (~3.5), and stationary phase (~5.3–8)—to evaluate the dynamic range of AHL synthesis for each derivative and to determine the optimal $OD_{600}$ during induction. AHLs were extracted and concentrated from 0.5 to 5 mL of MJ11 and variants were detectable and within the assay linear range, identifying that an $OD_{600}$ of ~1.0 was optimal. Replicate experiments were performed in which $OD_{600}$ was monitored at regular intervals, and AHLs were immediately extracted when cultures reached an $OD_{600}$ of 0.9–1.4. Any *binK* derivative culture identified as being dominated by suppressor mutants (i.e., exhibiting an abnormally fast growth rate accompanied by greater than wild-type luminescence and a high proportion of large colonies when plated) were discarded. Extracted samples were concentrated by evaporation under anhydrous nitrogen before analysis. 3-oxo-C6-HSL was quantified using the reporter strain *E. coli* VJS533 harboring plasmid pHV200I⁻, which responds to 3-oxo-C6-HSL by producing luminescence (*Pearson et al., 1994*). C8-HSL was quantified using the reporter strain *E. coli* MG4 harboring pQF50 (*bmaI1-lacZ* promoter fusion derived from *Burkhoderia mallei*) and pJN105 (an arabinose-inducible R gene), which expresses *lacZ* specifically in

response to exogenous C8-HSL with low sensitivity to 3-oxo-C6-HSL (*Duerkop et al., 2007*). LacZ activity was measured by a standard assay (*Miller, 1972*) and using the Dual-Light Luciferase and $\beta$-Galactosidase Reporter Gene Assay System (Applied Biosystems). The amounts of 3-oxo-C6-HSL and C8-HSL were determined by comparing the activity measured from a dilution series of the extracted samples to the linear range ($R^2 \geq 0.98$) of each standard curve generated from synthetic substrates (N-(ß-ketocaproyl)-L-homoserine lactone and N-octanoyl-L-homoserine lactone) (Cayman Chemical). A total of 10 cultures for each derivative from five combined experiments were assayed and reported with the exception of CFU, which was from three cultures. Differences in CFU/mL/$OD_{600}$, $OD_{600}$, nM 3-oxo-C6, nM C8-HSL, and luminescence (Lum) per 1 mL of culture for each variant reported relative to MJ11 were tested for significance using exact Fisher-Pitman permutation tests in the R package 'coin' ('oneway_test' method) (*Hothorn et al., 2008*).

## Metabolic profiling

Phenotype MicroArrays (Biolog, Hayward, CA) PM1 and PM2A were performed according to manufacturers' protocols (*Bochner et al., 2001*) with few modifications for *V. fischeri* analysis, specifically including supplementation of IF-0 with 1% NaCl. Briefly, for each strain, enough inoculum for two replicate plates was prepared by recovering and mixing bacterial colonies into 16 ml IF-0 to obtain a uniform suspension at $OD_{600}$ 0.175 and mixed with dye D mixture (1:5 dilutions). PM1 and PM2A duplicate (ES114, *binK1-* and *ΔbinK*-variants) or triplicate (MJ11 and blank) plates were inoculated with 100 µl of suspension per well, and incubated at 28°C for 48 hr. $OD_{490}$ was recorded by a Tecan Infinite M200 microplate reader every 4 hr to measure kinetic changes in color (redox state) of dye D. To determine which substrates elicited different kinetic responses among strains, we performed an ANOVA on $OD_{490}$ values following normalization against the blank control values for each timed measurement. The significance of strain activity differences for any substrate was determined after correcting for multiple tests using a False Discovery Rate of 0.05. To quantify the overall significance of metabolic responses for MJ11 *binK1* and MJ11 *ΔbinK* converging with ES114 while diverging from MJ11, we used the Exact Binomial Test under the null hypothesis that only 12.5% substrates should yield such a pattern across the four strains assayed ($2*0.5^4$) with the R method 'binom.test'.

## Statistical analyses

Unless otherwise specified, differential responses to colonization and experimental assays for different strains were tested using exact Fisher-Pitman permutation tests with the 'oneway_test' in the R package 'coin' (*Hothorn et al., 2008*). Results from experiments conducted in triplicate were combined by inclusion of a block variable to account for potential technical artefact.

## Acknowledgements

We thank Richard Klobuchar, Chris Payne and the Monterey Bay Aquarium, and Deborah S Millikan for *E. scolopes* specimens; Marcus Dillon, W Kelley Thomas, and Robert Sebra for library preparation and genome sequencing expertise; Spencer Nyholm, Sarah McAnulty and Bethany Rader for guidance in performing hemocyte binding; Karen Visick for insightful guidance on symbiotic polysaccharide studies, strains and constructs; Amy Schaefer for insightful guidance on quorum regulation, and constructs; Matthew Neiditch, Brandon McDonald, Ashley Gagnon, Nicole Clark, and Sarah Martini for technical assistance; and Louis Tisa, Alicia Ballock, Megan Striplin, Evan DaSilva, Feng Xu, Ashley Marcinkiewicz, Mark Mandel, Michelle Nishiguchi, William Soto, Stacia Sower, Kevin Culligan, Philip Gerrish, Caroline Turner, Todd Oakley and David Plachetzki for critical feedback and discussions. Finally, we are grateful for the critical feedback provided by anonymous reviewers, whose insight and suggestions improved the final manuscript. Funding was provided by the National Science Foundation (IOS-1258099) and the New Hampshire Agricultural Experiment Station through the USDA National Institute of Food and Agriculture Hatch program (Accession number 0216015). This is Scientific Contribution Number 2666.

## Additional information

### Funding

| Funder | Grant reference number | Author |
| --- | --- | --- |
| National Science Foundation | IOS-1258099 | Vaughn S Cooper<br>Cheryl A Whistler |
| U.S. Department of Agriculture | 0216015 | Cheryl A Whistler |

The funders had no role in study design, data collection and interpretation, or the decision to submit the work for publication.

### Author contributions

MSP, RLF, Conceptualization, Data curation, Formal analysis, Supervision, Investigation, Visualization, Methodology, Writing—original draft, Writing—review and editing; IMS, Investigation, Writing—review and editing; LAP, BMS, RAD, MC, Validation, Investigation; VSC, Conceptualization, Resources, Data curation, Software, Formal analysis, Supervision, Funding acquisition, Validation, Investigation, Visualization, Methodology, Writing—original draft, Writing—review and editing; CAW, Conceptualization, Resources, Supervision, Funding acquisition, Validation, Investigation, Methodology, Writing—original draft, Project administration, Writing—review and editing

### Author ORCIDs

M Sabrina Pankey, http://orcid.org/0000-0002-7061-9613
Randi L Foxall, http://orcid.org/0000-0003-2396-6695
Cheryl A Whistler, http://orcid.org/0000-0002-2301-2069

## Additional files

### Supplementary files

• Source code 1. Statistical analysis of transcriptome changes in R (Appendix 1, *Figure 5—figure supplement 2*).

• Source code 2. Statistical analysis of metabolic differences in BIOLOG assays in R (Appendix 2).

### Major datasets

The following datasets were generated:

| Author(s) | Year | Dataset title | Dataset URL | Database, license, and accessibility information |
| --- | --- | --- | --- | --- |
| Pankey MS, Foxall RL, Ster IM, Perry LA, Schuster BM, Donner RA, Coyle M, Cooper VS, Whistler CA | 2016 | Genomes of ancestral and evolved Vibrio fisheri | https://www.ncbi.nlm.nih.gov/bioproject/?term=PRJNA316342 | Publicly available at the NCBI BioProject (accession no: PRJNA316342) |
| Pankey MS, Foxall RL, Ster IM, Perry LA, Schuster BM, Donner RA, Coyle M, Cooper VS, Whistler CA | 2016 | Transcriptomes of ancestral, evolved and mutant binK Vibrio fischeri MJ11 | https://www.ncbi.nlm.nih.gov/bioproject/?term=PRJNA316360 | Publicly available at the NCBI BioProject (accession no: PRJNA316360) |

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

## Appendix 1

### Transcriptomic profiling

To gain insight into the breadth of pleiotropic effects of *binK* beyond the biofilm-associated (*Figures 5–7*) and luminescence (*Figure 8*) phenotypes, we used RNAseq to identify patterns in transcriptional differences among *binK* variants from cultures grown to early log phase ($OD_{600}$ 0.25) (see Materials and methods). Although using a low culture density reduced the potential to capture significant transcriptional changes relevant to biofilm production (e.g. Syp) or quorum sensing (e.g. luminescence), it minimized the potential for confounding effects of biofilm differentiation on transcription. The *ΔbinK* mutant had a modest impact on transcription under these conditions. Although most transcripts that were altered by the null mutation were not significantly affected by the *binK1* mutation, 101 out of the 114 of these significant *ΔbinK*-regulated transcripts were expressed by the *binK1* variant at levels intermediate to expression levels in wild-type MJ11 and the null *ΔbinK* mutant (*Appendix 1—table 1*). Expression patterns associated with *binK* variants include the repression of genes involved in cellulose synthesis, carbohydrate glycosylation, and sugar transport and metabolism. The *ΔbinK* mutant also increased transcription of serine and N-acetyl-glucosamine transporter genes. Transcriptional differences also indicated a significant effect of *binK* on iron metabolism and fatty acid biosynthesis pathways associated with quorum-sensing regulation, both of which are important during persistent host colonization (*Davenport et al., 2015*; *Graf and Ruby, 1998*; *Visick et al., 2000*; *Septer et al., 2013*, *Septer et al., 2011*; *Whitehead et al., 2001*). However, siderophore production remained undetectable in *binK* variants as it is in the MJ11 ancestor (*Appendix 1—figure 1*).

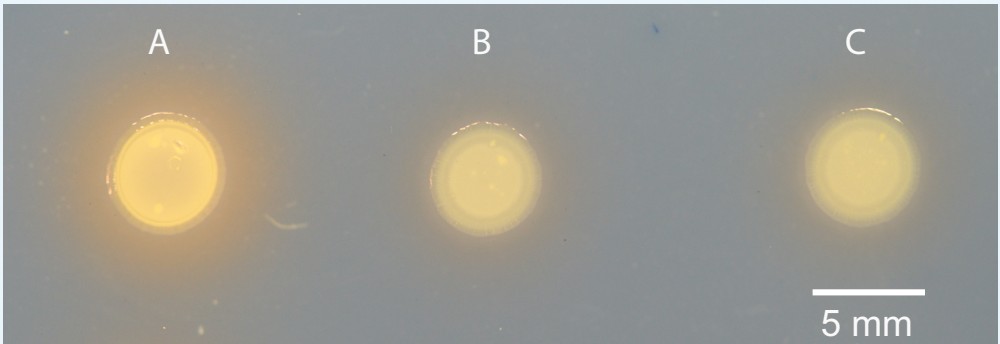

**Appendix 1—figure 1.** Siderophore production in MJ11 and *binK* variants. (**A**) Squid-native ES114, (**B**) squid-naïve MJ11 *binK*+ and (**c**) squid-evolved *binK1* plated on CAS agar.

**Appendix 1—table 1.** Transcript expression differences between wild-type *binK*+ (ancestral MJ11) and *binK* mutants (*ΔbinK* and *binK1*) as detected by RNAseq under Fisher's Exact test in edgeR. Positive fold-changes (logFC) indicate elevated expression in the wild-type relative to the indicated mutant; negative values indicate reduced expression in the wild-type relative to mutants. Loci with similar and significant expression changes in both ΔbinK and *binK1* relative to the wild-type are listed in bold. Only loci showing significant differences in transcript abundance compared with ancestral MJ11 are listed (FDR < 0.05).

| | Wildtype vs mutant ΔbinK | | | Wildtype vs evolved binK1 | | | |
|---|---|---|---|---|---|---|---|
| | Fold-change in expression | Average transcript abundance | Corr. P | Fold-change in expression | Average transcript abundance | Corr. P | |
| Locus Tag | logFC | logCPM | FDR | logFC | logCPM | FDR | Gene description |
| VFMJ11_0008 | −1.4 | 6.79 | 0.0027 | −0.3 | 5.94 | 0.6192 | Cystine-binding protein |
| VFMJ11_0013 | 2.7 | 7.87 | 0.0000 | 1.3 | 8.14 | 0.3143 | Small heat shock protein |
| VFMJ11_0195 | 1.0 | 7.43 | 0.0015 | 0.4 | 7.58 | 0.6346 | Co-chaperonin GroES |
| VFMJ11_0260 | 1.2 | 4.86 | 0.0003 | 0.7 | 4.98 | 0.2770 | Universal stress protein family protein |
| VFMJ11_0297 | −1.4 | 5.51 | 0.0063 | −0.6 | 4.81 | 0.4928 | Sulfite reductase NADPH flavoprotein alpha-component |
| VFMJ11_0307 | −1.4 | 5.27 | 0.0058 | −0.5 | 4.46 | 0.5758 | Sulfate adenylyltransferase subunit 2 |
| VFMJ11_0421 | −0.8 | 6.96 | 0.0011 | −0.6 | 6.68 | 0.2201 | mutY AG-specific adenine glycosylase |
| VFMJ11_0578 | 1.1 | 9.32 | 0.0005 | 0.7 | 9.37 | 0.2725 | ATP-dependent chaperone ClpB |
| VFMJ11_0628 | −1.0 | 9.16 | 0.0009 | −1.0 | 9.03 | 0.0887 | Inositol-1-monophosphatase |
| VFMJ11_0653 | −1.1 | 6.70 | 0.0000 | −0.2 | 5.95 | 0.8114 | Aminobenzoyl-glutamate transport protein |
| VFMJ11_0690 | 1.5 | 3.25 | 0.0001 | 0.6 | 3.53 | 0.5043 | urease accessory protein UreE |
| VFMJ11_0691 | 1.6 | 5.18 | 0.0000 | 0.4 | 5.55 | 0.5650 | Urease subunit alpha UreC |
| VFMJ11_1133 | 1.8 | 3.60 | 0.0000 | 0.6 | 3.94 | 0.3689 | Peptidase T pepT_1 |
| VFMJ11_1137 | 1.1 | 8.56 | 0.0000 | 0.6 | 8.69 | 0.2725 | Glutamate decarboxylase |
| VFMJ11_1138 | 1.5 | 7.03 | 0.0000 | 0.4 | 7.34 | 0.4196 | TrkA domain integral membrane protein |
| VFMJ11_1253 | −0.8 | 9.48 | 0.0100 | −0.6 | 9.21 | 0.3013 | Serine transporter |
| VFMJ11_1268 | −1.5 | 8.45 | 0.0000 | −0.5 | 7.57 | 0.3861 | Insulin-cleaving metalloproteinase outer membrane protein |
| VFMJ11_1269 | −1.2 | 6.24 | 0.0007 | −0.6 | 5.65 | 0.3122 | Thiol oxidoreductase |
| VFMJ11_1270 | −1.0 | 6.40 | 0.0062 | −0.4 | 5.87 | 0.4916 | Imelysin superfamily protein |
| VFMJ11_1305 | −1.4 | 3.64 | 0.0082 | −0.9 | 3.14 | 0.5490 | TonB protein |
| VFMJ11_1317 | 1.2 | 8.36 | 0.0000 | 0.1 | 8.77 | 0.9186 | Hemin receptor |
| **VFMJ11_1370** | **−1.2** | **7.83** | **0.0055** | **−1.4** | **7.82** | **0.0530** | **3-hydroxydecanoyl-ACP dehydratase fabA** |
| **VFMJ11_1398** | **−0.6** | **6.43** | **0.0634** | **−1.0** | **6.50** | **0.0335** | **Na-dependent nucleoside transporter family protein** |
| VFMJ11_1464 | −0.8 | 7.74 | 0.0088 | −0.3 | 7.27 | 0.6058 | Peptidase U32 |
| VFMJ11_1477 | −0.9 | 8.48 | 0.0065 | −1.0 | 8.43 | 0.1214 | Glycine betaine transporter |
| VFMJ11_1534 | −0.8 | 6.27 | 0.0098 | −0.8 | 6.10 | 0.1776 | ATP-dependent RNA helicase RhlE |
| VFMJ11_1579 | −0.8 | 5.34 | 0.0094 | −0.7 | 5.09 | 0.2410 | Amidase |

*Appendix 1—table 1 continued on next page*

*Appendix 1—table 1 continued*

| | Wildtype vs mutant ΔbinK | | | Wildtype vs evolved binK1 | | | |
|---|---|---|---|---|---|---|---|
| | Fold-change in expression | Average transcript abundance | Corr. P | Fold-change in expression | Average transcript abundance | Corr. P | |
| *Locus Tag* | logFC | logCPM | FDR | logFC | logCPM | FDR | Gene description |
| VFMJ11_1614 | 1.2 | 6.67 | 0.0000 | 0.8 | 6.74 | 0.2128 | |
| VFMJ11_1637 | −1.6 | 6.35 | 0.0000 | −1.1 | 5.79 | 0.1727 | Long-chain fatty acid transport protein |
| VFMJ11_1853 | −0.9 | 8.47 | 0.0050 | −0.6 | 8.14 | 0.2725 | |
| VFMJ11_1945 | −1.1 | 10.99 | 0.0001 | −1.0 | 10.79 | 0.1727 | Long-chain fatty acid transport protein |
| **VFMJ11_2039** | **−0.6** | **10.70** | **0.0678** | **−1.0** | **10.90** | **0.0335** | **Nitrate reductase catalytic subunit NapA** |
| VFMJ11_2045 | 1.1 | 5.71 | 0.0078 | 0.5 | 5.85 | 0.4159 | |
| VFMJ11_2103 | −0.9 | 9.12 | 0.0017 | −0.6 | 8.78 | 0.2201 | Queuine tRNA-ribosyl-transferase tgt |
| VFMJ11_2111 | 1.5 | 3.60 | 0.0002 | 1.0 | 3.74 | 0.2591 | Protein YgiW |
| VFMJ11_2127 | −1.0 | 9.24 | 0.0018 | −0.7 | 8.93 | 0.1727 | Peptidase U32 |
| VFMJ11_2165 | 1.2 | 4.76 | 0.0062 | 0.5 | 4.97 | 0.4470 | DNA-binding transcriptional activator CadC |
| VFMJ11_2221 | 1.5 | 9.28 | 0.0000 | 0.1 | 9.80 | 0.9638 | Autonomous glycyl radical cofactor GrcA |
| VFMJ11_2223 | 0.7 | 6.55 | 0.0079 | 0.4 | 6.62 | 0.4482 | Homoserine kinase thrB |
| VFMJ11_2231 | 1.0 | 6.47 | 0.0002 | 0.5 | 6.58 | 0.3889 | Glutamate synthase subunit beta gltD |
| VFMJ11_2259 | −1.0 | 9.66 | 0.0011 | −0.7 | 9.32 | 0.2410 | IronIII ABC transporter periplasmic binding protein |
| VFMJ11_2394 | 1.1 | 5.83 | 0.0058 | 0.4 | 6.03 | 0.6194 | Succinylglutamic semialdehyde dehydrogenase astD |
| VFMJ11_2416 | 1.0 | 9.76 | 0.0032 | −0.3 | 10.32 | 0.6532 | Argininosuccinate synthase argG |
| VFMJ11_2456 | 1.3 | 10.16 | 0.0000 | 0.1 | 10.58 | 0.9472 | Fumarate reductase flavoprotein subunit frdA |
| VFMJ11_2457 | 1.4 | 8.35 | 0.0000 | 0.0 | 8.88 | 1.0000 | Fumarate reductase iron-sulfur subunit |
| VFMJ11_2458 | 1.7 | 6.08 | 0.0000 | 0.4 | 6.50 | 0.7063 | Fumarate reductase subunit C |
| VFMJ11_2459 | 1.4 | 6.68 | 0.0069 | −0.1 | 7.22 | 0.9725 | Fumarate reductase subunit D |
| VFMJ11_2504 | −0.9 | 5.36 | 0.0048 | −0.5 | 4.93 | 0.3940 | 3-dehydroquinate dehydratase aroQ |
| VFMJ11_2505 | −1.0 | 8.61 | 0.0015 | −0.9 | 8.47 | 0.1727 | Acetyl-CoA carboxylase biotin carboxyl carrier protein subunit accB |
| VFMJ11_2506 | −1.2 | 10.80 | 0.0001 | −1.0 | 10.54 | 0.1384 | Acetyl-CoA carboxylase biotin carboxylase subunit accC |

*Appendix 1—table 1 continued*

| | Wildtype vs mutant ΔbinK | | | Wildtype vs evolved binK1 | | | |
|---|---|---|---|---|---|---|---|
| | Fold-change in expression | Average transcript abundance | Corr. P | Fold-change in expression | Average transcript abundance | Corr. P | |
| *Locus Tag* | logFC | logCPM | FDR | logFC | logCPM | FDR | Gene description |
| *VFMJ11_2693* | 0.9 | 5.74 | 0.0083 | 0.5 | 5.80 | 0.3940 | Branched-chain amino acid aminotransferase ilvE |
| *VFMJ11_2696* | −1.6 | 9.00 | 0.0000 | −1.2 | 8.52 | 0.1793 | Cold-shock DNA-binding domain |
| *VFMJ11_A0104* | 2.0 | 4.85 | 0.0000 | 1.1 | 5.07 | 0.1727 | dmsC |
| *VFMJ11_A0105* | 2.0 | 5.16 | 0.0000 | −0.1 | 5.83 | 0.9797 | dmsB |
| *VFMJ11_A0106* | 1.7 | 7.96 | 0.0000 | 0.1 | 8.50 | 0.9472 | Anaerobic dimethyl sulfoxide reductase chain a |
| *VFMJ11_A0107* | 1.2 | 4.47 | 0.0023 | 0.2 | 4.82 | 0.8697 | YnfI |
| *VFMJ11_A0111* | 1.0 | 7.20 | 0.0000 | 0.1 | 7.49 | 0.8523 | Outer membrane protein RomA |
| *VFMJ11_A0151* | 0.9 | 7.29 | 0.0001 | −0.2 | 7.72 | 0.7881 | Putative tripeptide transporter permease tppB |
| *VFMJ11_A0200* | 1.5 | 6.46 | 0.0000 | 0.7 | 6.68 | 0.3479 | L-lysine 6-monooxygenase |
| *VFMJ11_A0201* | 1.6 | 7.97 | 0.0000 | 0.7 | 8.18 | 0.3687 | Ferric aerobactin receptor |
| *VFMJ11_A0222* | 1.3 | 6.35 | 0.0000 | 0.6 | 6.54 | 0.3940 | Trimethylamine-n-oxide reductase 2 |
| *VFMJ11_A0224* | −1.8 | 6.14 | 0.0001 | −1.0 | 5.37 | 0.1748 | FhuE receptor |
| *VFMJ11_A0243* | 0.9 | 5.11 | 0.0075 | 0.2 | 5.35 | 0.7820 | |
| *VFMJ11_A0256* | −1.9 | 1.54 | 0.0091 | −1.6 | 1.03 | 0.3770 | Lipoprotein |
| *VFMJ11_A0280* | 1.2 | 3.75 | 0.0064 | 0.2 | 4.08 | 0.8702 | Methyl-accepting chemotaxis protein |
| *VFMJ11_A0317* | 1.4 | 7.28 | 0.0000 | 0.2 | 7.65 | 0.6304 | Anaerobic ribonucleoside triphosphate reductase |
| *VFMJ11_A0325* | 1.8 | 3.04 | 0.0003 | 1.5 | 3.12 | 0.1801 | YgiW |
| *VFMJ11_A0367* | 2.5 | 4.82 | 0.0000 | 0.6 | 5.32 | 0.3479 | |
| *VFMJ11_A0368* | 2.3 | 6.02 | 0.0000 | 0.9 | 6.36 | 0.1839 | |
| *VFMJ11_A0388* | −0.9 | 5.94 | 0.0049 | 0.4 | 5.09 | 0.4808 | Cyclic nucleotide binding protein |
| *VFMJ11_A0389* | −2.0 | 9.66 | 0.0000 | 0.2 | 8.00 | 0.7303 | Sodium glucose cotransporter |
| *VFMJ11_A0390* | −3.1 | 7.57 | 0.0000 | 0.4 | 4.86 | 0.5426 | UDP-glucose 4-epimerase |
| *VFMJ11_A0391* | −2.4 | 7.09 | 0.0000 | 0.2 | 5.11 | 0.8033 | Galactose-1-phosphate uridylyltransferase |
| *VFMJ11_A0392* | −1.8 | 6.97 | 0.0000 | 0.1 | 5.52 | 1.0000 | Galactokinase |
| *VFMJ11_A0393* | −1.5 | 6.89 | 0.0000 | 0.0 | 5.76 | 1.0000 | Aldose 1-epimerase |
| *VFMJ11_A0394* | −6.3 | 9.93 | 0.0000 | 0.3 | 4.23 | 0.7720 | Transporter AcrB-D-F |

*Appendix 1—table 1 continued*

| | Wildtype vs mutant ΔbinK | | | Wildtype vs evolved binK1 | | | |
|---|---|---|---|---|---|---|---|
| | Fold-change in expression | Average transcript abundance | Corr. P | Fold-change in expression | Average transcript abundance | Corr. P | |
| *Locus Tag* | logFC | logCPM | FDR | logFC | logCPM | FDR | Gene description |
| *VFMJ11_A0395* | −7.0 | 9.15 | 0.0000 | 0.1 | 2.80 | 0.9802 | Acriflavin resistance periplasmic protein |
| *VFMJ11_A0396* | −5.2 | 6.94 | 0.0000 | 0.0 | 2.50 | 0.9472 | Transcriptional regulator TetR family |
| *VFMJ11_A0397* | 5.1 | 4.38 | 0.0000 | 0.6 | 5.10 | 0.2727 | |
| *VFMJ11_A0398* | −1.4 | 7.07 | 0.0000 | 0.0 | 6.06 | 0.9926 | HTH-type transcriptional regulator GalR |
| *VFMJ11_A0408* | 1.5 | 3.36 | 0.0010 | 0.6 | 3.62 | 0.5359 | |
| *VFMJ11_A0487* | −1.4 | 7.99 | 0.0000 | −0.2 | 7.09 | 0.7403 | Pts system N-acetylglucosamine-specific iibc component |
| *VFMJ11_A0619* | −0.7 | 8.61 | 0.0099 | −0.2 | 8.16 | 0.6350 | ABC-type multidrug transport system ATPase and permease component |
| *VFMJ11_A0620* | −1.7 | 7.95 | 0.0000 | −0.9 | 7.22 | 0.1497 | Oxalate-formate antiporter |
| *VFMJ11_A0665* | 2.2 | 2.92 | 0.0000 | 1.2 | 3.19 | 0.3019 | |
| *VFMJ11_A0671* | 1.6 | 2.35 | 0.0051 | 0.9 | 2.53 | 0.4110 | |
| *VFMJ11_A0710* | 1.1 | 4.02 | 0.0082 | 1.0 | 4.01 | 0.2201 | |
| *VFMJ11_A0755* | 1.2 | 3.78 | 0.0020 | 0.8 | 3.87 | 0.2319 | Restriction endonuclease |
| *VFMJ11_A0768* | 1.4 | 2.57 | 0.0029 | 0.8 | 2.73 | 0.5409 | |
| *VFMJ11_A0875* | −1.3 | 7.69 | 0.0000 | −0.1 | 6.77 | 0.9728 | |
| *VFMJ11_A0879* | −0.9 | 6.03 | 0.0017 | −0.7 | 5.74 | 0.4159 | |
| *VFMJ11_A0882* | −1.7 | 4.69 | 0.0000 | −0.7 | 3.84 | 0.3793 | Lipoprotein |
| *VFMJ11_A0885* | −1.4 | 7.51 | 0.0000 | −0.6 | 6.79 | 0.3687 | Cyclopropane-fatty-acyl-phospholipid synthase |
| *VFMJ11_A0887* | −1.3 | 5.63 | 0.0000 | −0.4 | 4.89 | 0.6304 | Amine oxidase |
| *VFMJ11_A0888* | −1.6 | 5.22 | 0.0000 | −0.5 | 4.35 | 0.4916 | Short chain dehydrogenase |
| *VFMJ11_A0890* | −1.2 | 6.07 | 0.0000 | −0.2 | 5.30 | 0.9034 | Transcriptional activator ChrR |
| *VFMJ11_A0891* | −1.2 | 5.70 | 0.0013 | −0.2 | 4.94 | 0.8635 | RNA polymerase sigma factor |
| *VFMJ11_A0909* | −1.5 | 5.52 | 0.0000 | −1.0 | 5.04 | 0.1727 | Ferrichrome-iron receptor |
| *VFMJ11_A1000* | −0.7 | 6.90 | 0.0075 | −0.3 | 6.49 | 0.5856 | Cellulose synthase operon C protein |

*Appendix 1—table 1 continued on next page*

*Appendix 1—table 1 continued*

| | Wildtype vs mutant ∆binK | | | Wildtype vs evolved binK1 | | | |
|---|---|---|---|---|---|---|---|
| | Fold-change in expression | Average transcript abundance | Corr. P | Fold-change in expression | Average transcript abundance | Corr. P | |
| *Locus Tag* | logFC | logCPM | FDR | logFC | logCPM | FDR | **Gene description** |
| *VFMJ11_A1007* | −0.9 | 4.95 | 0.0040 | −0.5 | 4.55 | 0.4061 | Cellulose synthase operon protein YhjU |
| *VFMJ11_A1038* | 1.0 | 3.89 | 0.0040 | 0.7 | 3.94 | 0.2725 | Alkanal monooxygenase beta chain |
| *VFMJ11_A1039* | 1.4 | 4.02 | 0.0000 | 0.7 | 4.22 | 0.4313 | Alkanal monooxygenase alpha chain |
| *VFMJ11_A1040* | 1.7 | 3.16 | 0.0005 | 1.1 | 3.33 | 0.2320 | Acyl transferase |
| *VFMJ11_A1041* | 1.5 | 3.80 | 0.0003 | 0.7 | 4.01 | 0.3851 | Acyl-CoA reductase |
| *VFMJ11_A1048* | 1.0 | 7.25 | 0.0069 | −0.4 | 7.87 | 0.6194 | Carboxypeptidase G2 |
| *VFMJ11_A1058* | −2.8 | 8.75 | 0.0000 | −1.7 | 7.69 | 0.0887 | Pts system fructose-specific eiibc component |
| ***VFMJ11_A1059*** | **−3.0** | **7.69** | **0.0000** | **−1.8** | **6.50** | **0.0073** | **pfkB** |
| ***VFMJ11_A1060*** | **−2.9** | **7.77** | **0.0000** | **−1.6** | **6.52** | **0.0335** | **Bifunctional PTS system fructose-specific transporter subunit IIA Hpr protein** |
| *VFMJ11_A1061* | −2.1 | 4.34 | 0.0000 | −1.2 | 3.44 | 0.1793 | DNA-binding transcriptional regulator FruR |
| *VFMJ11_A1228* | 0.9 | 4.60 | 0.0075 | 0.5 | 4.70 | 0.4704 | |
| *VFMJ11_A1256* | 1.0 | 8.21 | 0.0000 | 0.4 | 8.38 | 0.4150 | Iron-regulated protein FrpC |

Appendix 1—table 1—Source data 1. Read counts estimated by RSEM for chromosome I transcript abundance (Appendix 1, *Figure 5—figure supplement 2*).

Appendix 1—table 1—Source data 2. Read counts estimated by RSEM for chromosome II transcript abundance (Appendix 1, *Figure 5—figure supplement 2*).

**Appendix 2**

## Metabolic profiling

To further examine pleiotropic changes associated with *binK* variants, we employed Biolog assays which measure redox as an indication of metabolic activity in the presence of individually arrayed substrates, as these assays have previously aided in identifying characteristics of experimentally evolved strains (*Soto et al., 2014*). *binK1* moderated metabolic activity in the presence of compounds found in glycans characteristic of eukaryote mucosal epithelia (*Koropatkin et al., 2012*) and in *Vibrio* biofilms (*Appendix 2—table 1*, *Appendix 2—figure 1*, *Appendix 2—figure 2*) (*Visick, 2009*). Compared with MJ11, both the *binK1* and a Δ*binK* derivative decreased redox in the presence of mannose and galactose derivatives, becoming more similar to the metabolism of ES114 (*Appendix 2—figure 1*). Greater substrate utilization in the presence of potentially squid-provisioned chitin and amino acid derivatives by *binK* variants was also congruent with the metabolism of ES114 (D-glucoronic acid, L-glutamine, glucuronamide, galacturonic acid, L-glutamic acid, β-methyl-D-glucoside) (*Graf and Ruby, 1998*; *Wier et al., 2010*; *Schwartzman et al., 2015*). In rare instances, metabolic responses that were altered by the *binK1* allele were not similarly altered by the Δ*binK* mutation (e.g. L-glutamine). Overall, the metabolic response of *binK* variants converged significantly with ES114, with variants responding more like ES114 than ancestral MJ11 for 17% (33/190) of metabolic substrates (Binomial test, p=0.048). Several of these metabolic changes also arose in ES114 following experimental evolution in a novel host, *Euprymna tasmanica* (*Soto et al., 2014*). This convergence supported the hypothesis that pleiotropic effects of the evolved *binK1* allele are adaptive and suggested that responses to these metabolites could contribute to symbiont growth in juvenile squid, and could promote more robust growth in light organs relative to ancestral MJ11.

**Appendix 2—table 1.** Metabolic convergence between squid native *V. fischeri* ES114 and squid-evolved *binK1*. The net changes in metabolic activity (as indicated by change in absorption of the Biolog tetrazolium redox dye) are shown for each *V. fischeri* strain after 48 hr of exposure to each substrate. Only substrates which induced significant (FDR < 0.05) differences across strains are listed. Metabolic changes in each strain relative to wild-type MJ11 *binK*[+] are colored to indicate relatively increased or decreased activity. Of the 190 substrates tested, 44 substrates yielded significant differences across strain, including 39 which indicate congruent metabolic responses by ES114 and *binK1* (Exact binomial test, p=1.405e-7).

| Well | Substrate | Metabolic activity (ΔA550 over 48 hr) | | | | Metabolic activity change relative to remS[+] MJ11 | | | | Convergence |
|------|-----------|---------|--------|--------|-------|---------|--------|--------|--------|-------------|
| | | *remS*[+] | *remS1* | Δ*remS* | ES114 | *remS*[+] | *remS1* | Δ*remS* | ES114 | |
| H11 | Phenylethylamine | 0.012 | 0.568 | 0.458 | 0.667 | 0.000 | 46.54 | 37.33 | 54.85 | + |
| H07 | Glucuronamide | 0.017 | 0.523 | 0.564 | 0.558 | 0.000 | 30.20 | 32.69 | 32.30 | + |
| G10 | Methyl pyruvate | 0.019 | 0.677 | 0.462 | 0.639 | 0.000 | 33.70 | 22.70 | 31.78 | + |
| H08 | Pyruvic acid | 0.013 | 0.187 | 0.276 | 0.395 | 0.000 | 13.09 | 19.76 | 28.71 | + |
| E01 | L-Glutamine | 0.026 | 0.620 | 0.125 | 0.665 | 0.000 | 22.82 | 3.78 | 24.54 | + |
| F03 | m-Inositol | 0.044 | 0.724 | 0.726 | 0.671 | 0.000 | 15.50 | 15.54 | 14.28 | + |
| E02 | m-Tartaric acid | 0.026 | 0.424 | 0.338 | 0.451 | 0.000 | 15.10 | 11.84 | 16.13 | + |
| D02 | D-Aspartic acid | 0.040 | 0.459 | 0.363 | 0.735 | 0.000 | 10.56 | 8.16 | 17.55 | + |
| A12 | Dulcitol | 0.030 | 0.402 | 0.091 | 0.608 | 0.000 | 12.61 | 2.07 | 19.63 | + |
| G03 | L-Serine | 0.032 | 0.467 | 0.235 | 0.360 | 0.000 | 13.66 | 6.39 | 10.29 | + |
| H02 | p-Hydroxy phenyl acetic acid | 0.027 | 0.063 | 0.037 | 0.636 | 0.000 | 1.27 | 0.36 | 22.11 | + |

*Appendix 2—table 1 continued on next page*

*Appendix 2—table 1 continued*

| Well | Substrate | Metabolic activity (ΔA550 over 48 hr) | | | | Metabolic activity change relative to remS⁺ MJ11 | | | | Convergence |
| | | remS⁺ | remS1 | ΔremS | ES114 | remS⁺ | remS1 | ΔremS | ES114 | |
|---|---|---|---|---|---|---|---|---|---|---|
| B06 | D-Gluconic acid | 0.048 | 0.300 | 0.324 | 0.628 | 0.000 | 5.24 | 5.75 | 12.06 | + |
| B09 | L-Lactic acid | 0.029 | 0.068 | 0.040 | 0.647 | 0.000 | 1.30 | 0.37 | 20.93 | + |
| E09 | Adonitol | 0.026 | 0.358 | 0.197 | 0.085 | 0.000 | 12.88 | 6.63 | 2.28 | + |
| H01 | Glycyl-L-proline | 0.039 | 0.206 | 0.258 | 0.443 | 0.000 | 4.32 | 5.65 | 10.44 | + |
| C05 | Tween 20 | 0.025 | 0.065 | 0.001 | 0.517 | 0.000 | 1.60 | −0.95 | 19.54 | + |
| E08 | β-Methyl-D-glucoside | 0.004 | 0.024 | 0.022 | 0.021 | 0.000 | 5.66 | 5.10 | 4.85 | + |
| G05 | L-Alanine | 0.056 | 0.322 | 0.295 | 0.355 | 0.000 | 4.74 | 4.25 | 5.33 | + |
| B11 | D-Mannitol | 0.018 | 0.177 | 0.085 | 0.034 | 0.000 | 8.64 | 3.64 | 0.88 | + |
| H09 | L-Galactonic acid—Lactone | 0.089 | 0.275 | 0.379 | 0.450 | 0.000 | 2.10 | 3.27 | 4.06 | + |
| F04 | D-Threonine | 0.017 | 0.126 | 0.044 | 0.041 | 0.000 | 6.37 | 1.55 | 1.37 | + |
| D01 | L-Asparagine | 0.026 | 0.069 | 0.080 | 0.129 | 0.000 | 1.61 | 2.02 | 3.86 | + |
| H06 | L-Lyxose | 0.088 | 0.051 | 0.036 | 0.481 | 0.000 | −0.42 | −0.59 | 4.48 | - |
| F8 | Mucic acid | 0.026 | 0.072 | 0.044 | 0.035 | 0.000 | 1.78 | 0.68 | 0.36 | + |
| C12 | Thymidine | 0.071 | 0.168 | 0.116 | 0.052 | 0.000 | 1.36 | 0.63 | −0.27 | - |
| G11 | D-Malic acid | 0.028 | 0.062 | 0.036 | 0.029 | 0.000 | 1.21 | 0.31 | 0.05 | + |
| F06 | Bromo succinic acid | 0.033 | 0.061 | 0.037 | 0.035 | 0.000 | 0.82 | 0.11 | 0.04 | + |
| A10 | D-Trehalose | 0.031 | 0.045 | 0.038 | 0.034 | 0.000 | 0.45 | 0.21 | 0.08 | + |
| D06 | α-Keto-glutaric acid | 0.045 | 0.074 | 0.043 | 0.042 | 0.000 | 0.65 | −0.04 | −0.08 | - |
| F9 | Glycolic acid | 0.039 | 0.062 | 0.032 | 0.040 | 0.000 | 0.61 | −0.17 | 0.04 | + |
| C11 | D-melibiose | 0.028 | 0.043 | −0.006 | 0.053 | 0.000 | 0.53 | −1.20 | 0.90 | + |
| D10 | Lactulose | 0.045 | 0.057 | 0.046 | 0.035 | 0.000 | 0.27 | 0.02 | −0.20 | - |
| A10 | Laminarin | 0.678 | 0.546 | 0.674 | 0.798 | 0.000 | −0.20 | −0.01 | 0.18 | - |
| E06 | 2-Hydroxy benzoic acid | 0.089 | 0.070 | 0.080 | 0.093 | 0.000 | −0.21 | −0.10 | 0.04 | + |
| A03 | α-Cyclodextrin | 0.191 | 0.122 | 0.089 | 0.158 | 0.000 | −0.36 | −0.54 | −0.17 | + |
| H07 | D,L-Octopamine | 0.200 | 0.111 | 0.067 | 0.186 | 0.000 | −0.45 | −0.66 | −0.07 | + |
| F07 | D-Ribono-1,4-lactone | 0.198 | 0.085 | 0.070 | 0.162 | 0.000 | −0.57 | −0.65 | −0.18 | + |
| D07 | Turanose | 0.188 | 0.065 | 0.060 | 0.137 | 0.000 | −0.66 | −0.68 | −0.27 | + |
| E02 | Caproic acid | 0.241 | 0.101 | 0.007 | 0.215 | 0.000 | −0.58 | −0.97 | −0.11 | + |
| G10 | L-Leucine | 0.214 | 0.051 | 0.075 | 0.135 | 0.000 | −0.76 | −0.65 | −0.37 | + |
| G02 | L-Alaninamide | 0.164 | 0.065 | 0.047 | 0.045 | 0.000 | −0.60 | −0.72 | −0.72 | + |
| G02 | Tricarballylic acid | 0.029 | 0.018 | −0.008 | 0.006 | 0.000 | −0.37 | −1.27 | −0.80 | + |
| C10 | α-Methyl-D-mannoside | 0.183 | 0.004 | 0.024 | 0.075 | 0.000 | −0.98 | −0.87 | −0.59 | + |
| D08 | α-Methyl-D- Galactoside | −0.011 | 0.018 | 0.008 | 0.000 | 0.000 | −2.56 | −1.68 | −0.99 | + |

Appendix 2—table 1—Source data 1. Raw data for redox activity over 48 hr in BIOLOG plate PM1 (Appendix 2).

**Appendix 2—table 1—Source data 2.** Raw data for redox activity over 48 hr in BIOLOG plate PM2A (Appendix 2).

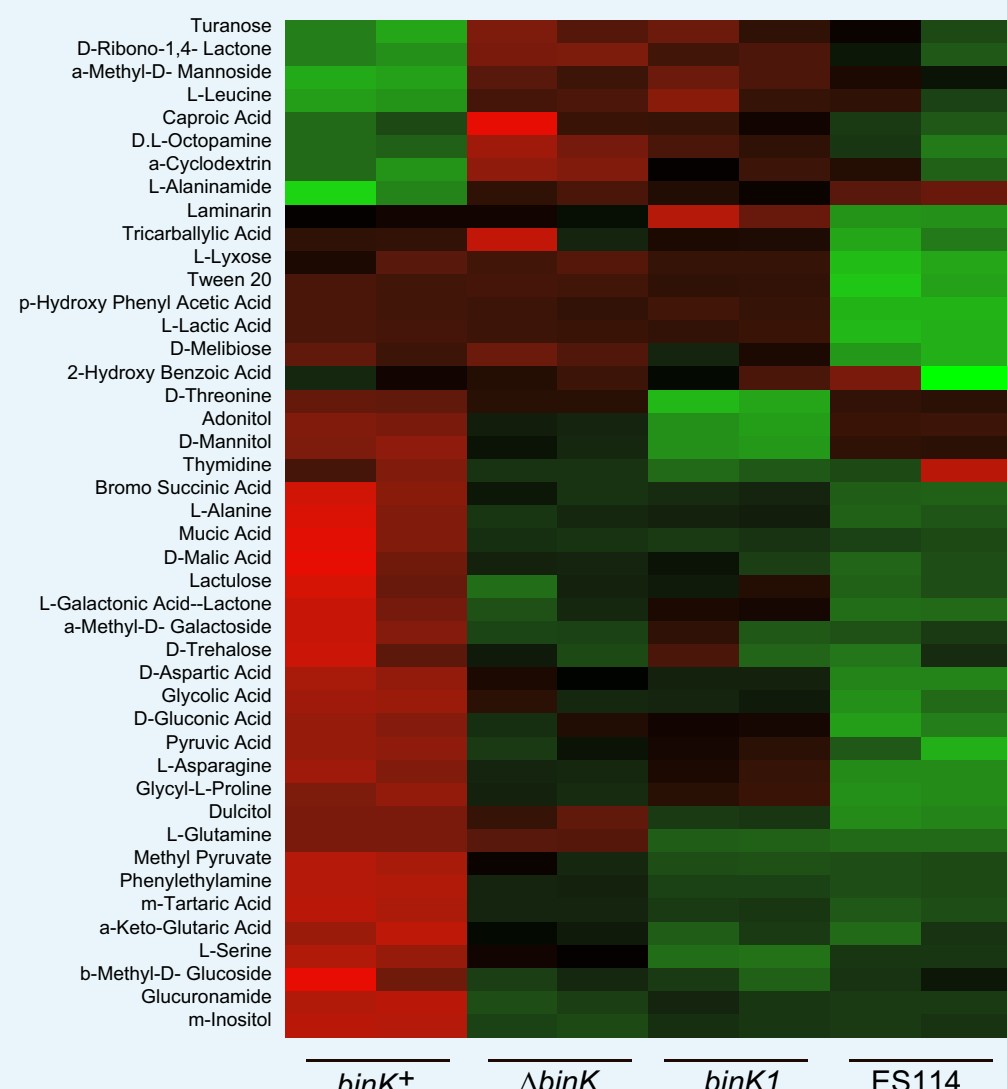

**Appendix 2—figure 1.** Metabolic shifts associated with *binK* variants. Significantly differing metabolic responses to BIOLOG compounds for wild-type MJ11 (*binK*+), squid-adapted MJ11 *binK1*, MJ11 Δ*binK and* squid-native ES114. Responses to all tested compounds are reported in the Figure Supplement.

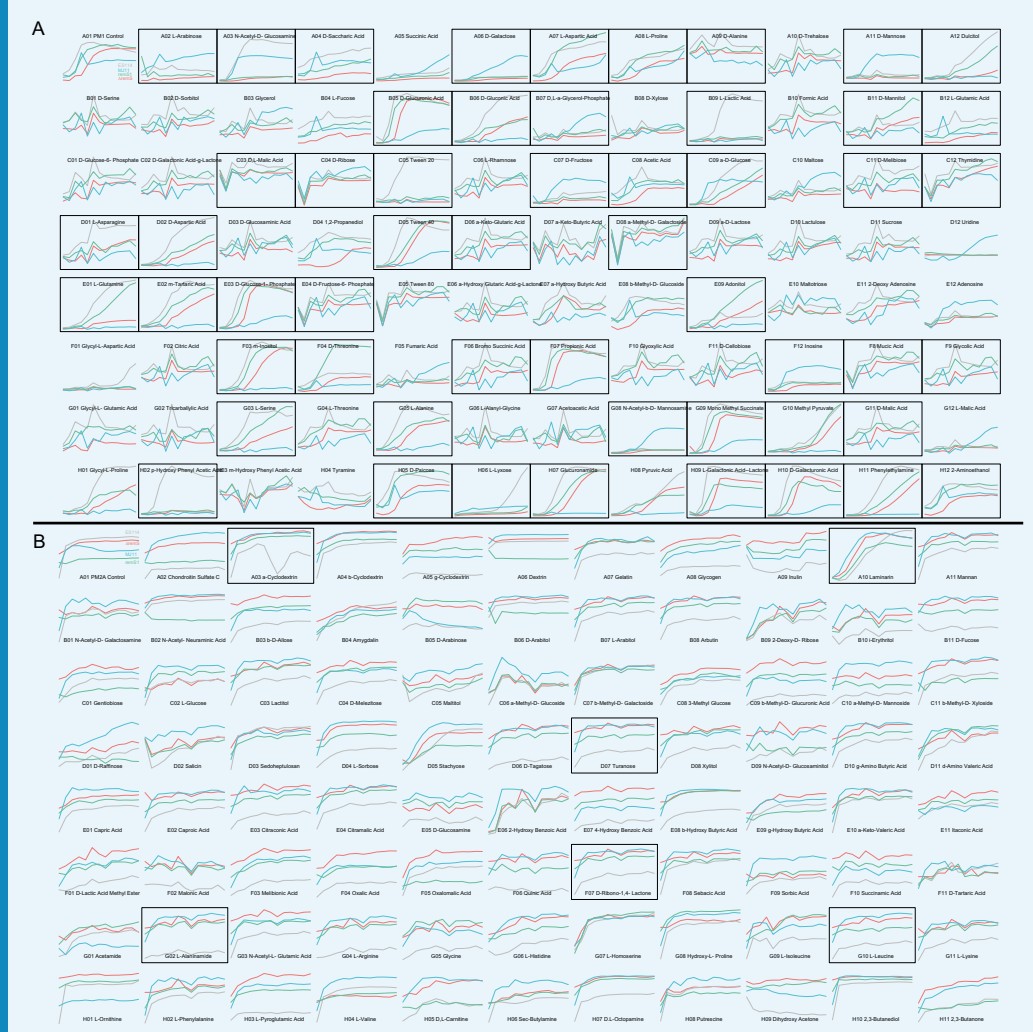

**Appendix 2—figure 2.** Metabolic profiles using BIOLOG phenotyping assays. Plots enclosed by boxes indicate substrates that are significantly differentially metabolized across strains (listed in *Table 2*). X-axis represents time (0–48 hr); Y-axis represents metabolic activity as detected by BIOLOG redox (tetrazolium) dye absorbance ($OD_{490}$).

