## [Decision Letter]

Thank you for submitting your article "Host-selected mutations converging on a global regulator drive an adaptive leap by bacteria to symbiosis" for consideration by *eLife*. Your article has been reviewed by two peer reviewers, and the evaluation has been overseen by a guest Reviewing Editor and Detlef Weigel as the Senior Editor. The reviewers have opted to remain anonymous.

The reviewers have discussed the reviews with one another and we have drafted this decision to help you prepare a revised submission. The verbatim comments of the reviewers are attached as well.

Summary:

This is an impressive example of bacterial experimental evolution in a host setting, identifying mutational change of a regulator as key for symbiotic association. It is the first description of adaptive significance of particular *binK* alleles for *V. fischeri* symbiosis initiation and persistence in *E. scolopes*, based on work with in vivo squid-host evolution system. The standard is exemplary: multiple lines of evolution, re-sequencing of original and full sequencing of all resulting strains, etc.

Essential revisions:

The major weakness of the work is that it potentially aims to do too much. The transcriptome (and several other) sections appear ancillary and distract from the core findings. These should either be removed or (preferably) moved to supplementary material. Similarly, *binK* phenotypes that do not connect to the rest of the story should be left out of this paper, and published once there is a clearer understanding of them. Such focusing will also help pare down the 10 pages of Discussion, which contains far too much speculation, mostly about the weakest findings.

Additional points: The reviewers raise many points regarding the presentation and Discussion (see below). Please pay attention to these in your revision, but note that there is no need to enumerate all your responses in the "response to reviewer" letter that will accompany your revision, only those where you disagree with the reviewers and are planning not to follow their advice.

*Reviewer #1:*

This article is appropriate for publication in *eLife*. I am in favor of publication because the article is the first description for the adaptive significance of particular *binK* alleles for *V. fischeri* symbiosis initiation and persistence in *E. scolopes*. The authors leverage their experimental, in vivo squid-host evolution system to arrive at this result; this application is a relatively novel and clever use of the system and should be a standard by which these sorts of experiments are performed (multiple lines of evolution, re-sequencing of original and full sequencing of all resulting strains, etc.). Furthermore, the authors have done a thorough job of characterizing multiple effects that a particular *binK* allele has on symbiosis-associated characteristics in the system – pushing knowledge of the importance of regulation of biofilm-associated phenotypes ahead in this system specifically and host-bacterial symbiosis in general.

The above written, the authors could be much more careful in their writing and editing of this manuscript. This is a long manuscript, and there are many editorial errors (some by omission). Also, although it is nice to have all of the data in one place, the story appears to be a potpourri of findings and overlong. I recommend that the authors streamline and narrow the focus of this story (for example, by leaving out the transcriptomics data, perhaps).

Furthermore, before publication the authors should include a Methods sub-section explicitly describing their statistical tests, the corrections for multiple comparisons, if any, and a more thorough explanation for technical replicates vs. experimental replicates. In the manuscript body, it is sometimes unclear, and always a bit opaque as to, in general, which calls/flags from the coin package were used in R to conduct each test. From the authors' descriptions in the "transparent reporting form" I was unsure if these particular permutation tests, like parametric tests, require corrections when multiple comparisons from the same experiment are completed. I understand why the authors may have chosen to use coin (for example, data non-normality or philosophical opinions about drawing from true, representative subpopulations, perhaps), however, the reader has to infer this and may be a bit confused in general as to these tests (or, for example why the authors choose to use them in all cases except one – see subsection “The large selective advantage conferred by squid-adapted *binK* improved fitness during both the initiation and maintenance stages of symbiosis, consistent with theoretical predictions”, first paragraph). I urge the authors to clarify their statistical reasoning/philosophy and methodology for the readers before publication. Finally, many experiments appear to contain technical replicates and/or experimental replicates, but few describe what these are and how they were analyzed appropriately via the tests above – for example, many lines (see subsection “Luminescence, homoserine lactone, and cell density determination” for an example) imply that data were all lumped together for technical/experimental replicates.

One final minor note: the authors' use neutral assumptions for their modeling of in-squid dynamics, but the argument is also made that the *binK* locus is demonstrably *not* under neutral conditions after initiation – how are these two statements reconciled in the authors' minds?

*Reviewer #2:*

This is an impressive example of bacterial experimental evolution in a host setting, identifying mutational change of a regulator as key for symbiotic association. A single gene, encoding a sensor kinase, is identified to confer the ability to colonize the host and remain in stable association over several generations, and the downstream effect of the sensor kinase on biofilm (EPS) formation, response of the innate immune system, quorum sensing, and metabolism are elucidated to provide a mechanistic explanation for the genetic result. The authors have conducted a very comprehensive piece of work, including proper controls (strain evolution in culture vs. in symbiosis; knockouts; transcriptomic and metabolism profiling).

I have only a few comments that need to be addressed prior to publication:

Introduction, last paragraph and Figure 2: when describing the experimental setup, the "neutral" /negative control (evolution of MJ11 in pure culture without selection) should be introduced; it is shown in Figure 2 but not mentioned anywhere at this point, so in order to understand the figure and controls, please introduce early in the text.

Figure 2: has the potential to explain the outcome of the experimental evolution in a nutshell but needs serious improvement: symbols are often way too small and the legend does not explain all the information hidden in the figure. Fx the *binK* "colored dots" in A and C are hard to distinguish (I cannot figure out, which MJ11 mutation occurred twice), even when zooming in, the "host symbols" are hard to recognize, and panel B does not work at all (see also comment below). The structural model (panel C) is poorly explained (I assume the faint grey bar depicts the membrane, so Cache is in the periplasm?), and the scoring matrix is not discussed any further and thus a strange way to suggest that the 4 occurring mutations are not functionally neutral – what is the point to display all other possibilities as well?

One question that arises (and that could have been answered in Figure 2) is whether the *binK* evolution results in a convergence of MJ11-evolved BinK to the *binK* variant of the native strain ES114? Or does it just somewhat lose functionality? Please address.

Figure 4: is this a conceptual figure or based on actual data? Please make clear.

Figure 9: please define "mean expression per locus": how was that calculated?please use colored labels for compounds that are clearly distinguishable at this size; please indicate whether the 4 replicates are technical or biological; please add gene symbols to the coding loci where possible; I would especially appreciate to add the genes discussed earlier in the study to the heat map, i.e. *binA, sypE, sypK, lux*, etc.

Subsection “Squid-adapted *binK* confers metabolic convergence with native symbionts”, last paragraph: Biolog does not measure redox. There's a redox dye that indicates substrate utilization, i.e. it is reduced and results in color precipitation. But saying "greater redox" is not correct.

---

## [Author Response]

*Essential revisions:*

*The major weakness of the work is that it potentially aims to do too much. The transcriptome (and several other) sections appear ancillary and distract from the core findings. These should either be removed or (preferably) moved to supplementary material. Similarly, binK phenotypes that do not connect to the rest of the story should be left out of this paper, and published once there is a clearer understanding of them. Such focusing will also help pare down the 10 pages of Discussion, which contains far too much speculation, mostly about the weakest findings.*

The reviewers raised concerns about datasets that distract from our core findings, and those identified as potentially tangential included the transcriptome and metabolism data. We agree, these datasets are not critical for most of the conclusions, especially regarding specific mechanisms of adaptation, and as such are suitable for the supplementary material. We have moved the major sections of text describing those results to the supplement. We have moved the transcriptomics heatmap to Figure 5—figure supplement 2 (and summary table to Supplementary file 1) due to the fact that this data provides important corroboration that cellulose is a regulatory target of BinK. The Biolog metabolic profiles are now in a Supplementary file 2. Due to the substantial fitness gain attained during sustained colonization (Figure 4), and improved yield in light organs (Figure 3—figure supplement 2), we feel the metabolic data should still be included in the manuscript as this data showing convergent metabolism with the native symbiont aligns with the documented fitness gains, demonstrates the global effects of the regulator, and provides additional context for the altered function of the evolved allele that is necessary for understanding why this regulator confers such an extraordinary fitness gain that cannot be explained by initiation phenotypes alone.

*Reviewer #1:*

*This article is appropriate for publication in eLife. I am in favor of publication because the article is the first description for the adaptive significance of particular binK alleles for V. fischeri symbiosis initiation and persistence in E. scolopes. The authors leverage their experimental, in vivo squid-host evolution system to arrive at this result; this application is a relatively novel and clever use of the system and should be a standard by which these sorts of experiments are performed (multiple lines of evolution, re-sequencing of original and full sequencing of all resulting strains, etc.). Furthermore, the authors have done a thorough job of characterizing multiple effects that a particular binK allele has on symbiosis-associated characteristics in the system – pushing knowledge of the importance of regulation of biofilm-associated phenotypes ahead in this system specifically and host-bacterial symbiosis in general.*

*The above written, the authors could be much more careful in their writing and editing of this manuscript. This is a long manuscript, and there are many editorial errors (some by omission). Also, although it is nice to have all of the data in one place, the story appears to be a potpourri of findings and overlong. I recommend that the authors streamline and narrow the focus of this story (for example, by leaving out the transcriptomics data, perhaps).*

*Furthermore, before publication the authors should include a Methods sub-section explicitly describing their statistical tests, the corrections for multiple comparisons, if any, and a more thorough explanation for technical replicates vs. experimental replicates. In the manuscript body, it is sometimes unclear, and always a bit opaque as to, in general, which calls/flags from the coin package were used in R to conduct each test. From the authors' descriptions in the "transparent reporting form" I was unsure if these particular permutation tests, like parametric tests, require corrections when multiple comparisons from the same experiment are completed. I understand why the authors may have chosen to use coin (for example, data non-normality or philosophical opinions about drawing from true, representative subpopulations, perhaps), however, the reader has to infer this and may be a bit confused in general as to these tests (or, for example why the authors choose to use them in all cases except one – see subsection “The large selective advantage conferred by squid-adapted binK improved fitness during both the initiation and maintenance stages of symbiosis, consistent with theoretical predictions”, first paragraph). I urge the authors to clarify their statistical reasoning/philosophy and methodology for the readers before publication. Finally, many experiments appear to contain technical replicates and/or experimental replicates, but few describe what these are and how they were analyzed appropriately via the tests above – for example, many lines (see subsection “Luminescence, homoserine lactone, and cell density determination” for an example) imply that data were all lumped together for technical/experimental replicates.*

We have added a Methods subsection to clarify the statistical tests used for assays, including how data from replicated experiments was combined and blocked for possible artefact. Use of a block factor obviates the need for multiple-test correction, as only a single p-value is calculated on the variable of interest. Further, for each Methods section, we have reported the R package and method used. To be consistent, we have also now replaced the only parametric test with the exact Fisher-Pitman permutation test as used elsewhere.

*One final minor note: the authors' use neutral assumptions for their modeling of in-squid dynamics, but the argument is also made that the binK locus is demonstrably not under neutral conditions after initiation – how are these two statements reconciled in the authors' minds?*

The model does not assume neutrality. We had included an estimate of the number of non-synonymous *binK* alleles that could theoretically arise under neutral evolution, but we have removed this estimate (as it does not affect the model or our interpretation) to prevent confusion.

*Reviewer #2:*

*[…] I have only a few comments that need to be addressed prior to publication:*

*Introduction, last paragraph and Figure 2: when describing the experimental setup, the "neutral" /negative control (evolution of MJ11 in pure culture without selection) should be introduced; it is shown in Figure 2 but not mentioned anywhere at this point, so in order to understand the figure and controls, please introduce early in the text.*

Corrected.

*Figure 2: has the potential to explain the outcome of the experimental evolution in a nutshell but needs serious improvement: symbols are often way too small and the legend does not explain all the information hidden in the figure. Fx the binK "colored dots" in A and C are hard to distinguish (I cannot figure out, which MJ11 mutation occurred twice), even when zooming in, the "host symbols" are hard to recognize, and panel B does not work at all (see also comment below). The structural model (panel C) is poorly explained (I assume the faint grey bar depicts the membrane, so Cache is in the periplasm?), and the scoring matrix is not discussed any further and thus a strange way to suggest that the 4 occurring mutations are not functionally neutral – what is the point to display all other possibilities as well?*

We have replaced panels B and C with a single panel that better conveys the message that mutations in these functional domains are rare in wild strains, including ES114, and thus are more likely to confer functional changes. We have also modified the text to better integrate the PSSM results.

*One question that arises (and that could have been answered in Figure 2) is whether the binK evolution results in a convergence of MJ11-evolved BinK to the binK variant of the native strain ES114? Or does it just somewhat lose functionality? Please address.*

ES114 and MJ11 wild-type are highly conserved across these sites. We have addressed this with the new panel 2B. We also specifically clarify in the text that these are not convergent with the native strain ES114.

*Figure 4: is this a conceptual figure or based on actual data? Please make clear.*

We have clarified that 4A is conceptual in the legend.

*Figure 9: please define "mean expression per locus": how was that calculated?please use colored labels for compounds that are clearly distinguishable at this size; please indicate whether the 4 replicates are technical or biological; please add gene symbols to the coding loci where possible; I would especially appreciate to add the genes discussed earlier in the study to the heat map, i.e. binA, sypE, sypK, lux, etc.*

We have updated the legend to indicate biological replication and to explain how expression scaling was calculated. The exact steps are also available in an R source file provided during submission. Labels for the genes of interest here are also now provided.

*Subsection “Squid-adapted binK confers metabolic convergence with native symbionts”, last paragraph: Biolog does not measure redox. There's a redox dye that indicates substrate utilization, i.e. it is reduced and results in color precipitation. But saying "greater redox" is not correct.*

Corrected.